# Arctic Climate Response to European Radiative Forcing: A Deep Learning Study on Circulation Pattern Changes

Sina Mehrdad[1], Dörthe Handorf[2], Ines Höschel[2], Khalil Karami[1], Johannes Quaas[1,3], Sudhakar Dipu[1], and Christoph Jacobi[1]

[1]Leipzig Institute for Meteorology, Leipzig University, Stephanstr. 3, 04103 Leipzig, Germany
[2]Alfred Wegener Institute for Polar and Marine Research, Research Unit Potsdam, D-14473Potsdam, Telegrafenberg A43, Germany
[3]ScaDS.AI - Center for Scalable Data Analytics and Artificial Intelligence, Leipzig University, Humboldtstraße 25, 04105 Leipzig, Germany

**Correspondence:** Sina Mehrdad (sina.mehrdad@uni-leipzig.de)

**Abstract.** Heterogeneous radiative forcing in mid-latitudes, such as that exerted by aerosols, has been found to affect the Arctic climate, though the mechanisms remain debated. In this study, we leverage Deep Learning (DL) techniques to explore the complex response of the Arctic climate system to local radiative forcing over Europe. We conducted sensitivity experiments using the Max Planck Institute Earth System Model (MPI-ESM1.2) coupled with atmosphere-ocean–land surface components. Large-scale circulation patterns can mediate the impact of the forcing on Arctic climate dynamics. We employed a DL-based clustering approach to classify large-scale atmospheric circulation patterns. To enhance the analysis of how these patterns impact the Arctic climate, the Poleward Moist Static Energy Transport (PMSET) associated with the atmospheric circulation patterns was incorporated as an additional similarity metric in the clustering process. Furthermore, we developed a novel method to analyze the circulation patterns' contributions to various climatic parameter anomalies. Our findings indicate that the negative radiative forcing over Europe alters existing circulation patterns and their occurrence frequency without introducing new ones. Specifically, our analysis revealed that while the regional radiative forcing alters the occurrence frequencies of the circulation patterns, these changes are not the primary drivers of the forcing's impact on the Arctic parameters. Instead, it is the shifts in the mean spatial characteristics of the atmospheric circulation patterns, induced by the forcing, that predominantly determine the effects on the Arctic climate. Our methodology facilitates the uncovering of complex, nonlinear interactions within the climate system, capturing nuances that are often obscured in broader seasonal anomaly analyses. This approach enables a deeper understanding of the dynamics driving observed climatic anomalies and their links to specific atmospheric circulation patterns.

## 1 Introduction

The Arctic region is experiencing a faster warming rate in comparison to the global average, as evident from empirical observations and climate modeling studies (Hahn et al., 2021; Rantanen et al., 2022; Wendisch et al., 2022). This phenomenon, known as Arctic amplification (AA), represents a fundamental characteristic of the Earth's climate. A complex interplay of

both local and remote processes contributes to the strong warming of the Arctic (Henderson et al., 2021; Taylor et al., 2022). Large-scale circulation governs the remote processes affecting AA, particularly the energy transport into the Arctic (Graversen, 2006; Graversen and Burtu, 2016; Naakka et al., 2019; Mewes and Jacobi, 2019; Nygård et al., 2020; Henderson et al., 2021) and can be influenced by AA (Jaiser et al., 2012; Rinke et al., 2017; Crasemann et al., 2017; Vavrus, 2018; Cohen et al., 2018). The atmospheric heat transport into the Arctic can contribute to AA through intricate interactions with local processes such as sea ice feedback, temperature feedback, and cloud feedback (Morrison et al., 2012; Pithan and Mauritsen, 2014; Taylor et al., 2022; Linke et al., 2023). The extent of these interactions is determined by various factors, including the spatiotemporal pattern of energy transport into the Arctic, which plays a significant role (Woods and Caballero, 2016; Pithan et al., 2018; Papritz and Dunn-Sigouin, 2020). Moreover, several studies have argued that AA, particularly sea ice loss, can affect the occurrence of preferred circulation patterns (Crasemann et al., 2017; Handorf et al., 2017; Vavrus, 2018), further underscoring the complex and interconnected nature of AA.

Heterogeneous radiative forcing, such as that resulting from aerosol, can induce direct changes in atmospheric circulation by altering the vertical and horizontal thermal structure of the atmosphere, thereby also leading to subsequent modifications in energy transport into the Arctic (Alexeev et al., 2005; Shindell and Faluvegi, 2009; Hannachi et al., 2017; Krishnan et al., 2020). The complex and nonlinear response of Arctic temperature to regional heterogeneous radiative forcing is more pronounced compared to the global temperature (Shindell and Faluvegi, 2009; Najafi et al., 2015; Persad and Caldeira, 2018; Lewinschal et al., 2019). In particular, the increase in anthropogenic aerosol emissions, predominantly from the European and North American sectors during the period of 1940-1980, has been postulated to have caused a cooling trend in Arctic temperatures and a concomitant increase in the sea ice concentration (SIC) (Fyfe et al., 2013; Gagné et al., 2017). Since the 1980s, air quality regulations have led to a reduction in aerosol burden over Europe and North America, which has contributed to AA (Acosta Navarro et al., 2016; Gagné et al., 2017). Acosta Navarro et al. (2016) pointed out that the decline of European aerosol emissions since the 1980s has contributed to AA by augmenting the top of atmosphere (TOA) net shortwave flux in the Arctic and enhancing energy transport to the Arctic during the summer season.

In this study, we apply the concept of atmospheric circulation regimes to understand changes in large-scale circulation. AA has been observed to be associated with changes in the frequency of occurrence of preferred large-scale atmospheric circulation patterns, which, in turn, influence mid-latitude weather conditions (Crasemann et al., 2017). The stratospheric pathway has been postulated as a plausible mechanism underlying Arctic-mid-latitude linkage (Jaiser et al., 2012; Nakamura et al., 2016; Cohen et al., 2018; Dethloff et al., 2019; Liang et al., 2024). It suggests that in the Barents-Kara Sea, the delayed sea ice refreezing during autumn increases the surface heat flux, from the ocean to the atmosphere and, consequently leads to a warming of the lower troposphere and enhanced sea-level pressure over the Ural region (Kim et al., 2014; Kug et al., 2015) and more frequent and persistent Ural blocking (Yao et al., 2018). Additionally, enhanced oceanic moisture release to the Arctic atmosphere in these months can contribute to an increase in Siberian snow cover and the strengthening of the Siberian high (Gastineau et al., 2017; Dethloff et al., 2019). The diabatic heating of the lower troposphere above the Barents and Kara seas combined with increased sea-level pressure over the Ural region, intensifies the forcing of planetary Rossby waves. These waves constructively interfere with the climatological waves (Honda et al., 2009; Outten et al., 2022), resulting

in an increased potential to enhance the vertical propagation of planetary waves, thereby disrupting the polar vortex (Jaiser et al., 2016; Gastineau et al., 2017; Henderson et al., 2018; Köhler et al., 2023; Xu et al., 2023). The downward propagation of the stratospheric circulation anomalies, associated with the disrupted polar vortex, then feeds back into the tropospheric circulation in late winter/spring, leading to an increase in the occurrence of the negative phase of the North Atlantic Oscillation (NAO) in late winter/spring (Kim et al., 2014; Jaiser et al., 2016; Nakamura et al., 2016; Sun et al., 2022; Liang et al., 2024). However, the stratospheric polar vortex response is relatively small compared to its internal variability, influencing the signal-to-noise ratio of the introduced forcing by sea ice on the weakening of the polar vortex (Sun et al., 2022). Besides, the proposed stratospheric pathway is also identified to show intermittency (Siew et al., 2020) and to exhibit greater strength in observations than the climate models, which further casts doubt on the role of sea ice in initiating this pathway (Cohen et al., 2020). An alternative hypothesis for the Arctic-mid-latitude linkage posits that the atmospheric circulation patterns and their associated teleconnections drive Arctic warming, sea ice loss, snow anomalies, and the weakening of the polar vortex by enhancing the vertical propagation of planetary waves (Henderson et al., 2018; Peings, 2019; Blackport et al., 2019; Blackport and Screen, 2021). Thus, the preferred circulation patterns and their teleconnections are suggested to play a crucial role in AA and its linkage to mid-latitudes.

Deep learning (DL), a branch of machine learning, has experienced remarkable success in recent years, attaining cutting-edge performance across numerous fields (LeCun et al., 2015). It has demonstrated efficacy in many Earth system applications, including data-driven weather prediction (Weyn et al., 2019, 2020), extreme event detection (Liu et al., 2016; Racah et al., 2017), weather pattern classification (Chattopadhyay et al., 2020; Mittermeier et al., 2022), and various other tasks (Huntingford et al., 2019; Irrgang et al., 2021). DL algorithms, particularly convolutional neural networks (CNNs), excel at learning complex nonlinear relationships within data and are considered to hold great promise for addressing challenging problems in Earth science characterized by datasets encompassing expansive spatiotemporal patterns (LeCun et al., 2015; Racah et al., 2017; Reichstein et al., 2019; Rolnick et al., 2022). Numerous cutting-edge DL architectures for visual pattern recognition utilize CNNs, owing to their ability to perform hierarchical feature learning (Liu et al., 2016; Racah et al., 2017; Rawat and Wang, 2017; Reichstein et al., 2019). Although machine learning algorithms can effectively learn underlying relationships within data, they do not consistently adhere to physical principles in their predictions (Reichstein et al., 2019; Kashinath et al., 2021). Leveraging the flexibility of machine learning algorithms, physics-informed machine learning frameworks were developed to address this issue by incorporating domain knowledge into the machine learning algorithms (Kashinath et al., 2021). Physics-informed machine learning, a category of robust machine learning techniques, has begun to be successfully applied in numerous Earth system tasks (Kashinath et al., 2021).

The intricate nature of AA and its debated association with mid-latitudes underscores the importance of understanding the fundamental mechanisms involved in this climatic phenomenon. In an attempt to analyze the effect of the regional radiative forcing over Europe on AA and its linkage to mid-latitudes, we have employed coupled ocean-atmospheric simulations with local radiative forcing over Europe with preindustrial background and initial conditions. We developed a DL algorithm, drawing upon physics-informed machine learning approaches, to investigate the impacts of regional radiative forcing on circulation patterns. We used DL algorithms to detect a possible anomalous response of the Northern Hemisphere extratropical large-scale

circulation to the exerted forcing and to perform clustering on the circulation regime considering the poleward energy transport pattern associated with the circulation pattern. Moreover, in our analysis, we put an emphasis on the stratospheric pathway and the role of the polar vortex in colder seasons, given their intrinsic connection to the Arctic's predominant climatic conditions.

## 2 Data and methodology

In this section, we describe the simulations, data, and analysis methods used in this paper.

### 2.1 Model simulations

We used the Max Planck Institute Earth System Model version 1.2 (MPI-ESM1.2) coupled atmosphere-ocean–land surface model. The atmospheric component of the MPI-ESM1.2 is based on the European Centre Hamburg Model version 6 (ECHAM6) (Stevens et al., 2013) with T63L47 spectral resolution (approximately 1.8° horizontal resolution) and 47 levels, and the model top at $0.01\,\mathrm{hPa}$. The atmospheric component also incorporates the JSBACH land-surface scheme (Reick et al., 2013; Stevens et al., 2013). The ocean and sea ice components are combined in the Max Planck Institute Ocean Model (MPIOM) (Jungclaus et al., 2013), which is applied to an idealized control mapping grid of about 1.5° with 40 levels. The 30-year model simulations were conducted with pre-industrial boundary conditions and atmospheric composition and were initialized using pre-existing pre-industrial equilibrium simulations. The pre-industrial climate is selected because of the availability of a balanced equilibrium state between the atmosphere and the ocean. Moreover, using the pre-industrial climate for initial and boundary conditions facilitates the sensitivity analysis as the only transient perturbation is imposed by sustained cloud modification.

We conducted two 30-year sensitivity experiments to examine the impact of regional radiative forcing over Europe on the Northern Hemisphere extratropical large-scale circulation and its subsequent effects on key variables such as Arctic sea ice cover, 2m temperature, and stratospheric dynamics. First, a single ensemble Control run was performed. In the second simulation (Experiment run), regional negative radiative forcing was imposed by cloud modification. This modification was designed to regionally alter radiation similar to aerosol forcing. The cloud modification was sustained throughout the simulation and limited to a rectangular region over Europe (43° N to 58° N and 5° E to 25° E). The cloud modification was implemented as an alteration to cloud water content ($q_l$) with a scaling factor of 10, and the cloud ice content ($q_i$) was multiplied by 0.1 in the model. The resulting optically thick boundary layer clouds ($q_l \times 10$) exert a negative radiative effect by reflecting solar radiation (Twomey, 1977), and by removing or thinning ($q_i \times 0.1$), the high-level clouds exert a negative radiative effect by transmitting the terrestrial radiation back to space (Mitchell and Finnegan, 2009). The cloud modification only affected the radiation process in the simulation and did not directly influence other processes. The Experiment run was performed with six ensemble members, and the ensemble mean was used throughout the analysis. The ensemble members were formed by a small perturbation in the atmospheric initial conditions while sharing the same external forcing. Figure 1 shows the mean net radiative flux anomaly at the TOA relative to the Control run. The figure shows that the regional cloud modification leads to a negative radiative forcing over Europe, which yields a magnitude of -18 ± 6 W/m$^2$. Areas with statistically significant

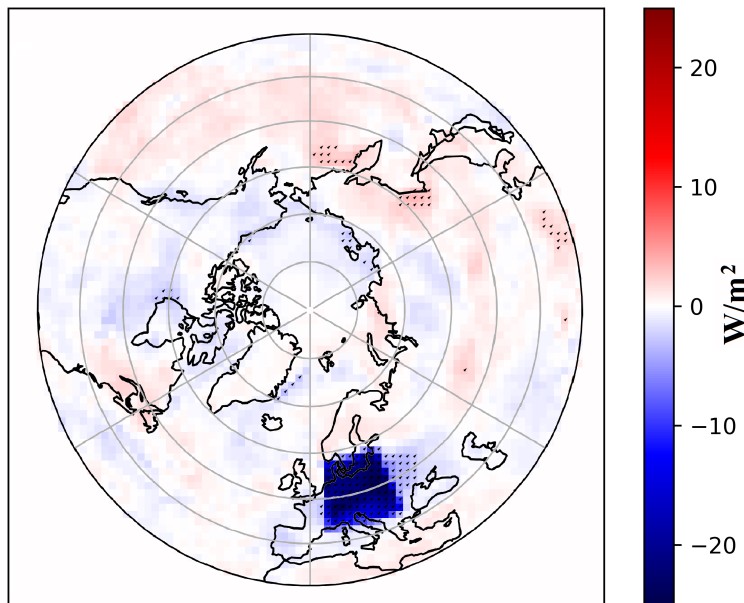

**Figure 1.** Top of the atmosphere (TOA) effective radiative forcing for latitudes poleward of 30° N calculated as the Experiment run (E) minus Control run (C) difference between mean TOA net radiative flux. Areas with statistically significant differences are dotted.

differences between the Experiment and Control runs TOA net radiative flux are dotted in Figure 1. The independent two-
sample t-test and Welch's unpaired t-test were used throughout the study to determine the statistical significance levels when samples have equal or unequal variances, respectively, and samples with p-value less than 0.05 were considered statistically significant.

## 2.2  Data

We used the daily mean sea level pressure (MSLP) and 700 to 300 hPa layer thickness ($\tau_{300-700}$) fields to analyze the Northern
Hemisphere extratropical large-scale circulation. Other fields like the SIC and 2m temperature were also used in this study.

Concerning atmospheric energy transport, the vertically integrated poleward moist static energy transport (PMSET) has been calculated for the upper and lower troposphere using the data fields of the meridional wind component ($v$), temperature ($T$), geopotential height ($z$), specific humidity ($q$), and pressure ($p$):

$$PMSET = \frac{1}{g} \int_{p_l}^{p_u} v(\eta) \left( c_p T(\eta) + gz(\eta) + Lq(\eta) \right) \frac{\partial p}{\partial \eta} d\eta, \tag{1}$$

where $g$ is the gravitational acceleration (taken as $9.81 \, \mathrm{ms^{-2}}$), $c_p$ represents the specific heat capacity of air at constant pressure ($1 \, \mathrm{kJkg^{-1}K^{-1}}$), $L$ is the latent heat of vaporization of water ($2500 \, \mathrm{kJkg^{-1}}$), and $\eta$ represents the model levels. The

lower ($p_l$) and upper ($p_u$) boundaries for lower and upper troposphere PMSET are surface pressure and $600\,\mathrm{hPa}$, and $600\,\mathrm{hPa}$ and $200\,\mathrm{hPa}$, respectively.

We derive the wave-mean flow interactions by exploiting the transformed Eulerian mean (TEM) equations to investigate the middle atmosphere dynamics (Andrews et al., 1987). The quasi-geostrophic Eliassen-Palm (EP) flux has been calculated from daily data fields. EP flux and its divergence describe the large-scale wave propagation using the eddy heat and momentum fluxes.

## 2.3 Deep learning (DL) approach

DL algorithms (LeCun et al., 2015) were used to analyze the Northern Hemisphere extratropical large-scale circulation response to local radiative forcing over Europe. We mainly used convolutional neural network (CNN) layers (Rawat and Wang, 2017) with rectified linear units (ReLU) as the activation function (Agarap, 2018) in our DL architecture. The Keras open-source python library (Chollet et al., 2015) with Google's TensorFlow backend (Abadi et al., 2016) was used to implement the DL algorithms. In this study, we utilized DL algorithms both as anomaly-detecting and feature-extracting tools, the specifics of which will be elaborated upon in the following sections.

### 2.3.1 Input data

The MSLP and $\tau_{300-700}$ fields were used as input to our DL algorithms. These two fields deliver pertinent information on the state and dynamics of the circulation regime. Adding $\tau_{300-700}$ enriches the dataset with complementary information about the growth and decay of the weather systems and improves the performance of data-derived weather predicting systems (Weyn et al., 2019, 2020). Although adding further data fields might strengthen the information content of the dataset, we used the two data fields of MSLP and $\tau_{300-700}$ to limit the DL algorithms' complexity and training computational costs.

As we mainly focus on analyzing the Northern Hemisphere extratropical large-scale circulation, only the data poleward of $30°\,\mathrm{N}$ were used. We transferred the MSLP and $\tau_{300-700}$ fields to the Lambert azimuthal equal-area projection (Snyder, 1987) with a $1.5°$ horizontal grid size, which is close to the MPI-ESM original grid size. The data fields in this projection are well suited to be analyzed with CNN layers as they preserve the scale of the weather systems regardless of their location. An example of the daily MSLP and $\tau_{300-700}$ fields is shown in Figure 2. We normalized both data fields to a range between 0 and 1 before feeding them to our DL algorithms. The normalized data fields are more compatible with the ReLU activation function. To normalize the data fields, we used their respective maximum and minimum values over all simulations.

Our DL algorithms input data points were defined as four-dimensional arrays of (time steps, x, y, channel). The "time steps" dimension refers to the number of successive daily mean fields stacking together in a data point. "x" and "y" are the projection axes (Figure 2), and "channel" determines the MSLP or $\tau_{300-700}$ data. The "time steps" dimension greater than two (at least two successive days) provides additional information about the dynamic processes of the input data points. Setting the "time steps" parameter to 8 days yielded the best performance regarding the DL algorithm reconstruction loss and overfitting measure. We generated the corresponding data point for each day in the Control and Experiment runs by stacking its normalized daily fields to its seven successive days' data fields. We assigned the date of the first day of a data point to the whole data point.

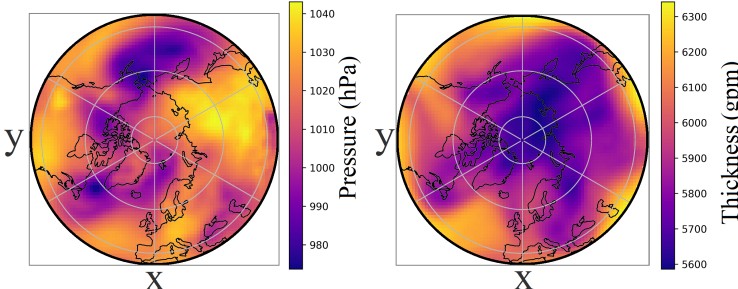

**Figure 2.** An arbitrary example of daily mean fields for a single day (day 1869.01.21, 19 years and 20 days after the start date of the experiments in 1850.01.01) in the Control run of MSLP (left panel) and $\tau_{300-700}$ (right panel) for latitudes poleward of 30° N in Lambert azimuthal equal-area projection.

### 2.3.2 Convolutional auto-encoder

Different DL algorithms were used in this study. First, a DL algorithm was used to detect possible anomalous circulation regimes in the Experiment run relative to the Control run. Further DL algorithms were used as feature-extracting methods for circulation regime clustering. The auto-encoder framework has been used as the core of all our DL algorithms (Baldi, 2012). An auto-encoder consists of an encoder and a decoder. The encoder projects an input data point into a lower-dimensional representation, the latent space. The decoder reconstructs the input data by projecting it back from the latent space into the input representation.

We split our 30-year dataset into a training and validation set. Roughly 27% of the whole dataset was used for validation, while the rest was used for training the DL algorithms. Starting from the second year of both simulations, we labeled the full-year data for every 4th year as the validation data points. Thus, the validation dataset represented both the annual cycle and long-term evolution.

We used Adam optimizer (Kingma and Ba, 2014) and reconstruction loss as the optimizer and the loss function for the training process, respectively. The reconstruction loss was calculated as the mean squared difference between the autoencoder's reconstructed output and the corresponding input data points. After initial unrestricted 200 epochs of training, an early stopping condition was used to determine the optimum time for stopping the training process. The training process is stopped if the validation loss, the reconstruction loss for the validation set, will not reach a new minimum after 50 epochs of training, and the auto-encoder final weights corresponding to the validation loss minimum are restored. Using the initial 200 epochs of training and early stopping minimized the validation loss while preventing the auto-encoders from overfitting.

Different auto-encoder architectures using various combinations of convolutional long short-term memory (LSTM) (Shi et al., 2015), dense, 2d, and 3d CNN layers (Tran et al., 2015; Rawat and Wang, 2017) were evaluated in preliminary analyses. The auto-encoders' final reconstruction loss over the whole dataset and the overfitting measure were used to assess their performance. For each auto-encoder, the overfitting measure is calculated by subtracting the auto-encoders final training loss

(the reconstruction loss for the training set) from its validation loss, normalized by the validation loss. The auto-encoder consisting mainly of 3D CCN layers performed better than the others, considering overall reconstruction loss and overfitting measure. Therefore, we used this architecture as the core of our DL algorithms. The encoder and decoder in our DL algorithms mainly consist of 3D CCN layers with $3 \times 3 \times 3$ kernels (Tran et al., 2015). We set the "same" padding for each 3D CNN layer. These layers are suitable for detecting the spatiotemporal pattern in a dataset with a time dimension. The ReLU activation function was used for all layers in the encoders and decoders in our DL algorithms, except for the last layers, where the sigmoid activation function was used.

The encoder and decoder with convolutional layers of representation will be referred to as convolutional encoder and decoder, respectively. Convolutional auto-encoders with successive layers of 3D CNN can capture the spatiotemporal patterns in a dataset as the hierarchy of the patterns with increasing scale in both the spatial and temporal domains. In the training process, neighboring local spatiotemporal patterns formed in the first layer's kernels can merge and create higher-level patterns in the next layer. The higher-level spatiotemporal patterns in each layer are formed by combining the patterns in the last layer. Hence, the depth of the convolutional layers determines the scale of the spatiotemporal patterns detected by the encoder. This approach to analyzing the data is suitable for investigating the atmospheric circulation regime and its synoptic-scale dynamic since the large-scale atmospheric processes consist of several processes at smaller scales. Furthermore, the patterns inherently contain the global processes as they are trained on the large-scale global domain.

## 2.4 Anomaly detection

An auto-encoder with a latent space dimension of 25600 was used for detecting the possible anomalous large-scale circulation response to the negative radiative forcing over Europe. The auto-encoder architecture is presented in Table A1 and referred to as AAE (Anomaly detecting Auto-Encoder). Only the Control run data were used for training the AAE, which consequently will only learn the spatiotemporal patterns in the Control run data. Therefore, it is expected to produce a higher reconstruction loss when fed with a data point containing unfamiliar spatiotemporal patterns. We, therefore, interpreted the reconstruction loss produced by feeding data points from the Experiment run to the AAE as a measure to quantify the presence of new spatiotemporal patterns in the data.

The reconstruction loss generated by feeding data points from both the Experiment and Control runs into the AAE is presented in Figure 3. Figure 3 shows that the reconstruction loss for both runs follows a bimodal distribution. The higher peak is predominantly attributed to summer, while the lower peak corresponds to winter, with autumn and spring lying in between. This indicates that the AAE exhibits greater skills in reproducing the data points for winter, suggesting that it better memorizes spatiotemporal patterns for colder winter months than summer patterns.

The reconstruction loss distribution is similar for both Experiment and Control runs, with the Control run distribution exhibiting a longer tail at lower values. We attribute this tail at lower values to the overfitting of the AAE. Although the AAE has not been exposed to the Experiment run data points, it can reproduce them as accurately as the Control run data points on which it has been trained. This implies that the negative forcing over Europe may not introduce new discernable spatiotemporal patterns in the Northern Hemisphere extratropical large-scale circulation. Moreover, training an auto-encoder exclusively

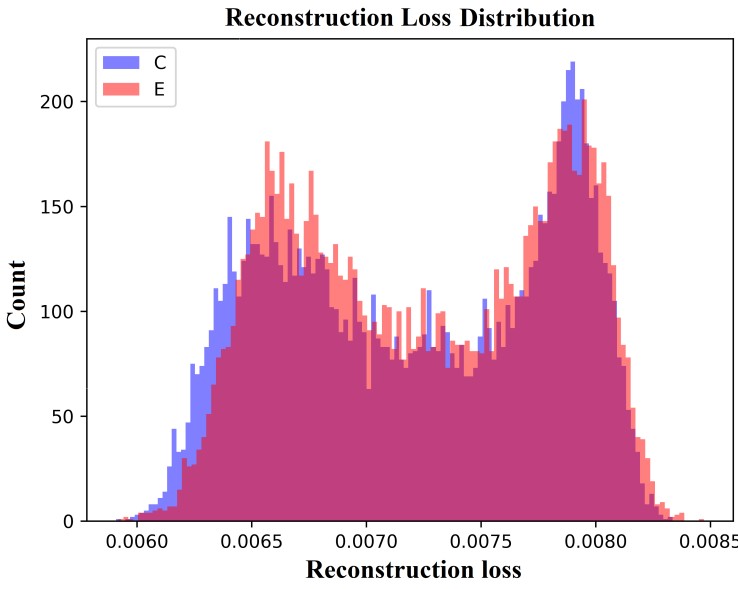

**Figure 3.** Reconstruction loss distribution for the Control (C) and Experiment (E) runs generated by the AAE.

with data points from the Experiment runs reveals the interchangeability of spatiotemporal patterns between the Control and Experiment runs. For more detailed information, we refer to Appendix A.

## 2.5 Feature extraction

In this study, we utilized the latent spaces of two distinct auto-encoders as feature spaces for clustering. One auto-encoder's latent space was formed without constraints, while the other was specifically constrained by PMSET indices. The following subsections provide detailed explanations of these methodologies.

### 2.5.1 PMSET profiles and indices

The PMSET longitudinal profiles for the upper and lower troposphere were used to investigate the poleward energy transport associated with the atmospheric circulation regimes. For each data point, 8-day mean profiles of upper and lower troposphere PMSET were calculated at the four latitudes 60°, 66°, 70°, and 75° N. Thus, for each data point, we have calculated eight profiles. Figure 4 shows an example of the lower troposphere PMSET longitudinal profile at 66° N. We further extracted five indices, namely the zonal mean PMSET, the maximum and minimum PMSET, and the maxium and minimum PMSET longitudes from each longitudinal profile (see Figure 4). So, for each data point, we have 40 indices comprehensively describing the associated three-dimensional PMSET spatial pattern. The normalized indices were used in our clustering method.

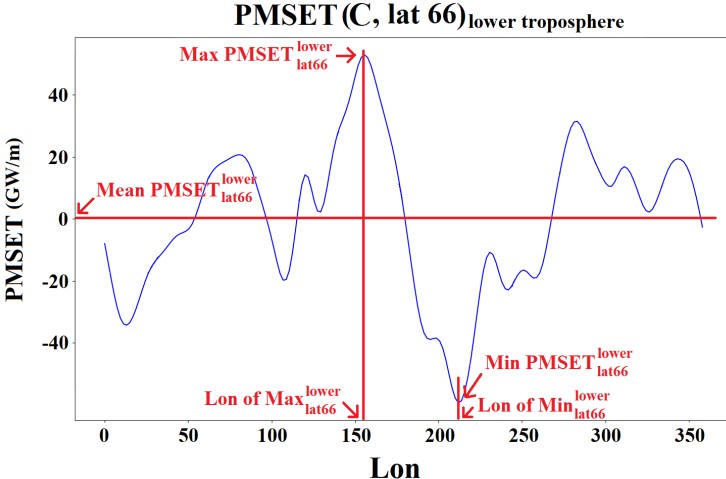

**Figure 4.** An example of the longitudinal profile of lower troposphere (below 600 hPa) PMSET at latitude 66° N for a data point in the Control run. Five indices extracted from the PMSET profiles are highlighted in the figure. The mean value is two orders of magnitude smaller than the variability.

### 2.5.2 Clustering

We used k-means clustering with prescribed 6 clusters to categorize all data points. We found 6 to be the optimal number of clusters (see Appendix B). The k-means clustering was performed on the 100-dimensional latent space of an auto-encoder. Hereafter, this auto-encoder will be referred to as MCAE (main clustering auto-encoder). The MCAE architecture is presented in Table A1. The MCAE was trained using both Control and Experiment data points to capture the spatiotemporal patterns in both simulations. We incorporated the knowledge regarding the data points' PMSET into the MCAE. In other words, the MCAE has been trained so that the first 40 dimensions of its 100-dimensional latent space converge to the 40 PMSET indices. The mean squared difference between the PMSET indices and the first 40 dimensions of the MCAE's latent space was added to the reconstruction loss with the same weight in the training process. No limitation was applied to the other 60 dimensions, and they were freely established in the training process. Hence, the latent space of the MCAE consists of 40 PMSET indices and 60 free dimensions.

We projected all the data points to the MCAE's latent space and performed the 6-class k-means clustering on their latent space representation. However, before clustering, it was crucial to address the circular nature of the angular features in PMSET indices to ensure continuity between angles close to 0 and 360 degrees. To achieve this, we transformed each angular feature in the MCAE's latent space into two dimensions using the sine and cosine of the corresponding angle. This procedure ensured that all angles close to each other in the circular space would also be proximate in the latent space used for clustering (see Appendix C).

The k-means clustering then categorizes the data points based on their Euclidean distance in the latent space representation. Performing the clustering in the latent space is more suitable for our analysis as we are interested in classifying the circulation

regime regarding their associated poleward energy transport. Given the PMSET indices, data points with similar PMSET patterns are located close to each other in the latent space. Furthermore, the 60 free dimensions allow the MCAE to determine these dimensions based on spatiotemporal patterns in the data set. Thus, the data points having similar spatiotemporal and associated PMSET patterns are close together in the MCAE's latent space.

In addition, to evaluate our clustering method compared to other prevalent methods and to gain deeper insight into the different data representations, we implemented five distinct clustering strategies. First, we categorized data points into six classes using the Self-Organized Map (SOM) method (Kohonen, 1990; Skific et al., 2009; Lee, 2017; Mewes and Jacobi, 2019). Each input data point in data space has $8 \times 82 \times 82 \times 2 = 107,584$ dimensions (see Section 2.3.1). Additionally, we applied the 6-class k-means clustering method directly to the data representation. Furthermore, we utilized an Empirical Orthogonal Function (EOF) analysis for dimensionality reduction (Hannachi et al., 2007; Crasemann et al., 2017). We applied a 6-class k-means clustering on the time series of principal components (PCs) within the dimensionally reduced space. Another approach involved performing the 6-class k-means clustering on a representation consisting solely of PMSET indices. The circular nature of the angular features was addressed prior to clustering. In this representation, the proximity of data points is determined solely by the similarities in their associated PMSET patterns. Finally, we used the latent space of an auto-encoder similar to AAE as the data representation (see Table A1) to perform the k-means clustering. This auto-encoder was called FAE (free auto-encoder). FAE is similar to AAE except that it is trained on all data points. The FAE's latent space has a dimension number of 25,600, and the latent space was established in the training process without any limitation. The data points with similar spatiotemporal patterns are positioned closer in this representation. The results of these clustering methods can be found in Appendix D. Comparing these clustering methodologies, along with our proposed approach, offers a comprehensive insight into different data representations, and provides an understanding of different methods' effectiveness.

The latent space of the FAE is assessed to determine its capability to reproduce the mean of the data points. In our analysis, the FAE demonstrated the best performance with the validation loss of $3.84 \times 10^{-5}$ and the overfitting measure of 6.37. First, we computed the mean for the whole data points in the data space and the FAE latent space. The latent space mean is then transferred back to the data space using the FAE's decoder to form the reconstructed mean. The corresponding differences between the data space mean and the reconstructed mean yield standard deviations of 0.46 hPa for the MSLP field and 4.74 gpm for the $\tau_{300-700}$ field. Figure 5 shows the average (over the time dimension) of the reconstructed mean, the average of the data space mean, and their difference. Figure 5 demonstrates the congruence of spatial patterns between the data space mean and the reconstructed mean for both MSLP and $\tau_{300-700}$ fields. This similarity substantiates our ability to accurately reconstruct the data space mean using the mean of the latent space.

## 2.6 Class contribution

During clustering, the data points have been categorized into 6 clusters based on their similarities in spatiotemporal and associated PMSET patterns. We evaluated the mean contribution of each cluster to the seasonal anomaly of the respective analyzed parameters, such as SIC, 2m temperature, EP flux, and zonal mean zonal wind. We calculated the seasonal anomaly of a parameter by subtracting its seasonal mean over the Experiment data points from its seasonal mean over the Control data points.

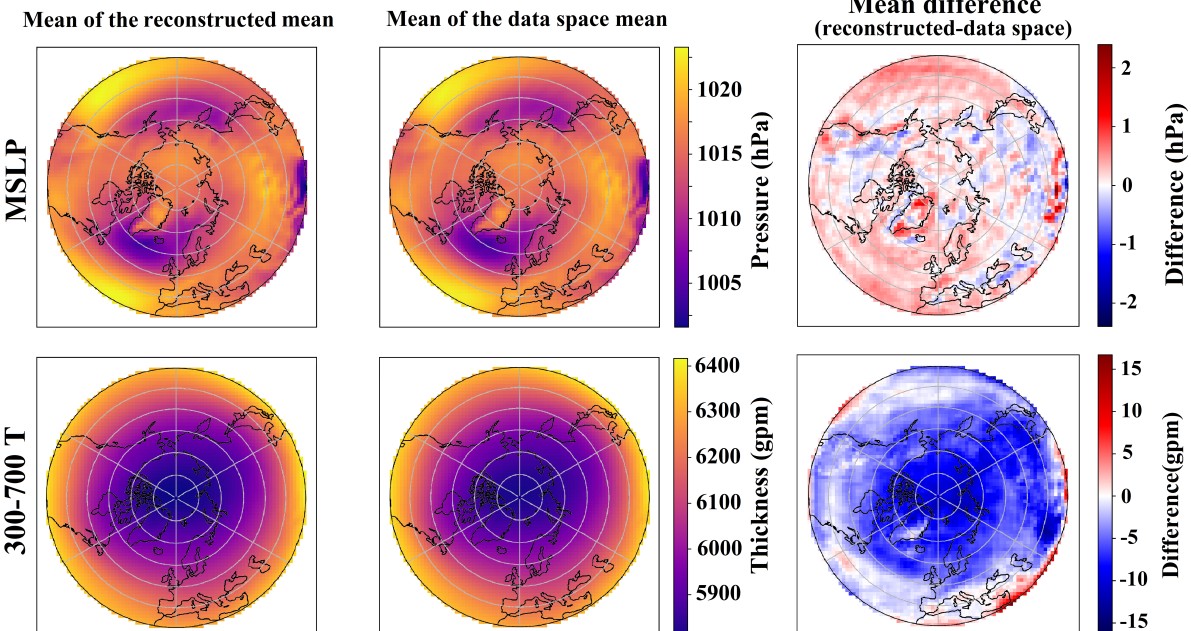

**Figure 5.** Mean fields of the MSLP (first row) and $\tau_{300-700}$ (second row), averaged over the time dimension, for both the Control and Experiment runs. The first column represents the fields reconstructed from the mean of all data points in the FAE latent space, while the second column represents the corresponding mean fields calculated directly from the data space. The difference between these two is shown in the third column.

For each data point, the associated 8-day mean value of the parameter is assigned to the data point. The seasonal anomaly of parameter $X$ for season $s$ can be calculated using Equations 2 and 3:

$$A_X^s = \sum_{i \subset 6\,classes} K_i^s(X), \tag{2}$$

$$K_i^s(X) = \frac{k_{i,s}^E(X) - k_{i,s}^C(X)}{n_s}, \tag{3}$$

Here $A_X^s$ is the seasonal anomaly of parameter $X$ for the season $s$, $k_{i,s}^E(X)$ is the sum of $X - \bar{X}_s^C$ for data points that belong to season $s$ and were categorized as cluster $i$ in the Experiment run, and $\bar{X}_s^C$ is the seasonal mean of parameter $X$ for the season $s$ in the Control run. $k_{i,s}^C(X)$ is similar to $k_{i,s}^E(X)$ but calculated for data points in the Control run, and $n_s$ is the number of data points belonging to season $s$ in the 30-year simulation period. $K_i^s(X)$ can be interpreted as the contribution of cluster $i$ to $A_X^s$. $K_i^s(X)$ has the same unit as $A_X^s$, and the sum of $K_i^s(X)$ over all clusters is equal to $A_X^s$ (see Equation 2). In Equation 3 $k_{i,s}^C(X)$ and $k_{i,s}^E(X)$ can be written as follows:

$$k_{i,s}^C(X) = N_{i,s}^C(\bar{X}_{i,s}^C - \bar{X}_s^C), \tag{4}$$

$$k_{i,s}^E(X) = N_{i,s}^E(\bar{X}_{i,s}^E - \bar{X}_s^C), \tag{5}$$

where, $N_{i,s}^C$ and $N_{i,s}^E$ are the numbers of data points for season $s$ that have been classified as cluster $i$ in the Control and Experiment runs, respectively; and $\bar{X}_{i,s}^C$ and $\bar{X}_{i,s}^E$ are the mean values of $X$ for those data points in the corresponding runs. We can express $K_i^s(X)$ in terms of two components:

$$K_i^s(X) = \frac{N_{i,s}^C(\bar{X}_{i,s}^E - \bar{X}_{i,s}^C)}{n_s} + \frac{n_{i,s}(\bar{X}_{i,s}^E - \bar{X}_s^C)}{n_s}, \tag{6}$$

where, $n_{i,s} = N_{i,s}^E - N_{i,s}^C$. Equation 6 shows that the contribution of cluster $i$ to $A_X^s$, denoted by $K_i^s(X)$, can be decomposed into two factors. The first term on the right-hand side of Equation 6 represents the contribution from the within-cluster variability, arising from the difference between the seasonal mean value of $X$ in the Control and Experiment for that cluster, weighted by $N_{i,s}^C$. This factor is named the Within-Cluster Variability Contribution (WCVC) in the following. Additionally, the 2nd term on the right-hand side of Equation 6 represents the contribution of the deviation of the cluster's seasonal mean of $X$ in the Experiment run from the seasonal mean value of $X$ the Control run, weighted by $n_{i,s}$. This factor is referred to as the Frequency-weighted Seasonal Deviation Contribution (FSDC) in the following. For each cluster, we define the normalized value of WCVC, $K_{i,s}^{WCVC}(X)$, and FSDC, $K_{i,s}^{FSDC}(X)$, as follows:

$$K_{i,s}^{WCVC}(X) = \frac{N_{i,s}^C(\bar{X}_{i,s}^E - \bar{X}_{i,s}^C)}{n_s} \Big/ \sqrt{\left(\frac{N_{i,s}^C(\bar{X}_{i,s}^E - \bar{X}_{i,s}^C)}{n_s}\right)^2 + \left(\frac{n_{i,s}(\bar{X}_{i,s}^E - \bar{X}_s^C)}{n_s}\right)^2}, \tag{7}$$

$$K_{i,s}^{FSDC}(X) = \frac{n_{i,s}(\bar{X}_{i,s}^E - \bar{X}_s^C)}{n_s} \Big/ \sqrt{\left(\frac{N_{i,s}^C(\bar{X}_{i,s}^E - \bar{X}_{i,s}^C)}{n_s}\right)^2 + \left(\frac{n_{i,s}(\bar{X}_{i,s}^E - \bar{X}_s^C)}{n_s}\right)^2}, \tag{8}$$

## 3 Results

In this section, we begin our analysis by comparing the mean states between the Control and Experiment runs. We analyze the anomalies in key climate variables, PMSET, 2m temperature, SIC, zonal mean zonal wind, and atmospheric wave propagation, attributed to the exerted forcing. We then assess the impact of negative radiative forcing over Europe on Northern Hemisphere extratropical large-scale circulation, highlighting it as the primary pathway through which the forcing influences remote regions such as the Arctic. Subsequently, in an attempt to uncover the mechanisms behind the observed anomalies in key climate variables, we examine the contributions of changes in circulation patterns to these anomalies.

## 3.1 Mean anomalies exerted by the forcing

Figure 6 presents the climatology and anomalies of tropospheric integrated PMSET, 2m temperature, and SIC. The top row illustrates the baseline climatology for the Control run, while the bottom row depicts the annual anomalies, which are derived by subtracting the mean fields of the Control run from those of the Experiment run. Specifically, the left panels detail the climatology and anomalies for the vertically integrated PMSET over the troposphere, ranging from the surface up to 200hPa. These anomalies show statistically significant increases in Northwest Canada, Alaska, and the Barents-Kara Seas. Conversely, decreases are noted in the Atlantic sector of the Arctic, Northeast Canada, and the Eastern Siberian area. It is also worth mentioning that the anomaly patterns qualitatively display an inverse correlation to the climatology of the Control run in most regions north of the 70° latitude. That is, in areas northward of 70° latitude, the regions with a positive poleward anomaly of the vertically integrated PMSET mostly have a negative climatology with values an order of magnitude larger and vice versa. The seasonal anomalies of the integrated PMSET over the troposphere are detailed in Section 3.3.1.

The lower middle and lower right panels of Figure 6 show the yearly anomalies of the 2m temperature and SIC, respectively. In the Experiment run, the 2m temperature exhibits statistically significant decreases over the North Atlantic, Greenland Sea, Laptev Sea, East Siberian Sea, Sea of Okhotsk, and the mid-latitude regions of Asia and North America. At the same time, it shows statistically significant increases over the eastern part of the Barents Sea, Kara Sea, and Bering Sea. While the 2m temperature anomaly also exhibits positive values over the northern parts of Canada and Alaska, these temperature increases are mostly not statistically significant. Qualitatively, the SIC anomaly is predominantly inversely correlated with the anomalies in 2m temperature and PMSET in regions with sea ice. That is, areas experiencing an increase in PMSET typically exhibit higher temperatures and a reduction in sea ice, and vice versa. Notably, there are statistically significant decreases in SIC over the eastern part of the Barents Sea, Kara Sea, Bering Sea, and northwest Canada. Conversely, SIC increases over the North Atlantic, Greenland Sea, Laptev Sea, East Siberian Sea, and the Sea of Okhotsk. The seasonal anomalies of the 2m temperature and SIC are presented in Section 3.3.2.

The climatology of seasonal zonal mean zonal wind, EP flux, and its divergence for the Control run, and their seasonal anomalies are presented in Figure 7. The seasonal anomalies of zonal mean zonal wind, EP flux divergence, and the vertical and meridional components of EP flux are each detailed in Section 3.3.3. These anomalies, with areas of statistical significance highlighted, are presented separately in the first row of Figures 15, 16, 17, and 18, respectively. The stratospheric variability in terms of sudden stratospheric warming (SSW) does not influence our conclusion when examining the zonal mean zonal wind anomaly, as the total number of SSW occurrences remains unchanged in the Experiment run compared to the Control run (see Table A4). Moreover, the slight shift in seasonal SSW occurrences observed in the Experiment run does not notably impact our analysis of upper atmospheric dynamics, as the associated anomalies in zonal wind patterns do not straightforwardly correlate with these changes. As planetary waves do not penetrate the stratosphere in summer, only data from autumn, winter, and spring are presented here.

In autumn, an increase in upward wave propagation from the troposphere to the stratosphere is observed northward of 50° latitude in the Experiment run, while a decrease is noticed between latitudes 30° and 50°. Accompanying this is an increase

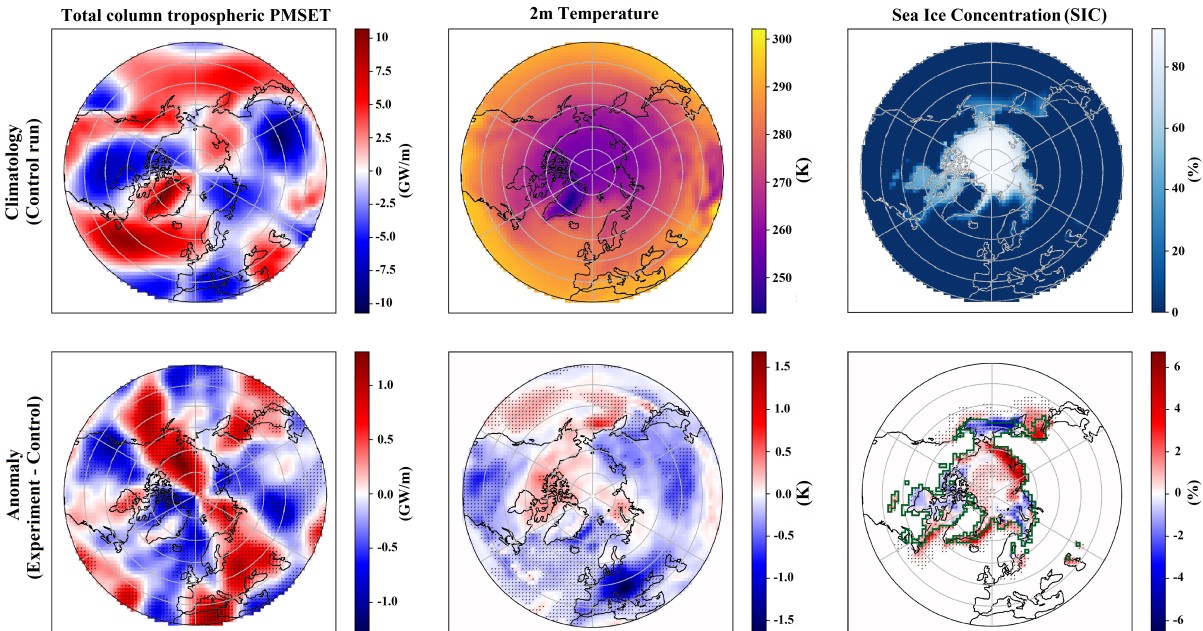

**Figure 6.** Climatology and anomalies for (from left to right) the total column tropospheric PMSET, 2m temperature, and SIC. Top row: Climatology from the Control run. Bottom row: Anomalies (Experiment - Control). Anomalies are stippled where statistically significant. The mean winter sea ice extent (SIC > 20%) from the Control run is outlined in green in the lower right panel.

in equatorward wave propagation in the stratosphere, which is also visible in the lower left panel of Figure 7. These anomalies result in a decrease in EP flux divergence in the high-latitude stratosphere and an increase in the mid- and low-latitude stratosphere. Consequently, there is an equatorward shift of the polar vortex in the lower stratosphere-upper troposphere and a poleward shift in the upper part of the stratosphere.

In winter, there is a noticeable decrease in upward wave propagation from the troposphere to the stratosphere between latitudes 40° and 55°, a trend that extends to the upper stratosphere. This decrease is accompanied by a convergence in equatorward wave propagation within the stratospheric mid-latitudes, leading to a statistically significant rise in EP flux divergence in this region. These changes, in turn, result in a poleward shift of the polar vortex across the upper troposphere and stratosphere in the Experiment run. The poleward shift of the vortex is more pronounced at higher altitudes.

In spring, a decrease in upward wave propagation from the troposphere to the stratosphere is observed between latitudes 55° and 75°, which extends to the mid-stratosphere. While the increase in upward wave propagation northward of 80° latitude is not statistically significant in the upper troposphere and lower stratosphere, there is a statistically significant increase in the high-latitude middle and upper stratosphere. There is an increase in equatorward wave propagation in regions with decreased upward wave propagation and vice versa. These anomalies result in increased EP flux divergence in the high-latitude lower stratosphere, extending to the mid-latitude stratosphere, and a decrease in the high-latitude upper stratosphere. The zonal mean

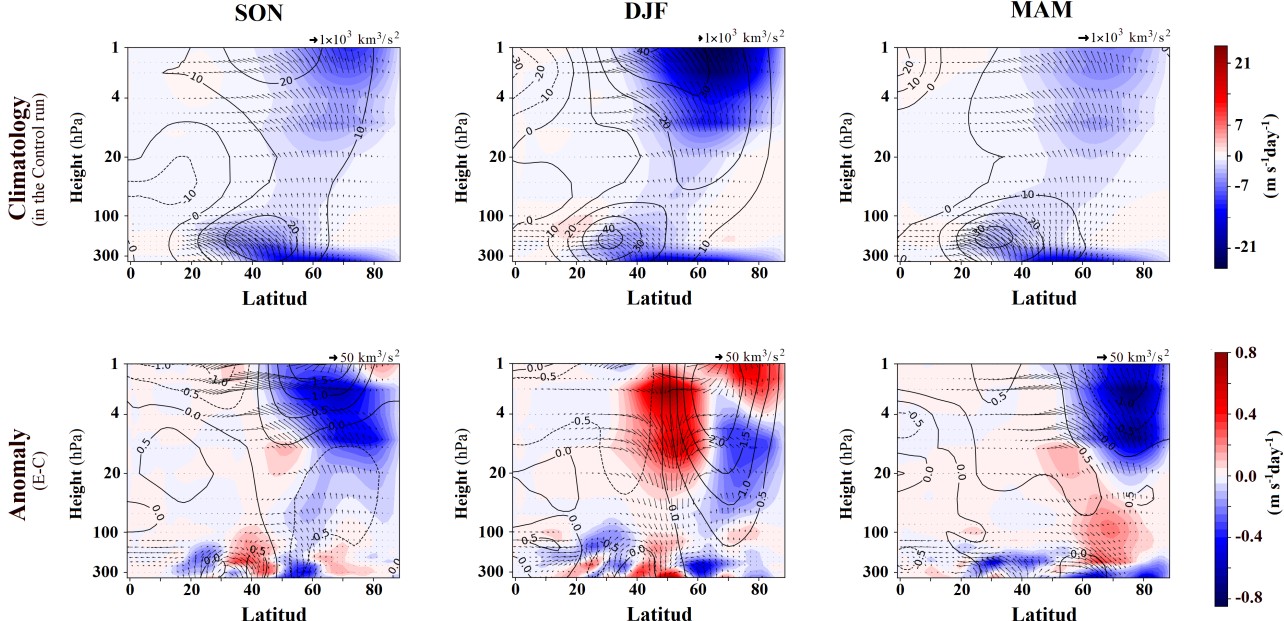

**Figure 7.** Climatology of seasonal mean EP flux (black vectors), its divergence (color shading), and zonal mean zonal wind (contour lines in $m.s^{-1}$) for the Control run (top row) as well as their anomalies in the Experimental run (bottom row), for autumn (SON, left column), winter (DJF, center column), and spring (MAM, right column). Anomalies are calculated by subtracting the Control run's seasonal mean from the Experimental run. Reference vectors for EP flux are provided in the top right of each panel; note the different scales. For zonal wind contours, solid lines indicate positive values and dashed lines negative values.

zonal wind also shows an increase in the mid-latitudes upper troposphere and stratosphere as well as high-latitude upper troposphere and lower stratosphere, along with a decrease in the high-latitude upper stratosphere.

## 3.2 Large-scale circulation clustering

We considered the large-scale circulation regime as the main pathway by which the negative radiative forcing over Europe influences remote regions such as the Arctic. Our anomaly analysis, detailed in Section 2.4, indicated that regional forcing might not induce new circulation patterns. Consequently, we examined how this forcing modifies the existing circulation patterns. To explore these modifications, it is necessary to group large-scale circulation regimes into different categories based on their similarities. Current methods for categorizing circulation patterns do not effectively capture the variables of interest associated with these patterns during the classification process. Here, we employed k-means clustering to categorize similar large-scale circulation regimes. Given our focus on the Arctic climate, we utilized the PMSET pattern associated with each circulation data point as an additional metric for similarity in our clustering analysis. The MCAE was trained to ensure that proximity in its latent space reflects similarities in both PMSET and circulation patterns, as demonstrated in Appendix D.

Accordingly, we performed k-means clustering on the latent space of MCAE to classify the data points from both the Control and Experiment runs into 6 clusters.

Figure 8 shows the average (over the time dimension) of the MSLP fields of the cluster centers in the data space. The cluster centers' MSLP fields are plotted as anomalies relative to the mean MSLP field over all data points in the data space (see Figure 5). Cluster 1 exhibits a pattern similar to a positive NAO (NAO+), characterized by a high-pressure anomaly in Siberia and a low-pressure anomaly over the Aleutian Islands (C1). Cluster 2 displays the Atlantic and Pacific oceans' high-pressure systems accompanied by continental low-pressure anomalies (C2). Cluster 3 features a high-pressure blocking system over Scandinavia and Siberia, countered by a low anomaly over Greenland, the Arctic Ocean, Alaska, and north Canada (C3). Cluster 4 presents a positive NAO-like pattern, underscored by a widespread east-west low-pressure anomaly and a high-pressure anomaly over Alaska and north Canada (C4). Cluster 5 embodies a negative NAO-like pattern (C5). Finally, Cluster 6 portrays a transcontinental high, with an Atlantic high extending to Scandinavia (C6). Although C1 and C4 resemble the positive phase of the NAO, they lack the clear dipole pattern typically associated with NAO+. Notably, these clusters feature a positive MSLP anomaly in southern Europe and northern Africa, which differs from the classical NAO+ distribution.

The monthly occurrence frequencies of different clusters in the Control and Experiment runs are shown in Figure 9. In a given simulation, the occurrence frequency of a cluster for a specific month corresponds to the proportion of data points assigned to that cluster within that month over the entire simulation period. The error bars at the top of the bar charts in Figure 9 indicate the standard deviation, calculated across 1,000 bootstrap ensembles generated via resampling with replacement for each simulation. The monthly occurrence frequencies of different clusters exhibit discernible seasonality. For instance, C2 and C6 predominantly occur during warmer months, whereas the remaining clusters are more frequently observed during cooler periods. The occurrence frequencies of the clusters in the Experiment run, which is influenced by negative regional forcing over Europe, exhibit variations when compared to the Control run. During the boreal autumn season (September to November), C3 shows a substantial decrease in occurrence frequency in the Experiment run compared to the Control run, while C4 displays a marked increase in occurrence frequency (see Table A2). In winter (December to February), C4 demonstrates an increased occurrence frequency in the Experiment run, while C1 exhibits a notable decline. In spring (March to May), the discrepancies in occurrence frequencies seem less pronounced, with C6 presenting the highest increase in the Experiment run and C2 experiencing the most substantial decrease. During summer (June to August), the occurrence frequency of C2 has the greatest increase in the Experiment run, and C6 displays the largest drop in occurrence frequency. It is worth noting that C3 shows decreases in occurrence frequencies across all seasons in the Experiment run, while C4 displays increases in occurrence frequencies in each season. The seasonal changes in the occurrence frequencies of different clusters are presented in Table A2.

The seasonal centers of the clusters for both Control and Experiment runs are depicted in Figure 10. These centers are plotted as MSLP anomalies relative to the seasonal mean fields in the Control run. The primary characteristics of the clusters' centers, as shown in Figure 8, are also observed in the seasonal cluster centers, with certain exceptions. Notably, during the summer periods for clusters C4 and C5, the low occurrence frequency of these clusters (see Figure 9) led to the cluster center not being entirely consistent with other seasons. Moreover, the seasonal cluster centers exhibit notable similarities between the

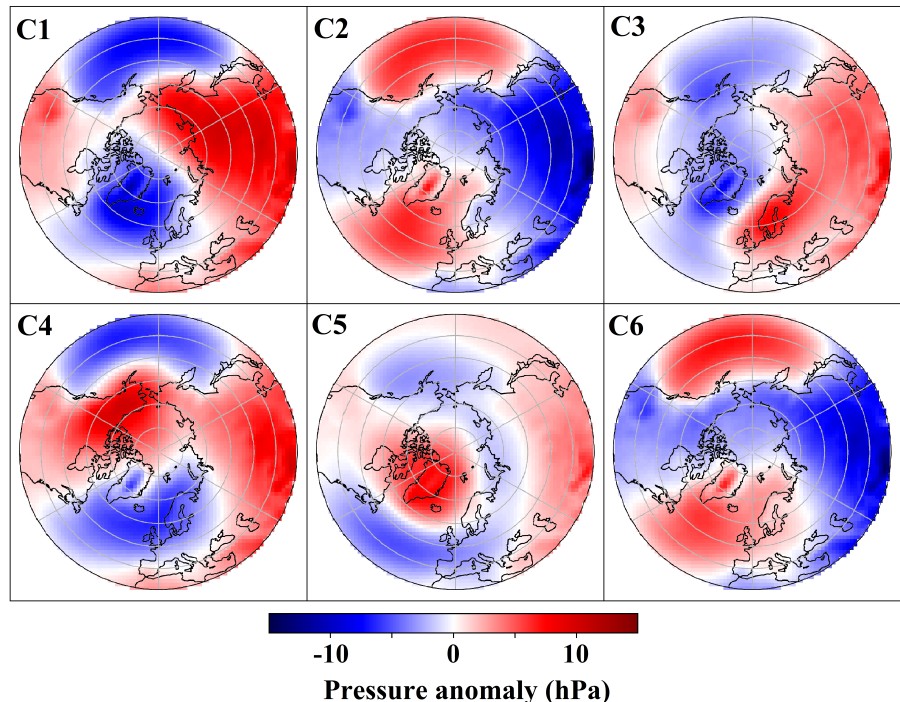

**Figure 8.** Mean MSLP fields for clusters derived from k-means clustering (6 clusters) performed on the latent space of MCAE. The fields for each cluster are plotted as anomalies relative to the mean MSLP field for all data points, encompassing both the Control and Experiment runs.

Control and Experiment runs, albeit with slight variations. The seasonal discrepancies between the class centers in Control and Experiment runs are presented in Figure E1. More discussion on the cluster seasonal discrepancies can be found in Appendix E.

The k-means clustering results on different representations of data points are shown and discussed in Appendix D. K-means clustering algorithms group data points based on their proximity in the selected representation space. The differing outcomes from clustering across these representation spaces stem from how these spaces prioritize proximity among data points. Specifically, we applied k-means clustering to the latent space of the MCAE, taking into account similarities in both PMSET and circulation patterns for grouping the data points. While some clusters resemble recognized circulation patterns, it is crucial to consider the diverse conceptual definitions of circulation that shape these outcomes.

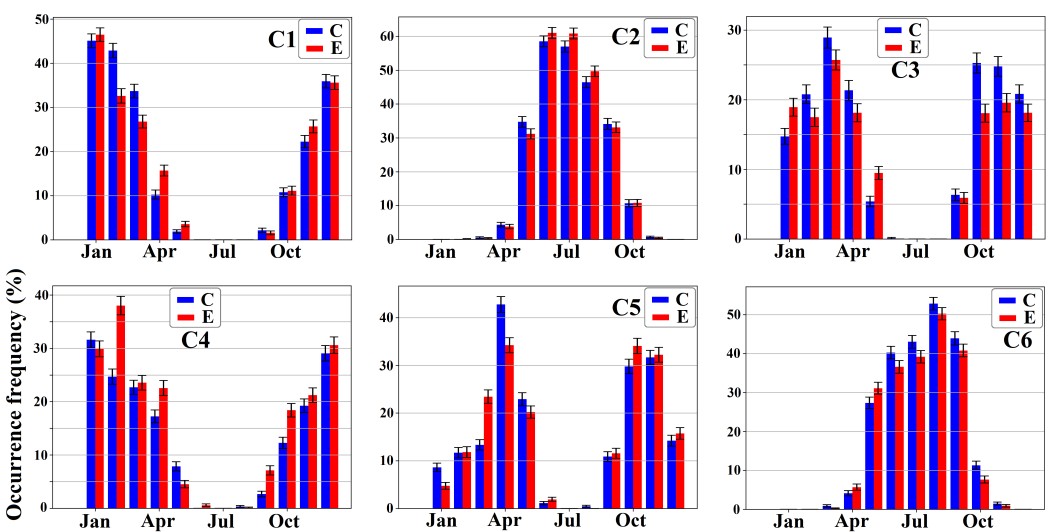

**Figure 9.** The monthly occurrence frequency of the 6 clusters shown in Figure 8 for the Control (blue) and Experiment (red) runs. The error bars at the top of the bar charts represent the standard deviation calculated via bootstrapping, which involved resampling with replacement to create 1,000 ensembles for each simulation.

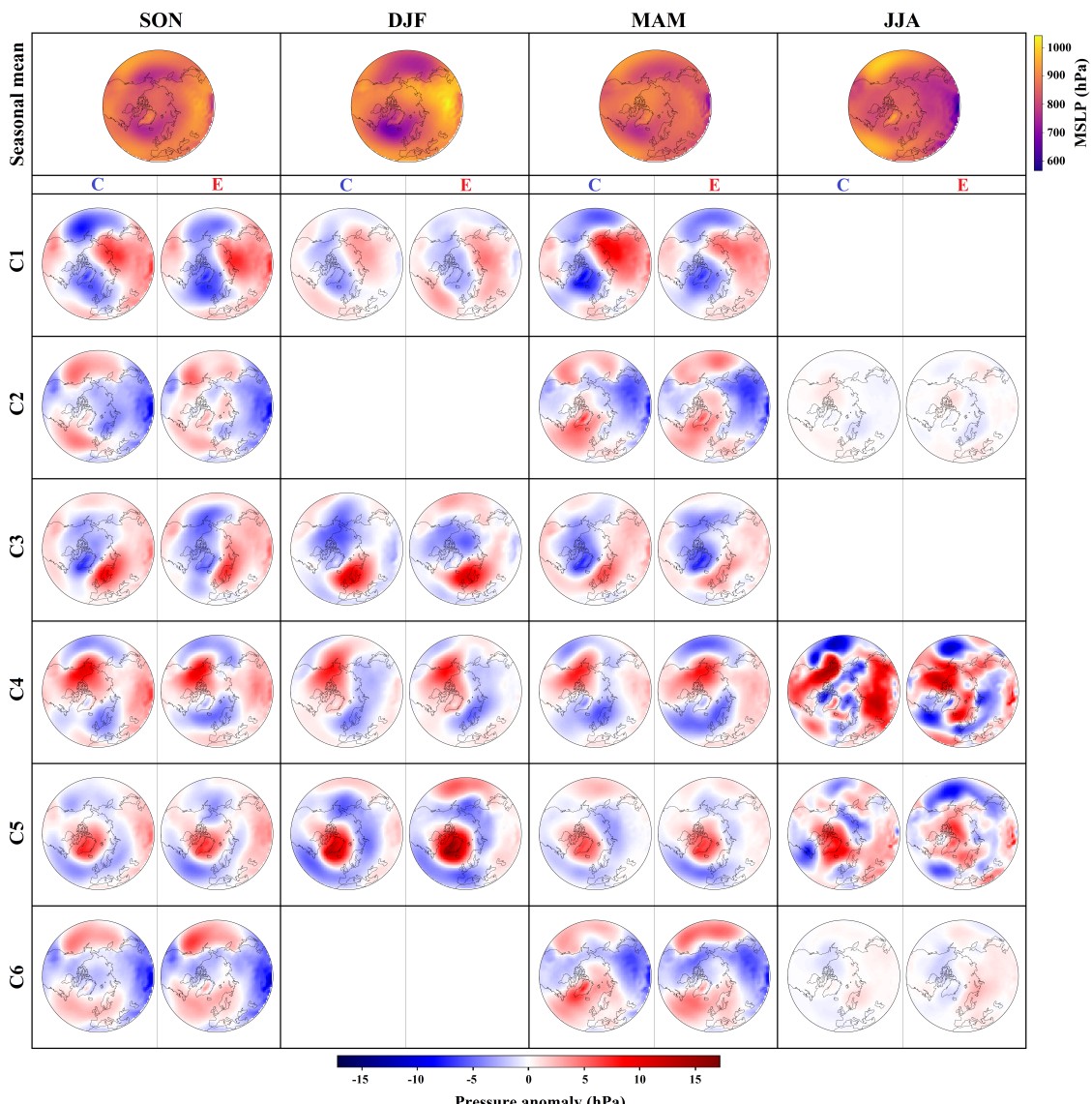

**Figure 10.** First row: The seasonal mean MSLP field calculated for the Control run for each season: autumn (SON, first column), winter (DJF, second column), spring (MAM, third column), and summer (JJA, fourth column), with the corresponding color bar on the right-hand side of the row. Rows 2-7: Clusters' centers for Control (C) and Experiment (E) runs for clusters 1 to 6, respectively. The clusters were derived from performing the k-means clustering on the latent space of MCAE. In each season's column, the left panel (C) portrays the mean field of MSLP for the Control run, while the right panel (E) represents equivalent fields for the Experiment run. Cluster centers are obtained by subtracting the first row's Control run seasonal mean MSLP from the respective seasonal mean field for each cluster and season in the corresponding simulation (E or C). The color bar for class centers is located at the figure's bottom, and empty spaces within a specific season (columns) and cluster (rows) indicate that the cluster has no representative for that season.

### 3.3 Class contribution to the observed anomalies

In this section, we analyze the extratropical seasonal anomalies of PMSET, 2m temperature, SIC, zonal mean zonal wind, and atmospheric wave propagation. To understand the dynamic interactions of each circulation pattern that lead to the observed anomaly, we examine how changes in circulation patterns, induced by regional negative radiative forcing over Europe in the Experiment run, contribute to the observed anomalies in key climate variables. We further investigate how changes in the seasonal occurrence frequency (FSDC) and seasonal discrepancies (WCVC) of the preferred circulation patterns form the

class contributions to the anomalies. A summary of the main contributions of different clusters to the seasonal anomalies is presented in Table A3.

#### 3.3.1 PMSET

Figure 11 shows the seasonal anomalies of the mean PMSET at 66° N latitude across the lower, upper, and total troposphere as well as the class contributions to these anomalies. We considered the latitude of 66° N as the Arctic circle. In most cases,

the WCVC is the dominant component of the class contribution to the anomaly. A notable negative correlation exists between the contributions of classes to the anomalies in the lower and upper troposphere. However, exceptions include the WCVC component for C1 in winter and the WCVC components for C3 and C6 in summer. This correlation is further affirmed by a statistically significant Pearson correlation coefficient of -0.88 between the daily mean PMSET values of the upper and lower troposphere at 66° N latitude for both Control and Experiment runs.

In autumn, the upper troposphere of the Experiment run significantly increases in the mean PMSET, primarily driven by C2, where WCVC dominates (see Figure 11). The reduced occurrence frequency of C3 in autumn, represented by the FSDC component, leads to a negative contribution to the total tropospheric mean PMSET, primarily attributed to the FSDC's impact on the lower troposphere, not counterbalanced in the upper troposphere. In winter, only the total troposphere shows a statistically significant mean PMSET anomaly, while both the lower and upper components do not. This arises from the enhanced mean

PMSET in the lower troposphere which is not compensated by the energy transport in the upper troposphere. C1, steered by WCVC, is the main contributor to the total tropospheric anomaly, showcasing positive contributions in both tropospheric layers. Meanwhile, C5's substantial contributions in both the lower and upper troposphere cancel each other out, leading to a small net effect on the total troposphere. In Spring, during the Experiment run, the mean PMSET decreases in the lower and total troposphere but increases in the upper troposphere. All these changes are statistically significant. C3 and C5, dominated

by WCVC, are the main contributors to the mean PMSET anomaly in the lower and upper troposphere, yet their impact on the total troposphere remains subdued. Moreover, C1 and C4, both WCVC-driven, stand out as key contributors to the total tropospheric mean PMSET anomaly. In summer, the mean PMSET anomalies show positive values for the lower, upper, and total troposphere, with only the total troposphere anomaly statistically significant. The main contributor to the total tropospheric mean PMSET anomaly is C6, dominated by the WCVC component.

The seasonal anomalies of the vertically integrated PMSET are shown in the first row of Figure 12. Subsequent rows in the figure delineate the contributions from various classes to these seasonal anomalies. For each seasonal column in Figure 12, the

sum of plots from rows 2 through 7 corresponds to the seasonal anomaly depicted in the first row (see Section 2.6). We present only the class contributions to the total tropospheric PMSET, as the contributions to both the lower and upper troposphere are notably similar to it (not shown here). The arrows on the class contribution plots show the relative roles of WCVC and FSDC to the class contribution. The horizontal component of the arrows corresponds to the normalized value of the WCVC and points rightward for positive values, while the vertical component represents the normalized value of the FSDC and points upward for positive values.

In autumn, C5 and C4 predominantly contribute to the increase in the vertically integrated PMSET over Alaska. The WCVC component is primarily dominant in the class contributions of these clusters in this region, except in C4's western part of the Bering Sea, where FSDC also positively contributes to the class contribution. C5, C4, and C3 contribute to the decrease over East Siberia, mainly driven by the WCVC component. C3 and C4 are pivotal in the increase PMSET over the Barents-Kara Seas region. While WCVC primarily dictates the contributions of these clusters, FSDC has a notable role in the Barents-Kara Seas area. The increase in the occurrence frequency of C4 (see Table A2) and the decreased occurrence frequency of C3 positively affect the vertically integrated PMSET over the Barents-Kara Seas region. C3 plays a significant role in the PMSET reduction over the North Atlantic and Greenland, with WCVC and FSDC contributing fairly equally. It also produces high PMSET in west Greenland and northeast Canada, primarily driven by the WCVC component. These effects are offset by the contributions of clusters C1 and C4, dominated mainly by WCVC.

In winter, clusters C4 and C1 primarily drive the increase in the vertically integrated PMSET over the Arctic Pacific sector and the corresponding decrease over the East Siberian region, with the WCVC component predominantly responsible. These clusters, along with C3, also contribute to the decline in vertically integrated PMSET over the Atlantic sector of the Arctic and to the increase over the Barents-Kara Seas. In these regions, the class contributions are mainly dictated by the WCVC component, although in C4, the FSDC component also contributes to the anomaly. The increased occurrence frequency of C4 contributes to the decrease in vertically integrated PMSET in the Atlantic sector of the Arctic and the increase in the Barents-Kara Seas.

In spring, C4 and C5 contribute the most to the decrease in the vertically integrated PMSET over the Chukchi Sea, with WCVC being the dominant component. C3 and C5 are the primary factors leading to the decreased vertically integrated PMSET over the Atlantic sector of the Arctic, predominantly driven by WCVC.

During summer, only clusters C2 and C6 contribute to the anomaly. Specifically, C2 primarily drives the anomaly in the vertically integrated PMSET over the Pacific and East Siberian regions, especially contributing to the increase over East Siberia. In this area, the class contribution of this cluster is dominated by the WCVC component. Conversely, C6 exerts a more marked influence on the anomaly patterns over the Atlantic and Barents-Kara Seas region, particularly contributing to the increase over the Atlantic sector of the Arctic. The WCVC component dominated the class contribution of this cluster in this area.

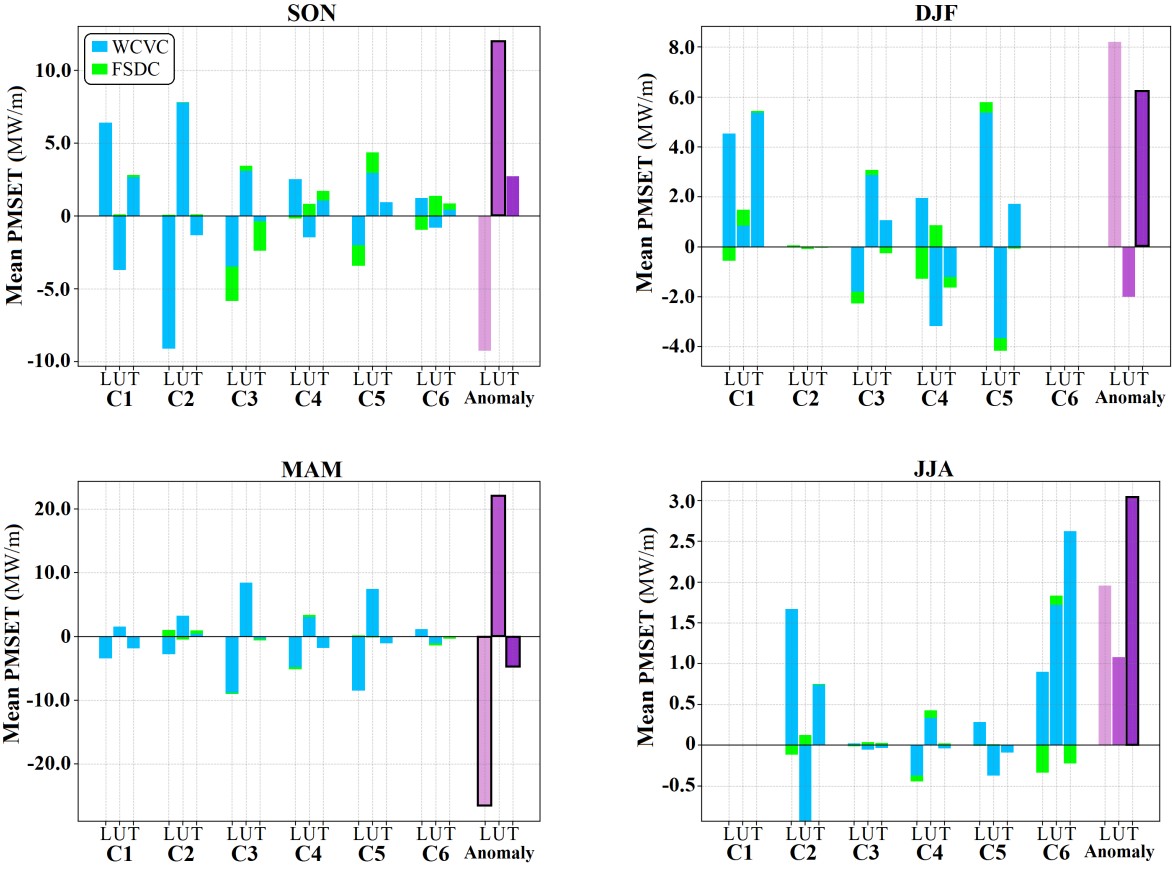

**Figure 11.** Grouped bar charts representing the class contributions (C1-C6) to the seasonal mean PMSET passing through latitude 66° N in the lower (L), upper (U), and total (T) troposphere. In each grouped bar chart representing the class contributions, the two components of WCVC and FSDC are highlighted by blue and green colors, respectively. The additional grouped bar charts on the right-hand side of each seasonal panel represent the seasonal anomaly of mean PMSET for L (light purple), U (medium purple), and T (dark purple). The anomaly bars with black outlines indicate statistical significance. The bars depict both positive (poleward transport anomaly) and negative (equatorward transport anomaly) values with the 0-axis marked by a horizontal line.

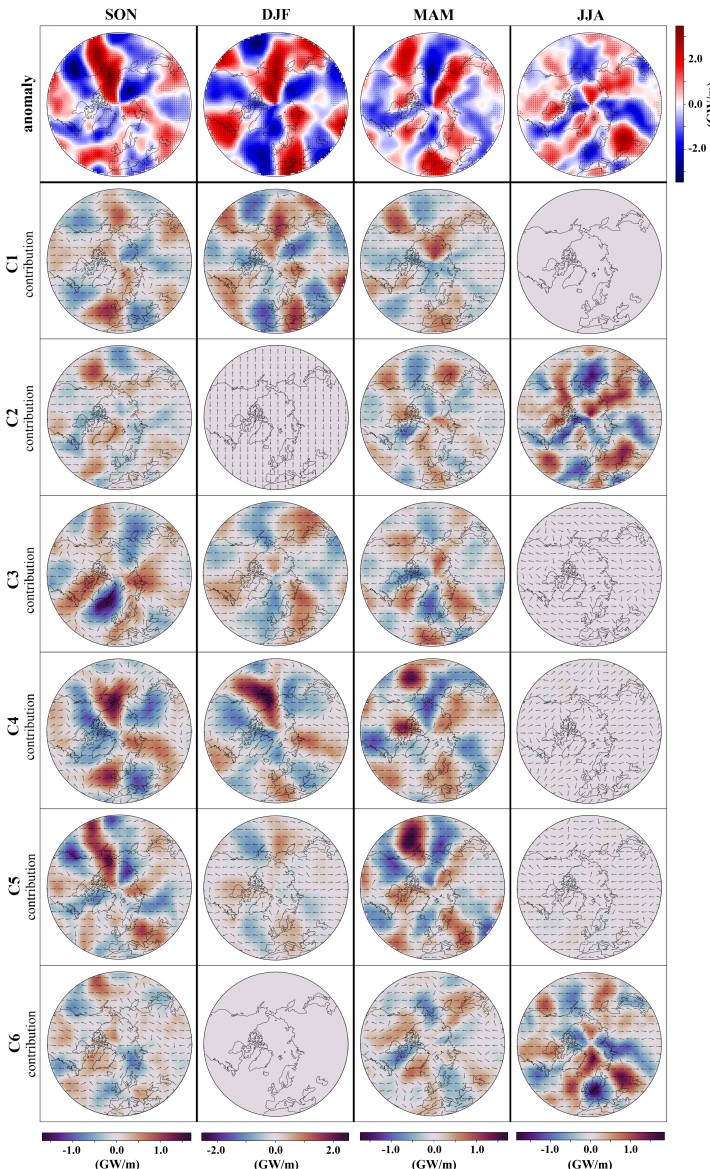

**Figure 12.** The seasonal anomaly of the vertically integrated PMSET over the troposphere (first row) and the respective contributions from different classes (rows 2-7) for each season: autumn (SON, first column), winter (DJF, second column), spring (MAM, third column), and summer (JJA, fourth column). The anomalies are determined by subtracting the seasonal mean vertically integrated PMSET field of the Control run from that of the Experiment run, with statistically significant areas stippled. Each subsequent row from 2 to 7 represents the contribution of classes C1 through C6 to the seasonal anomalies, respectively. The orientation of the arrows on the class contribution plots elucidates the relative roles of the Within-Cluster Variability Contribution (WCVC) and Frequency-weighted Seasonal Deviation Contribution (FSDC) to the class contribution. The horizontal component of the arrows corresponds to the normalized value of the WCVC and points rightward for positive values, while the vertical component represents the normalized value of the FSDC and points upward for positive values. The color bar for the anomaly field is located to the right of the first row, and color bars for seasonal class contributions are located at the bottom of each seasonal column.

### 3.3.2  2m temperature and SIC

The seasonal anomalies of the 2m temperature and SIC are shown in the first rows of Figure 13 and 14, respectively. Their subsequent rows illustrate the contributions of different classes to these seasonal anomalies. In autumn, C3 predominantly contributes to the decreases in SIC over the Barents-Kara Seas, with the WCVC component being dominant. Additionally, C3 plays a major role in the temperature increase over northeast Canada, with both WCVC and FSDC contributing to this increase. C3 and C4 are the main contributors to the 2m temperature increase over the Barents-Kara Seas, with WCVC being the dominating factor in their contributions, although in C4, the FSDC component also contributes to the anomaly. The increased occurrence frequency of C4 acts as a mitigating factor in this region. In general, the FSDC component of C4's class contribution to the 2m temperature anomaly is negative throughout the Arctic, and its contribution to the SIC anomaly is positive. Conversely, for C3, the FSDC component is mostly positive for the 2m temperature anomaly and negative for the SIC anomaly within the Arctic, except in the Atlantic sector and the Barents-Kara Seas regions. C4 and C5 substantially contribute to the increased 2-meter temperature over northwest Canada and Alaska while contributing to colder conditions in North America's mid-latitudes. In northwest Canada and Alaska, WCVC dominates their contributions to the temperature increase, whereas the FSDC component of these clusters notably contributes to the temperature decrease in North America's mid-latitudes. In C4, the FSDC component even dominates the class contribution in this region. Meanwhile, all clusters contribute to the increased SIC and decrease in the 2-meter temperature over the Laptev and East Siberian Sea.

In winter, Clusters C2 and C6 do not contribute to the anomaly as they primarily represent summer conditions. C1 largely mirrors the winter anomaly pattern for 2m temperature. Specifically, it plays a major role in increasing the 2m temperature over the Arctic Ocean and the North Pole, accounting for cooler mid-latitude Asian and warmer North Pole conditions, and contributing the most to the decreased SIC over the Bering Sea, with the WCVC dominating all these contributions. Furthermore, C1 and C4 are the primary contributors to the increased 2m temperature over Western Canada and Alaska and the decreased 2m temperature in mid-latitude North America. For C1, the class contribution in these regions is dominated by the WCVC component, while for C4, the FSDC also plays a role. C4 is also the main contributor to the SIC decrease over the Sea of Okhotsk, with the WCVC domination. Nearly every cluster contributes to the 2m temperature increase over the Barents-Kara Seas and the associated reduction in SIC, each with its distinct pattern. These clusters also contribute to the 2m temperature decrease over the North Atlantic and Greenland Sea and the consequent increase in SIC in these regions.

In spring, the WCVC component primarily shapes the class contributions, except for the contributions of C2 and C6, summer clusters, to the 2m temperature anomaly. In these specific instances, the FSDC component dominates the class contributions. Even though the contributions of these clusters are minor, it is worth mentioning that the decreased frequency of C2 leads to a negative temperature contribution in every region, while the increased frequency of C6 exhibits a positive effect in every region. C3 and C5 contribute the most to the increase in SIC over the North Atlantic, Greenland Sea, and Barents-Kara Seas in spring. They also drive most of the 2m temperature decrease over these same regions. In summer, while both C2 and C6 contribute with similar patterns to either SIC and 2m temperature, it is C2 that appears to contribute the most to the increased

SIC over the Laptev and East Siberian Seas, as well as the reduced SIC over the Barents-Kara Seas, with WCVC being the dominant part.

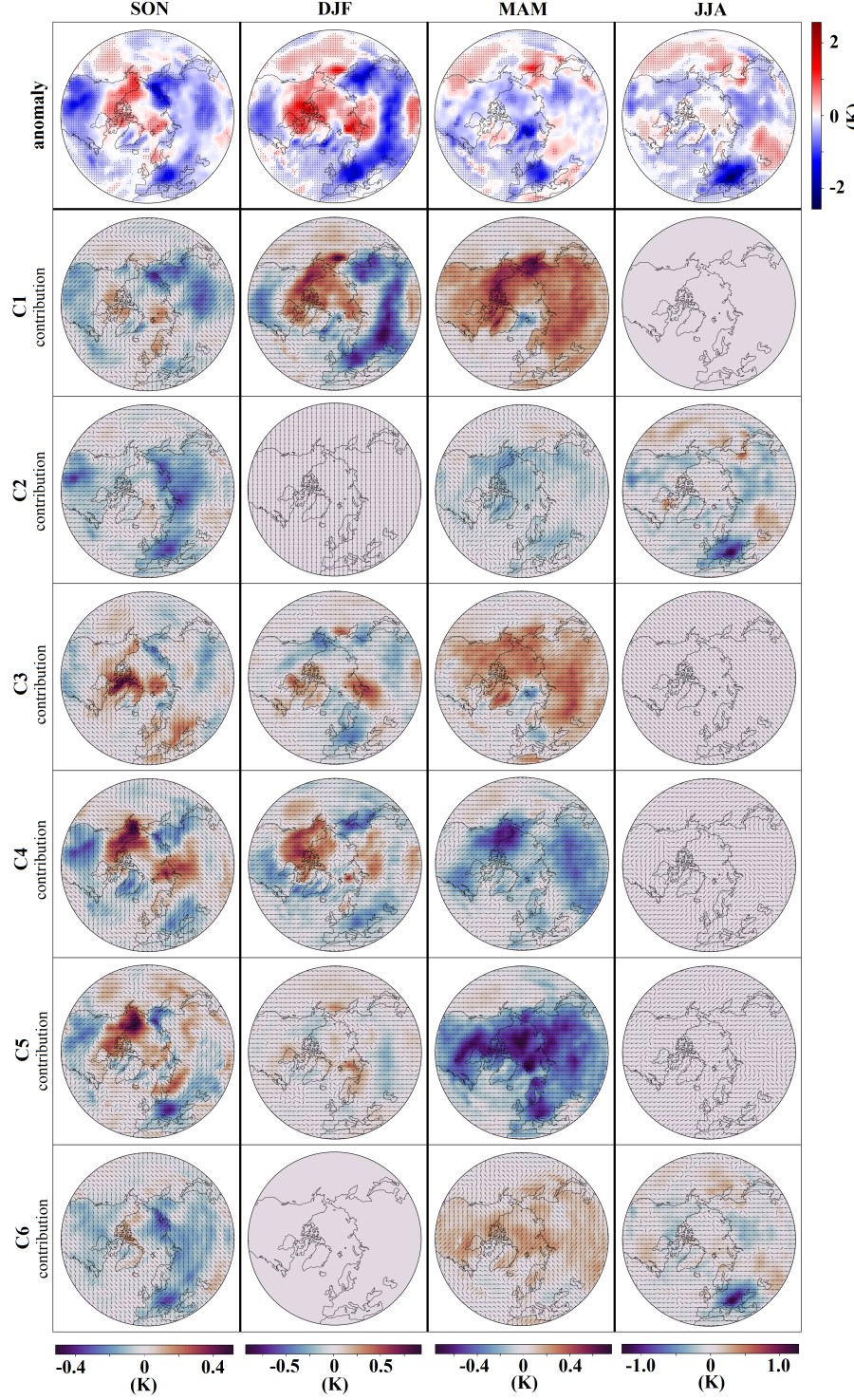

**Figure 13.** Analogous to Figure 12 but calculated for 2m temperature.

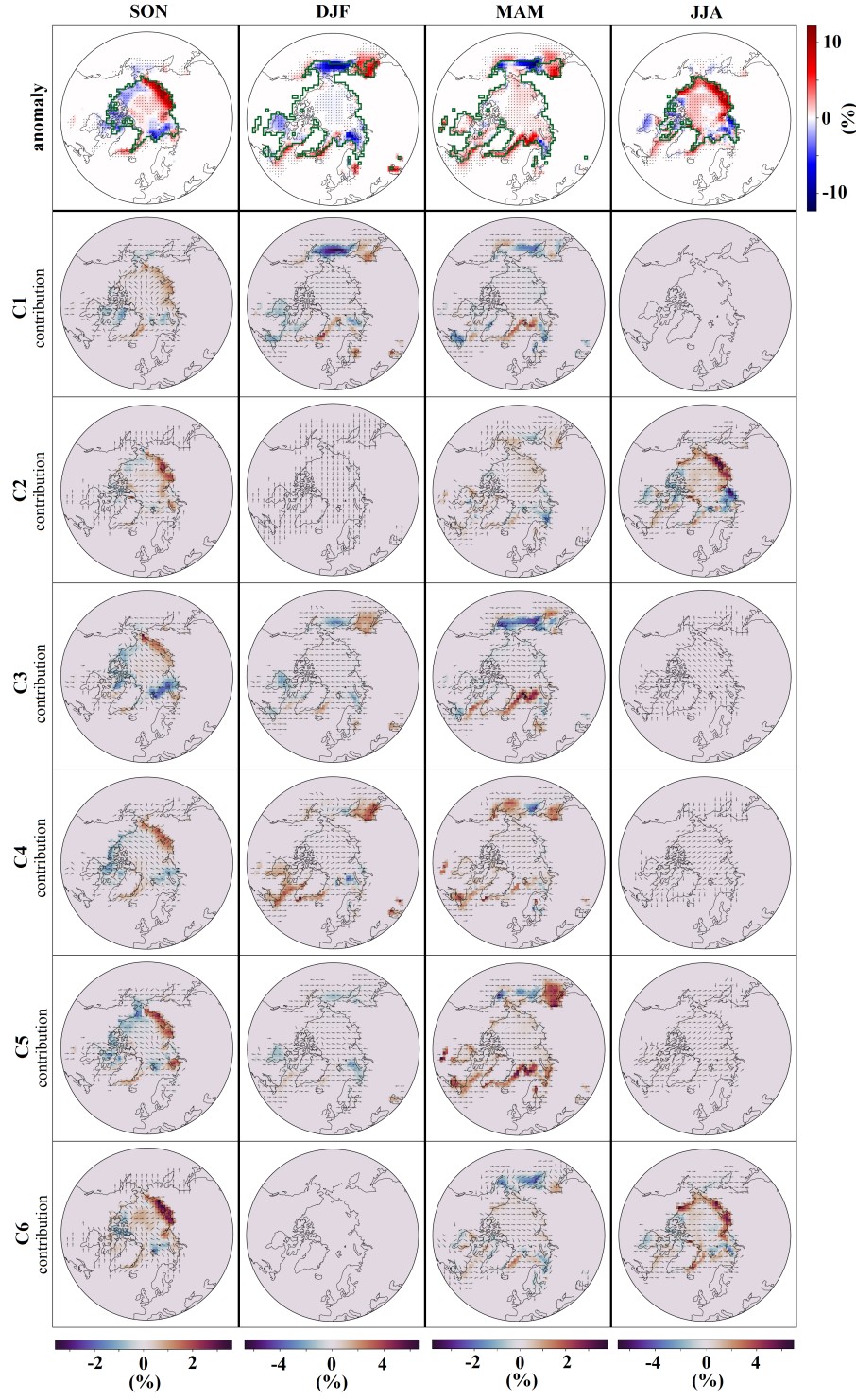

**Figure 14.** Analogous to Figure 12 but computed for SIC. The green lines in the seasonal anomaly plots in the first row delineate the mean seasonal sea ice extent calculated for the Control run.

### 3.3.3 Stratospheric dynamics and EP Flux

The seasonal anomalies of zonal mean zonal wind, EP flux divergence, EP flux vertical and meridional components are presented in the first row of Figures 15, 16, 17, and 18, respectively. It is important to note that the EP flux divergences do not always align perfectly with the pattern of zonal mean zonal wind anomalies. This discrepancy can arise because zonal winds are influenced by factors beyond just EP flux divergences. EP flux divergences represent the momentum deposited by resolved waves within the model. However, changes in zonal winds can also be affected by non-resolved parameterized gravity waves. Changes in the zonal mean zonal wind due to the resolved waves may affect the parameterized momentum depositions by the gravity waves. This, in turn, could affect the prevailing zonal flow and induce changes in the residual circulation below the breaking level. Such dynamics might explain the anomalies of the zonal mean zonal wind far away from the regions of significant EP fluxes (Chandran et al., 2013; Cohen et al., 2013; Limpasuvan et al., 2016).

In autumn, certain clusters, notably C4, C3, and to a lesser extent, C1, impact wave propagation and EP flux divergence in the high-latitude upper stratosphere. They primarily contribute to increased upward and equatorward wave propagation, thereby reducing the EP flux divergence in this region, with both the WCVC and FSDC components playing a role. The WCVC component predominates in C1's contributions. The FSDC component of C3's contributions to vertical and equatorward wave propagation exhibits a negative sign in most parts of the upper troposphere and stratosphere, especially in latitudes northward of 30°, which means that a reduced occurrence frequency of this cluster in autumn diminishes the upward and equatorward wave propagations in these regions. Conversely, the FSDC component of C4's contributions to the vertical and equatorward wave propagation shows a positive sign in similar areas, indicating increased upward and equatorward wave propagations due to the increased occurrence frequency of this cluster in autumn. The reduced upward wave propagation between 30° and 50° latitude, extending to the upper stratosphere, is mainly due to C5 and C4, with WCVC dominating. Moreover, all clusters contribute to the zonal mean zonal wind decrease in the high-latitude upper troposphere and lower stratosphere. The most notable contributors to the poleward shift of the vortex in the upper stratosphere are C1 and C5, along with C3. In C3, the combination of FSDC and WCVC reproduces the vortex's poleward shift in this region despite the negative FSDC component. Only C5 increases the EP flux divergence in the upper stratosphere, attributed to the decreased equatorward and upward wave propagation within this cluster. The WCVC part mostly dominates the class contributions of C5.

The influence of the WCVC component is paramount in winter, governing the class contributions associated with wave propagation and polar vortex dynamics. In winter, C4 and, to a lesser extent, C3 primarily contribute to the reduction in upward wave propagation between latitudes 40° and 55° in the upper troposphere and stratosphere during winter. C4 is also the main driver of the decrease in equatorward wave propagation within the same sector, with C3 just showing a weak contribution. As a result, both clusters contribute to the increase in EP flux divergence in this region, with C4 causing a more concentrated increase in EP flux divergence between latitudes 40° and 55° in the upper stratosphere. C3 and C1 contribute to the poleward shift and strengthening of the polar vortex, while C4 does not significantly contribute to these changes in the polar vortex. C5 is the main contributor to the polar vortex's poleward shift and strengthening. The poleward shift of the polar vortex, attributed to C5, diminishes from the stratosphere to the troposphere. In addition, C5 and C3 are the main contributors to the

reduced zonal mean zonal wind in the upper troposphere between latitudes 20° and 40° and from 300 to 100hPa. C5 notably counteracts the reduction in upward wave propagation in the mid-latitude stratosphere by inducing a substantial increase in upward wave propagation within this region. Furthermore, C5 influences equatorward wave propagation, causing an increase in the mid-latitude upper stratosphere. Consequently, C5 mitigates the increase in EP flux divergence in the mid-latitude upper stratosphere.

In spring, except for C2 and C6 class contributions, which have minor effects on the anomaly fields, the WCVC component dominates the class contributions. C5 and C4 are the primary contributors to the increase in upward wave propagation in the high-latitude stratosphere. Both clusters significantly contribute to the amplified equatorward wave propagation in the upper stratosphere, with C5's contribution being particularly substantial. As such, they become the main drivers behind the decrease in EP flux divergence in the high-latitude upper stratosphere. Unexpectedly, these two clusters are the main contributors to the zonal mean zonal wind increase across nearly the entire stratosphere. Moreover, C3 and C1 primarily contribute to the decrease in upward wave propagation, extending from the high-latitude upper troposphere to the mid-latitude stratosphere. C3 also contributes to the increased wave propagation in the latitudes northward of 85° in the upper troposphere and stratosphere, leading to a decrease in EP flux divergence in the high-latitude upper stratosphere. In terms of influencing zonal mean zonal wind, C3 and C1 are the main drivers of the decrease in the high-latitude upper stratosphere. More broadly, C3 leads to a decrease in the zonal mean zonal wind throughout the stratosphere, while C1 specifically decreases the zonal mean zonal wind in the high-latitude stratosphere.

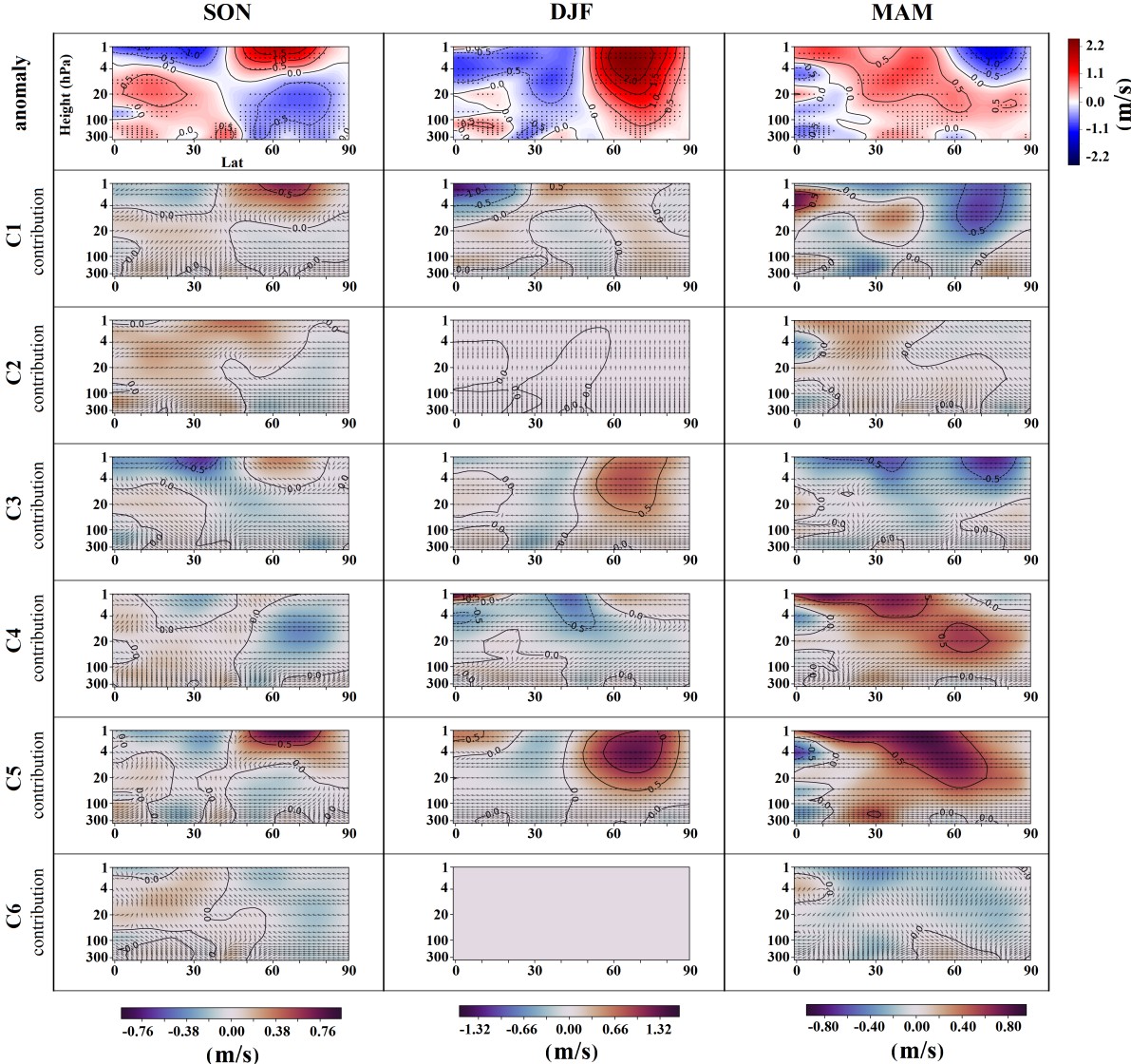

**Figure 15.** Zonal mean zonal wind anomaly (first row) depicted as latitude-height graphs for the northern hemisphere, spanning from 300 to 1hPa in height for each season: autumn (SON, first column), winter (DJF, second column), and spring (MAM, third column). Anomalies are calculated by subtracting the Control run's seasonal mean zonal mean zonal wind field from the Experiment run's equivalent, with statistically significant areas hashed. Each subsequent row from 2 to 7 represents the contribution of classes C1 through C6 to the seasonal anomalies, respectively. The orientation of the arrows on the class contribution plots elucidates the relative roles of WCVC and FSDC to the class contribution. The horizontal component of the arrows corresponds to the normalized value of the WCVC and points rightward for positive values, while the vertical component represents the normalized value of the FSDC and points upward for positive values. The color bar for the anomaly field is located to the right of the first row, and color bars for seasonal class contributions are located at the bottom of each seasonal column. The contours in the plots delineate the zonal mean zonal wind isolines, with a difference of $0.5\,m\,s^{-1}$ between each. Dashed contours represent negative values.

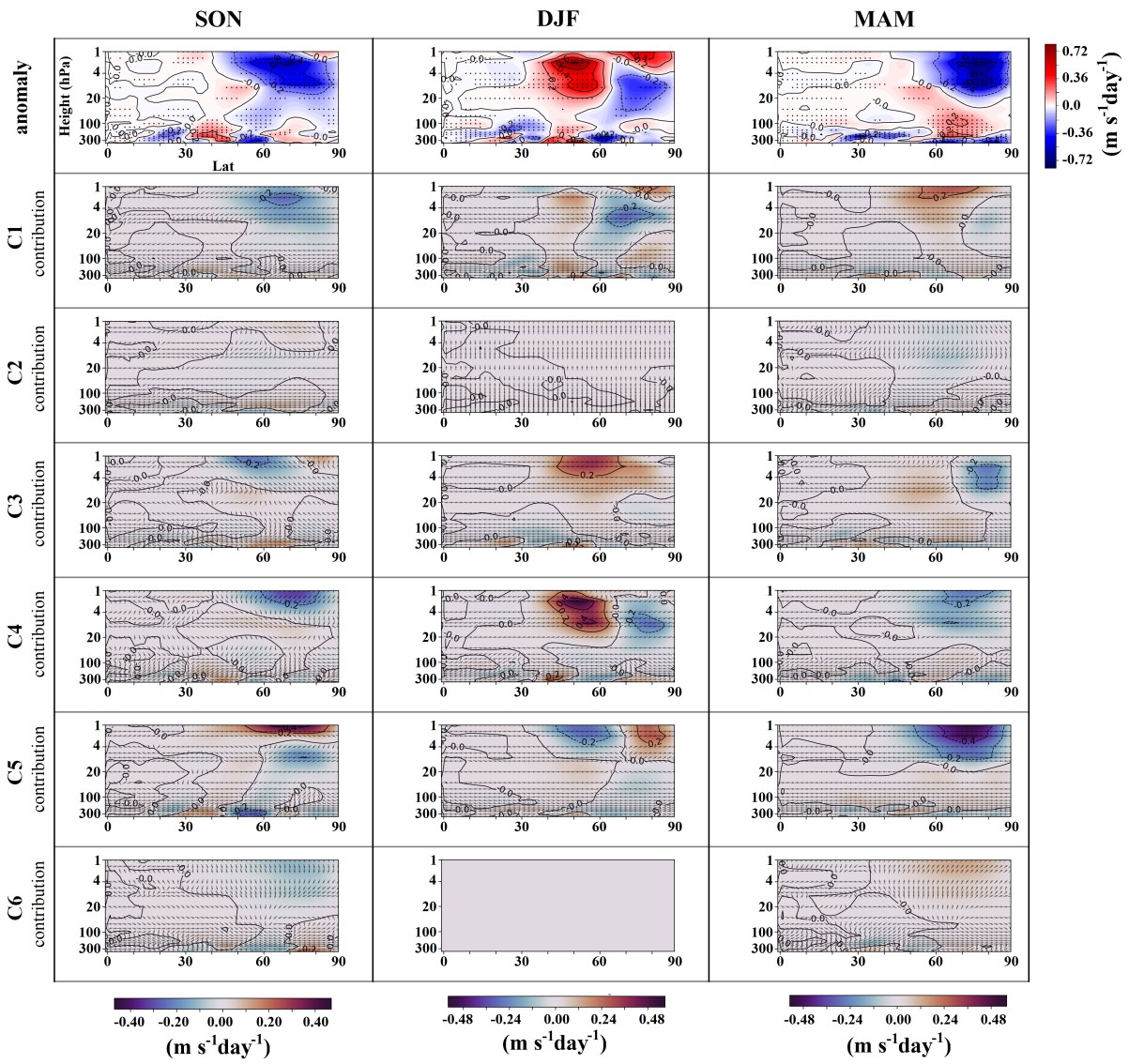

**Figure 16.** Analogous to Figure 15 but computed for EP flux divergence. The contours in the plots delineate EP flux divergence isolines, with a difference of $0.2\,m\,s^{-1}\,day^{-1}$ between each. Dashed contours represent negative values.

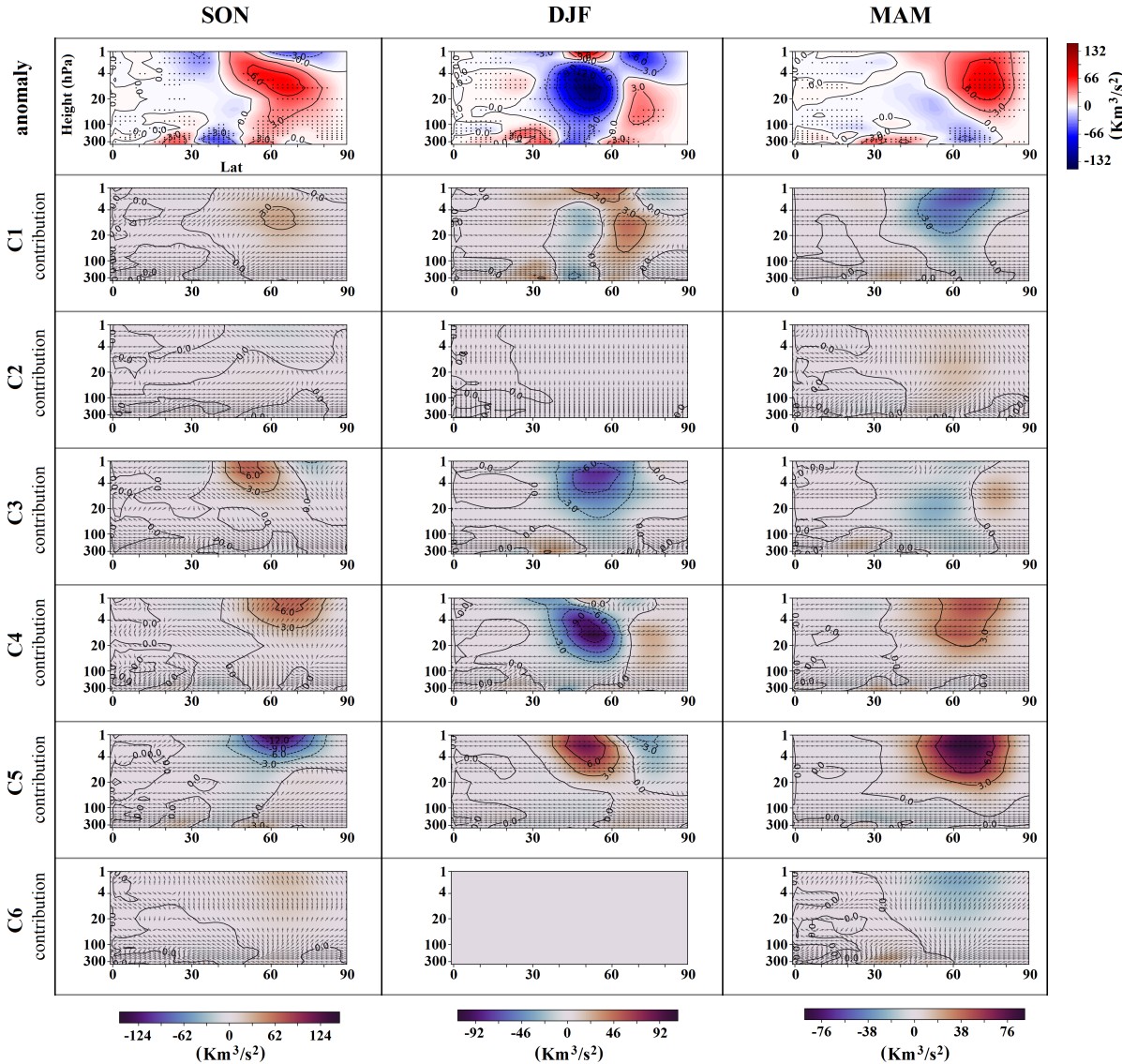

**Figure 17.** Analogous to Figure 15 but computed for the EP flux vertical component. Upward EP flux is assigned positive values. The contours in the plots delineate isolines of the vertical EP flux, with a difference of $3 \times 10^4 \, m^3 \, s^{-2}$ between each. Dashed contours denote negative values.

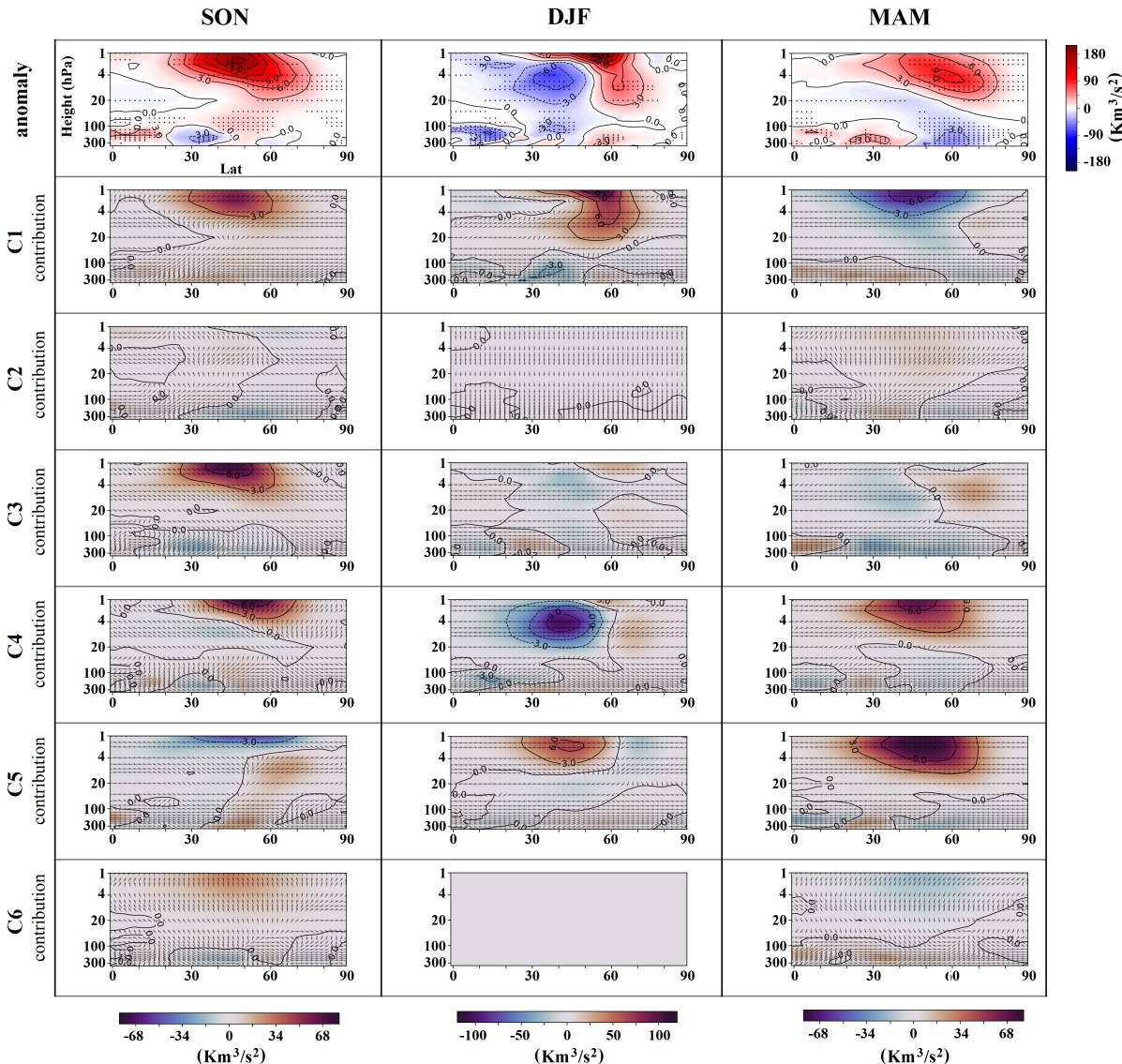

**Figure 18.** Analogous to Figure 15 but computed for the EP flux meridional component. Equatorward EP flux anomaly is assigned positive values. The contours in the plots delineate isolines of the vertical EP flux, with a difference of $3 \times 10^4 \, m^3 \, s^{-2}$ between each. Dashed contours denote negative values.

## 4    Discussion

The CNN-based deep auto-encoders demonstrated promising capabilities in predicting circulation patterns when given the additional thickness field between two pressure levels (Weyn et al., 2019, 2020). This highlights their proficiency in memorizing the spatiotemporal patterns present in such a dataset. We employed the CNN-based auto-encoders as feature extraction

algorithms on the dataset consisting of MSLP and thickness for the 700 to 300 hPa layers. We investigated the different representations of our dataset characterizing the circulation pattern in the Control and Experiment runs. Given that the input data encompasses a temporal dimension of 8 days, the features extracted within the latent space of the auto-encoders used in this study incorporate the variability of the circulation patterns over about one week (Tran et al., 2015). Our findings showed that the FAE effectively created a latent space representation of our dataset. Notably, the FAE could accurately reconstruct the means of data points in the original space, using the mean of data in the latent space representation. Additionally, the clustering results demonstrated the capability of FAE's latent space to discern subtle variations among data points in transitional seasons. This allowed us to distinctly segregate data points related to autumn from those of spring - a level of granularity not achievable with clustering on the conventional data space representations, such as k-means and SOM on data pace representation, and k-means applied to an EOF reduced space, where autumn and spring data points often intermingled. Thus, the FAE's robust spatiotemporal feature extraction performance suggests it is a compelling method for generating a concise yet informative representation of our dataset.

External forcings can modify circulation patterns in complex and nonlinear ways (Gillett and Fyfe, 2013; Hannachi et al., 2017) by introducing new spatiotemporal patterns, changing the preferred circulation patterns, altering their frequencies, or by a combination of them. We employed the AAE, which uses only the Control run's data points for training and shares the same architecture as FAE, to detect anomalous large-scale circulation patterns in the Experiment run. The accuracy with which the AAE regenerated data points from the Experiment run, comparable to those from the Control run, suggests that the imposed negative radiative forcing over Europe did not give rise to new spatiotemporal patterns in large-scale circulation. Moreover, training an auto-encoder exclusively on Experiment run data points demonstrates the interchangeability of spatiotemporal patterns between the Control and Experiment runs. However, employing the novel MCAE approach and performing k-means clustering on its latent space, we revealed a robust change in the seasonal occurrence frequency of certain circulation patterns in response to the imposed negative radiative forcing over Europe in MPI-ESM1.2.

In our study, we leveraged an auto-encoder with a partly predefined latent space, which offered us the flexibility to tailor the latent space in line with our specific research objectives. This approach enabled us to group data points based on both the similarities in spatiotemporal patterns and the alignment with specific features we targeted. Using the MCAE, incorporating the PMSET indices into the first 40 dimensions of the 100-dimensional latent space, enabled us to cluster data points based on similarities in both associated PMSET indices and spatiotemporal patterns. The strong influence of PMSET indices on clustering was expected as they form a large portion of the latent space. This assertion is further strengthened by comparing the clustering outcomes derived from the data points' representation containing only the PMSET indices and those from the MCAE's latent space representation. Both show partially similar features, verifying the crucial role of PMSET indices in our clustering process. Additionally, the manifestation of a seasonal cycle, the most dominant pattern within our dataset, in MCAE's latent space clustering results substantiates the similarities in spatiotemporal patterns within clusters. This also indicates that the 60 free dimensions in MCAE's latent space carry information regarding the main spatiotemporal patterns in the data set.

We employed k-means clustering on the MCAE's latent space to group data points into six clusters. The class centers of C1 and C4 resemble the positive phase of the NAO, with C4 also mirroring the positive phase of the Pacific–North American

(PNA). C3 represents Scandinavian blocking, C5 corresponds to the negative phase of the Arctic Oscillation (AO), and C2 and C6 denote patterns typical of warmer months. These clustering outcomes, which consider PMSET indices' similarities, align closely with findings from previous research (Crasemann et al., 2017; Hannachi et al., 2017; Hochman et al., 2021; Lembo et al., 2022). In these studies, circulation patterns are often classified using reanalysis data and focused on specific regions of the Northern Hemisphere through different methodologies. Our results confirm well-established patterns like NAO, AO, PNA, and Scandinavian blocking, also consistently identified in the previous research.

Increased PMSET can lead to higher Arctic temperatures, which in turn could result in decreased SIC (Woods and Caballero, 2016; Graversen and Burtu, 2016). Our analysis showed a notable correlation among the PMSET, 2m temperature, and SIC total anomalies. Specifically, a positive PMSET anomaly is associated with an increase in 2m temperature and a corresponding decrease in SIC concentration across most Arctic regions. This pattern holds true for seasonal anomalies in all but the summer months, underscoring the significance of PMSET as a key factor in determining the 2m temperature and SIC concentration in the Arctic during colder periods. The absence of this correlation in summer highlights the potential influence of other processes in controlling the 2m temperature and SIC in the Arctic during that season. Moreover, the observed negative correlation between the mean PMSET in the lower and upper troposphere at latitude 66°N may be intricately linked to the dynamics of the Ferrel and Polar cell interactions, particularly around the polar front. Specifically, the lifting of warm, moist air at the warm front of a cyclone could enhance PMSET in the lower troposphere, while the descending cold, dry air at the trailing cold front might counteract or even reverse PMSET in the upper troposphere (Wallace and Hobbs, 2006). Any class contribution that does not exhibit this negative correlation may indicate the absence of this specific mechanism.

Heterogeneous aerosol forcing over Europe has been recognized to impact Arctic SIC (Fyfe et al., 2013; Acosta Navarro et al., 2016; Gagné et al., 2017). Our findings suggest that regional radiative forcings, like those induced by aerosols, can alter Arctic SIC through modifications in circulation patterns and the associated PMSET. Moreover, although some phenomena observed in individual clusters do not appear in the seasonal anomaly, they can still have significant impacts. For example, C3 is associated with an increase in PMSET over Northeast Canada in autumn. This increase is mitigated by decreases in PMSET from C4 and C5, rendering it less noticeable in the seasonal anomaly. However, unlike its effects on PMSET, the increase caused by C3 coincides with a decline in SIC over Northeast Canada in autumn, a change that C4 and C5 do not mitigate. Thus, the reduction in SIC associated with C3 is evident in the seasonal anomaly, whereas the corresponding increase in the PMSET is mitigated by C4 and C5, resulting in no discernible effect in the seasonal PMSET anomaly. This highlights the ability of our methodology to capture the complex, nonlinear interactions within the climate system that may not be readily apparent when analyzing broader seasonal anomalies.

As previously discussed, the class contributions for each cluster are composed of WCVC and FSDC. WCVC arises from the seasonal discrepancies between the class centers in the Control and Experiment runs, weighted by the cluster occurrence frequency in the Control run. FSDC reflects the deviation of the cluster's seasonal mean in the Experiment run from the seasonal mean value in the Control run weighted by changes in occurrence frequency between the two runs. This pattern is evident in our results, specifically in C3's contribution to the PMSET anomaly during autumn. C3's impact on vertically integrated PMSET in the troposphere varies by region. For instance, it leads to decreased PMSET over the North Atlantic and increased PMSET

over East Canada. In the North Atlantic, both WCVC and FSDC contribute to the decrease. The reduced occurrence frequency of this cluster in autumn, which consists of the high-pressure anomaly over Scandinavia and low anomaly over Greenland,

caused the FSDC part to show negative values over the North Atlantic region. The FSDC does not contribute to the increase in East Canada as the class center does not show any pressure pattern associated with energy transport in this region. WCVC, however, does contribute to PMSET changes in both regions, guided by higher pressure over Greenland and lower pressure in North America and the North Atlantic in the Experiment run.

C1, partly mirroring the positive phase of NAO, is characterized by the Siberian high and Aleutian low. In the Experiment run

during winter, C1's class centers exhibit higher pressure anomalies over Eurasia, particularly Siberia. This cluster is the main contributor to the observed warm Arctic and cold mid-latitudes anomaly in winter, driven mainly by the WCVC component. Thus, the elevated pressure across Eurasia in the Experiment run is the primary driver of this anomaly. This aligns with Galytska et al. (2023), indicating the direct relationship between Arctic near-surface temperature and Siberian high, a key element of the warm Arctic-cold mid-latitudes pattern. While the C3 and C5 class centers also display elevated pressure over Eurasia in the

Experiment run, they don't replicate the distinct warm Arctic and cold mid-latitude pattern in their near-surface temperature anomaly contributions, despite WCVC's dominance. This underscores C1's pivotal role in shaping this anomaly and highlights the climate's nonlinear response to seemingly analogous triggers (Hannachi et al., 2017).

Given the proposed role of delayed sea ice refreezing in the Barents-Kara Seaas in initiating the stratospheric pathway during autumn (Jaiser et al., 2016; Nakamura et al., 2016; Dethloff et al., 2019), we explored the possible interplay between SIC in the

670 Barents-Kara Seas and stratospheric dynamics in this season. In our analysis, we observed a pronounced SIC reduction in the Barents-Kara Seas near the sea ice edge during autumn in the Experiment run. This phenomenon is concurrently aligned with diminished EP flux divergence in the mid- and high-latitude stratosphere, as well as a reduction in the zonal mean zonal wind in the high-latitude upper troposphere and lower stratosphere. These findings, derived from analyzing the seasonal anomaly fields, support the proposed mechanism linking SIC changes in the Barents-Kara Seas with alterations in stratospheric dynamics

(Jaiser et al., 2016; Xu et al., 2023). Delving into the specific mechanisms, C3 emerged as the main contributor to the SIC reduction in the Barents-Kara Seas, with the WCVC component steering this anomaly. The WCVC component of the class contribution to the EP flux divergence decreases in the mid- and high-latitude stratosphere and is the prevailing factor in the mid-latitude upper stratosphere. It also dominates the class contribution to the zonal mean zonal wind in the high-latitude lower stratosphere, showing a decrease in this region. In contrast, the FSDC component, manifesting the effect of the cluster's robust

decreased frequency in this season, acts mainly as a moderating factor against the WCVC component in these stratospheric regions, though its impact is relatively subdued. Hence, while the reduced occurrence frequency of the cluster resembling the Scandinavian blocking pattern diminishes vertical wave propagation and intensifies EP flux divergence in mid- and high-latitude stratospheres (Peings, 2019; Blackport and Screen, 2021), it plays a minor role compared to the other controlling factors, namely the seasonal discrepancies between the Control and Experiment runs. In summary, the WCVC component of

C3 class contributions predominantly influenced the SIC reduction in the Barents-Kara Seas, as well as the EP flux divergence and zonal mean zonal wind in the upper atmosphere. Although the FSDC component also impacted the upper atmosphere mainly as a mitigating factor for the WCVC component, its role in modulating EP flux divergence in the aforementioned

regions was less pronounced. These findings indicate that the reduction in SIC in the Barents-Kara Sea, attributed to the WCVC component of the C3, coincides with the reduced zonal mean zonal wind in the high-latitude upper troposphere and

690 lower stratosphere. This alignment is consistent with mechanisms suggested for autumn in other studies (Jaiser et al., 2016; Xu et al., 2023). However, compared to other clusters, the WCVC of C3 is not the dominant influential factor; other classes also contribute significantly to the observed seasonal anomalies in these atmospheric dynamics. The broader contribution from other clusters may explain the intermittency of this mechanism (Siew et al., 2020). Thus, the observed anomalies, even when they can be attributed to a certain class, are the results of the complex interplay of the WCVC and FSDC components. In particular,

even minor seasonal discrepancies between Control and Experiment runs can cause a pronounced class contribution, especially when a cluster predominates in that season.

Although specific parts of the anomaly can be traced back to a certain cluster, as seen with the reduced SIC in the Barents-Kara Seas in autumn, anomalies typically result from the interplay of multiple clusters. For instance, the EP flux divergence anomaly and the associated zonal mean zonal wind anomaly in autumn emerge from combined contributions of various clusters,

each influencing distinct regions. Notably, in colder seasons, C4 and C5 significantly contribute to the anomaly fields of both EP flux divergence and zonal mean zonal wind. These observations highlight the multifaceted interactions among the various clusters.

Moreover, certain anomalies cannot be attributed to specific clusters, as they appear in all clusters' contributions. An example of this is the increased SIC in the Laptev and East Siberian Seas in autumn. There are potential explanations for these kinds

of unattributable anomalies. One possibility is that the underlying process responsible for the anomaly is present in all the clusters, making it difficult to identify a main contributor. Another explanation could be that the targeted anomaly persists longer than the underlying mechanism that caused it. For instance, a cluster might cause reduced SIC in one region during a season, and this effect could persist across the season, being observed in all other class contributions. Our class contribution calculations only account for changes that coincide with circulation patterns without considering any lag correlations beyond

an 8-day time dimension inherent in the data points. Given the proposed lag correlation between certain circulation patterns and other parameters (Galytska et al., 2023), this could be a further explanation for the anomalies that cannot be easily traced to specific clusters.

Several studies have identified atmospheric circulation patterns as key drivers of various Arctic phenomena and their connections to mid-latitudes. In particular, these patterns have been suggested to influence Arctic SIC, temperature patterns across

regions, and stratospheric pathways linking the Arctic and mid-latitudes (Mewes and Jacobi, 2019; Peings, 2019; Blackport et al., 2019; Siew et al., 2020; Blackport and Screen, 2021). In our methodology, we inherently assume causality from the circulation pattern to other parameters, such as the Arctic 2m temperature. This reflects our underlying assumption that circulation patterns, influenced by regional forcing over Europe, primarily impact the Arctic climate by altering the PMSET (Cohen et al., 2018). Specifically, we perform clustering based on PMSET, and then we calculate the class contribution based on these

clusters, thereby suggesting a causal link from the circulation patterns assigned to similar PMSET indices to the underlying parameters. However, this should be approached with caution. While our approach successfully hinted at this causality in many cases, the calculated class contributions are not definitive evidence of a causal relationship, especially in the absence

of physical justification. It is crucial to recognize that our class contribution calculation merely illustrates changes that align with circulation patterns rather than establishing direct causality. This nuance can have implications for interpretation. For example, with supporting evidence, one might interpret the class contribution with inverse causality, i.e., from the parameter to the circulation pattern. A case in point is the stronger stratospheric polar vortex observed in the class contributions of C5, resembling the negative phase of NAO, during winter and spring.

In winter and spring, the C5 class contributions to the seasonal anomalies of EP flux, EP flux divergence, and the zonal mean zonal wind are dominated by the WCVC component. This indicates that these contributions largely arise from the seasonal differences between the class centers in the Control and Experiment runs. The complex relationship between the class contribution of C5 calculated for EP flux, EP flux divergence, and the zonal mean zonal wind renders it challenging to derive a clear causal link. Therefore, the coinciding stronger polar vortex with C5 might be interpreted in two ways: 1) C5's seasonal discrepancies between the class centers in the Control and Experiment runs leading to a stronger vortex; 2) A stronger vortex causing the change in C5. By underlining this complex relationship, we emphasize the importance of careful interpretation of class contributions and the potential for multifaceted causality, depending on the specific context and available supporting evidence.

## 5    Conclusions

DL assists humans in tackling many complex tasks through its problem-solving ability and flexibility to adapt to new, intricate challenges. The ability of DL models to discern intricate patterns from data offers substantial promise for climate applications, though this is still an emerging field. DL models offer unprecedented flexibility, enabling targeted focus on specific data aspects deemed crucial. In this study, we utilized DL-based analysis to examine the complex response of the climate system to local radiative forcing over Europe. The flexibility of DL models not only facilitates our analysis of anomalies in response to the forcing but also enables us to categorize circulation patterns based on a critical target variable. Focusing on the effects of circulation on the Arctic climate, we selected PMSET as this target variable for our clustering approach, employing MCAE model. This strategy allows us to concentrate on our parameter of interest—the PMSET patterns—while accounting for the inherent structure of the data points in grouping the circulation patterns. This level of flexibility, unattainable with conventional methods, enables our MCAE method to produce a representation that captures both the inherent data structure and metadata nuances, facilitating meaningful clustering.

We developed a formulation for class contributions that, while independent of DL models, significantly benefits from the results of DL-based clustering. This formulation enables us to discern the effects of changes in various circulation patterns on anomalies within different climatic parameters. Unlike traditional linear analyses, our approach captures the nonlinear behavior of the climate system, demonstrating how different circulation clusters distinctly react to the same forcing. Our study revealed that negative radiative forcing over Europe, resembling the heterogeneous radiative forcing exerted by aerosols, influences the climate system by altering existing circulation patterns and their frequencies without introducing new patterns. The clusters change between the Experiment and Control runs in two main ways: changes in occurrence frequency and seasonal

discrepancies between the class centers in the Experiment and Control runs. While pronounced changes in seasonal occurrence frequency can significantly contribute to the observed seasonal anomaly, even subtle alterations in the seasonal differences between class centers can profoundly affect the class's contribution to the anomaly, especially if that cluster frequently occurs.

Specifically, changes in the circulation pattern with the high-pressure system over Scandinavia emerged as the main determinant of the reduced SIC in the Barents-Kara Seas in autumn through manipulating the PMSET in this region. The seasonal discrepancy between the Control and Experiment runs for this cluster is the main contributor to the reduced SIC in this region. This discrepancy also makes this cluster one of the main contributors to the reduced EP flux divergence in the mid-latitude upper stratosphere and the reduced zonal mean zonal wind in the high-latitude upper troposphere and lower stratosphere. The reduced occurrence of this pattern in autumn represents one of the most notable shifts in circulation pattern frequencies. However, it does not contribute to the autumnal SIC reduction in the Barents-Kara Seas. It is a counteractive factor to the aforementioned reduction in EP flux divergence and zonal mean zonal wind, playing a minor role in the dynamics of the upper atmosphere. Nevertheless, other circulation patterns, particularly those resembling the negative and positive phases of the NAO, appear to play a more pronounced role in shaping the anomalies in the upper troposphere and lower stratosphere. In conclusion, although the proposed mechanism that links SIC reduction in the Barents-Kara Seas with disturbances in the stratospheric polar vortex is observed, arising from the discrepancy of the circulation patterns featuring a high-pressure system over Scandinavia in the Experiment run, its impact on the dynamics of the upper troposphere and lower stratosphere remains relatively minor compared to other clusters. This limited influence likely accounts for the observed intermittency of the proposed mechanism (Siew et al., 2020).

These findings illustrate the intricate interplay between various circulation patterns and climatic parameters, revealing the underlying processes that drive observed anomalies in different seasons. Notably, a complex interplay of different circulation patterns, particularly those resembling different phases of NAO, determines wave propagation and stratospheric dynamics. Although we did not specifically investigate the stratospheric pathway in this study, we demonstrate that regional negative radiative forcing over Europe can alter both Arctic climatic parameters and stratospheric dynamics.

Our methodology facilitated multifactor analysis by combining DL with human expertise to uncover the mechanisms explaining the observed phenomena. Given the flexibility of our approach, it stands as a valuable tool that can aid in understanding the physical mechanisms behind certain proposed patterns. Specifically, this methodology excels in analyzing diverse climate datasets by adapting its framework to focus on any chosen target variable, thereby enabling inferring and understanding of the nonlinear patterns across different climatic factors. Such capabilities lead to deeper and more accurate insights into climate behavior.

*Code and data availability.* The datasets from the Control and Experiment simulations, along with the Deep Learning (DL) algorithms and processing code examples utilized in this study, are publicly accessible at https://zenodo.org/communities/arctic_dl. This repository includes all relevant materials required to replicate the analyses and results presented in this manuscript.

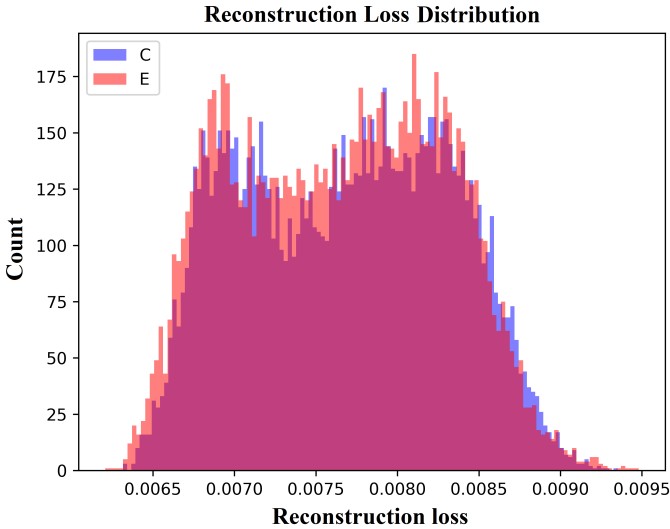

**Figure A1.** Reconstruction loss distribution for the Control (C) and Experiment (E) runs generated by the AAEE.

## Appendix A: Analyzing pattern interchangeability in the Control and Experiment runs

To investigate the interchangeability of patterns captured in both the Control and Experiment runs, we also trained an auto-encoder with a similar architecture as AAE (see Table A1). This auto-encoder, however, was trained only on data points from the Experiment run, employing the same training strategy utilized for the AAE. We named this AAEE (Anomaly detecting Auto-Encoder trained on Experiment run only). The reconstruction loss of AAEE, calculated by inputting data points from both the Experiment and Control runs, is illustrated in Figure A1. Similar to AAE, AAEE managed to reconstruct data points from both the Control and Experiment runs with comparable levels of accuracy. This outcome suggests a notable degree of pattern interchangeability between the Control and Experiment runs, as captured by the AAEE model.

## Appendix B: Optimum number of clusters

Determining the optimum number of clusters in an unsupervised clustering context, such as with the k-means algorithm, can be challenging. We assigned a score to each potential cluster count to identify the optimum number of clusters. This score quantifies how closely the clustering outcome of that specific number of clusters aligns with what would be considered an ideal clustering scenario with the same cluster count. The ideal clustering for each number of clusters is considered a scenario that consistently leads to the emergence of stable clustering patterns, irrespective of the k-means algorithm's random initialization. In other words, in the ideal clustering scenarios, the same set of data points consistently falls within the same cluster, regardless of how the k-means algorithm is initiated.

For each potential number of clusters ranging from 4 to 24, we calculated the score by performing the k-means on the MCAE latent space representation of the data points. We only used the MCAE latent space representation since it performs best in discerning large-scale patterns in our early test with 12 clusters.

We computed a score for each potential number of clusters using the following procedure: We executed the k-means algorithm 1000 times for each number of clusters, enabling us to construct a frequency matrix for each number. Each frequency matrix is a square, two-dimensional array with dimensions equivalent to the square of the total number of data points. In this array, an element denoted by $x_{(i,j)}$ indicates the number of times data points $i$ and $j$ were assigned to the same cluster across the 1000 iterations.

For an ideal clustering scenario, each row of the frequency matrix, regardless of the number of clusters, would just contain elements of either 1000 or zero. The 1000-value elements represent the data points consistently grouped together across iterations, and the rest being zeroes. Consequently, the distribution of the elements within each row mirrors a Dirac delta function peaked at 1000.

For each number of clusters, we then calculated the Wasserstein distance (Rüschendorf, 1985), a metric that quantifies the dissimilarity between two distribution functions, between the distribution function of each row in the observed frequency matrix and the corresponding ideal Dirac delta function. The average of these Wasserstein distances over all rows was taken as the score for that specific number of clusters. This score represents the degree of deviation of our observed clustering with a specific number of clusters from the ideal scenario with the same number of clusters.

Figure B1 shows the clustering scores for different numbers of clusters ranging from 4 to 24. Selecting 5 and 6 clusters yields significantly superior clustering scores, with respective values of 21.7 and 25.5. These results suggest that a choice of 5 or 6 clusters facilitates a more consistent grouping of data points. Despite the closely aligned scores, we opted for 6 as the optimal number of clusters in this study. This choice not only yields a satisfactory clustering score but also provides a more extensive cluster set, which is beneficial for analyzing large-scale circulation patterns.

## Appendix C: Circular continuity in angular features

Angular features represented within the 0-360 degree range can create discontinuity issues due to the circular nature of angular space. Specifically, angularly adjacent values near the boundaries of 0 and 360 degrees can appear numerically distant, presenting challenges for similarity analysis. To address this continuity challenge, we transform each angular feature into two dimensions using sine and cosine functions. By converting each angle into a pair of coordinates based on the sine and cosine of the angle, we establish a two-dimensional representation where the proximity of angles accurately reflects their true angular relationship. Figure C1 shows two proximate angular values, $\theta_1$ and $\theta_2$, along with their corresponding two-dimensional representations on the unit circle, labeled A and B, respectively. Despite their proximity, $\theta_1$ and $\theta_2$ are numerically distant if represented in the angular space, as highlighted by the purple arc in Figure C1. However, the distance between their two-dimensional representations, highlighted by the green line, accurately reflects their proximity. This method ensures that any

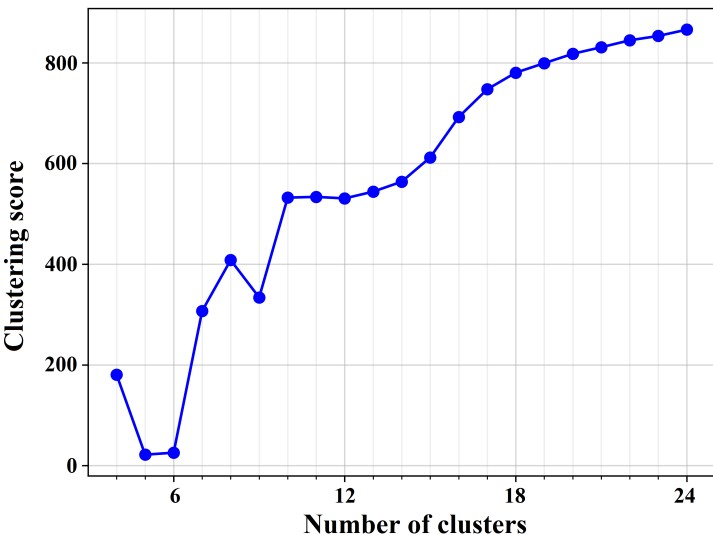

**Figure B1.** The clustering scores computed for a range of potential numbers of clusters.

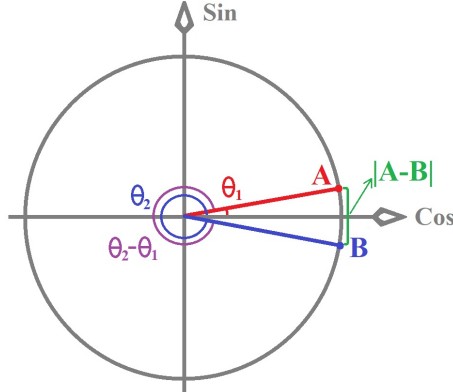

**Figure C1.** Visualization of the transformation applied to angular features to address the angular continuity challenge

angles that are proximate in angular space remain close in this two-dimensional representation, regardless of their numeric angular values.

## Appendix D: Clustering on different data representations

We performed k-means clustering with 6 clusters on various representations of the data points. Moreover, we implemented the SOM with 6 clusters on data space representation for an additional perspective. This approach allowed us to evaluate the underlying patterns within different data representations. The class centers, similar to those presented in Figure 8, and the

monthly occurrence frequency, like those shown in Figure 9, are illustrated in Figures D1 to D10 for different representations. The two-dimensional visualization of the different data representations is presented in Figures D11. We employed t-distributed stochastic neighbor embedding (t-SNE) (Van der Maaten and Hinton, 2008), with a perplexity of 30, a learning rate of 200, and
1000 iterations, to derive the visualizations. In these visualizations, data points are color-coded to represent multiple aspects of the dataset. These aspects include the months during which the data points occurred (left column in Figures D11), highlighting the seasonal cycle across different representations. The color-coding also distinguishes the clusters to which data points belong (middle column in Figures D11), underlining the structure of clustering within these representations. Additionally, we color-coded the longitude where the maximum PMSET is observed at the latitude of $66°\,$N in the lower troposphere for each data
point (right column in Figures D11). This feature is chosen as a representative of the PMSET indices. The t-SNE visualizations in Figures D11 provide an insight into how seasonal cycles, clustering patterns, and PMSET characteristics are embedded across each representation.

    As described in Section 2.5.2, we applied SOM clustering on the data space representation (Kohonen, 1990; Skific et al., 2009; Lee, 2017; Mewes and Jacobi, 2019). We configured the SOM to identify six classes, arranged in a grid of 3 columns
by 2 rows, over 100 iterations. The class center of these SOM classes, along with their monthly occurrence frequency are illustrated in Figures D1 and D2, respectively. Furthermore, we performed k-means clustering with 6 clusters on different data representations including the data space (Figures D3 and D4), a feature space obtained via EOF analysis (Figures D5 and D6), FAE's latent space (Figures D7 and D8), and a representation based on PMSET indices (Figures D9 and D10).

    To derive the EOF feature space, we implemented the EOF analysis with the following specifications. The EOF analysis
was employed as a method for reducing dimensionality (Hannachi et al., 2007; Crasemann et al., 2017). The EOF analysis was applied to the whole Control and Experiment runs data points treating them as one time series. The first five leading EOFs were only used for spanning the reduced representation. The corresponding PC deriving from these EOFs serves as the features within the reduced representation. The five leading EOFs account for $64.7\%$ of the total variance within our dataset.

    Performing the SOM clustering on data space and k-means clustering across data space, EOF feature space, and the FAE's
latent space representation leads to clusters that predominantly characterize the seasonal cycle. A notable difference emerged with k-means clustering on the FAE's latent space, which effectively groups data points within the same season, as depicted in Figure D8. This contrasts with the outcomes from SOM clustering on data space and k-means clustering on both data space and EOF feature space. In these methods, data points are clustered into similar phases of both spring and autumn (Figure D2, D4, and D6). These results indicate that k-means clustering on the FAE's latent space distinctly prioritizes intra-
seasonal similarities, in contrast to the inter-seasonal connections that are more pronounced in the clusters obtained from the data space and EOF feature space. This characteristic is also visible in the t-SNE visualization of these data representations (Figures D11). Specifically, within the EOF and data space visualizations, data points corresponding to transitional seasons (i.e., spring and autumn) are positioned closely together, leading to their categorization into the same group. This suggests a less distinct separation of these transitional seasons within these representations. Conversely, in the t-SNE visualization of the
FAE's latent space representation, data points from these transitional seasons are further apart. This spatial separation facilitates

their categorization into distinct groups. This distinction highlights the capability of the FAE to capture the nuances of seasonal transitions in the generated latent space, distinguishing it from other data representations being evaluated.

Performing the clustering on PMSET indices representation resulted in clusters exhibiting some degree of independence from the seasonal cycle, the dominant pattern presented in the dataset. That is, each cluster is present in every season (Figures D9 and D10). The main features discerned from clustering on the latent space of the MCAE are partially mirrored when performing the clustering on the PMSET indices representation. For instance, the high-pressure anomaly over Scandinavia (cluster 3) showed a decreased occurrence frequency in the Experiment run during autumn-early winter, while the positive NAO-like pattern (cluster 5) showed an increased occurrence frequency in the Experiment run in late winter-early spring.

The t-SNE visualizations of our dataset's representations reveal insightful patterns about the dominance of the seasonal cycle and the embedding of PMSET indices (Figures D11). These visualizations highlight that the seasonal cycle is a prevalent pattern within the data space, EOF, and FAE latent space representations. The seasonal cycle is also discernible within the MCAE latent space representation, where data points belonging to the same season tend to lie closer together, though not as prominently as in the other representations. However, the seasonal cycle is absent in the PMSET indices representation. Moreover, we examine the embedding of a specific PMSET index, the longitude of the maximum PMSET at the latitude of 66° N in the lower troposphere, across these representations (Figures D11). In the t-SNE visualizations of the data space, EOF, and FAE latent space representation, the color-coded PMSET index does not exhibit any discernible pattern, appearing rather as noise within the dataset. This contrasts with how it is embedded in the PMSET indices and, to a lesser extent, the MCAE latent space representations, where data points with similar longitudes of maximum PMSET are found to lie closer together. Thus, the MCAE latent space representation is the only representation that captured both the seasonal cycle, highlighting the dominant pattern observed in the data space, and the PMSET similarity, indicative of the associated PMSET patterns. This quality underlines the MCAE latent space's unique ability to capture and represent both critical aspects of the dataset, distinguishing it from the other representations analyzed.

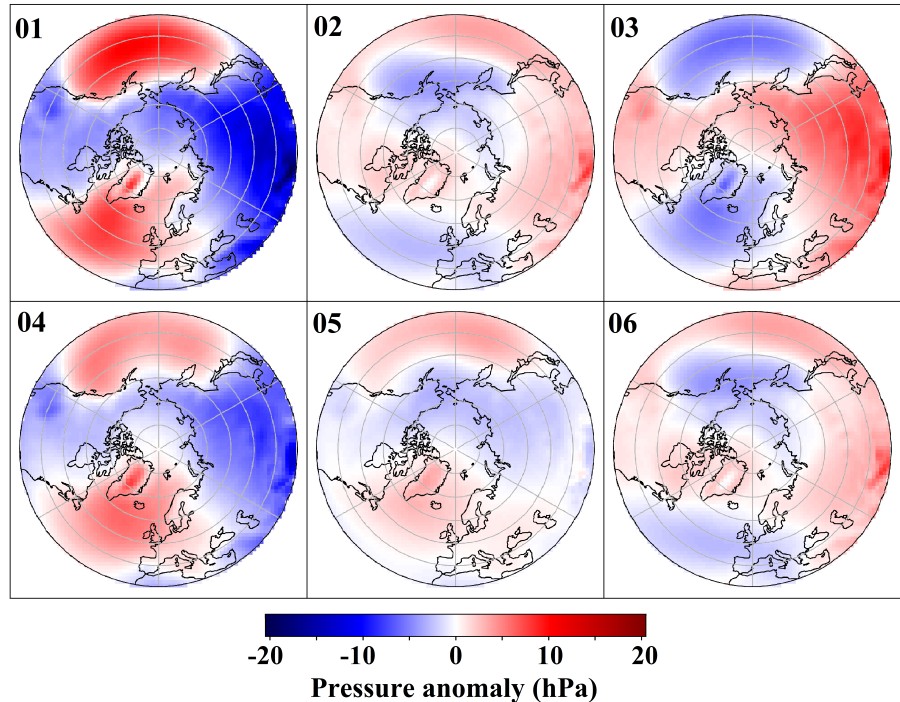

**Figure D1.** Same as Figure 8, but the clusters were derived from applying SOM clustering on the data space.

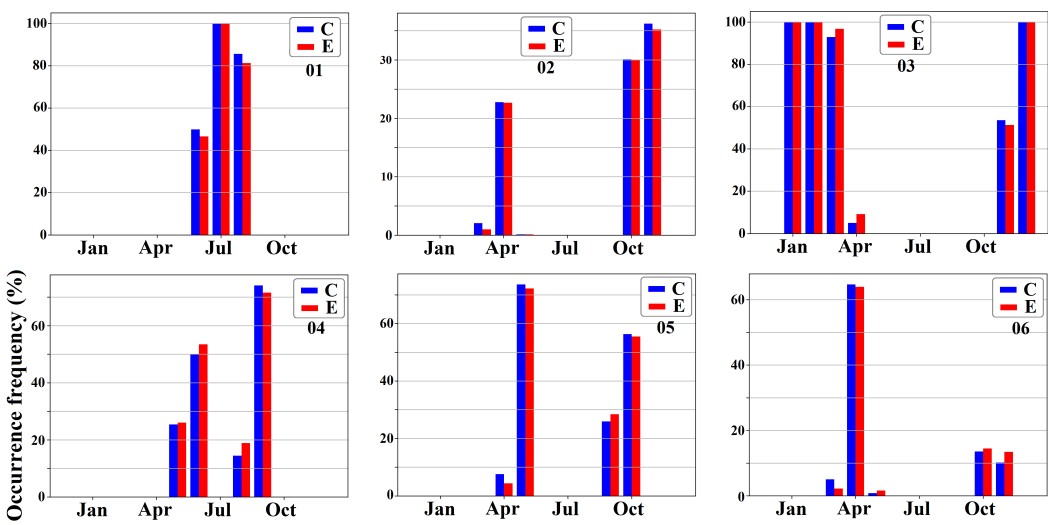

**Figure D2.** The monthly occurrence frequency of the 6 clusters shown in Figure D1 for the Control (blue) and Experiment (red) runs.

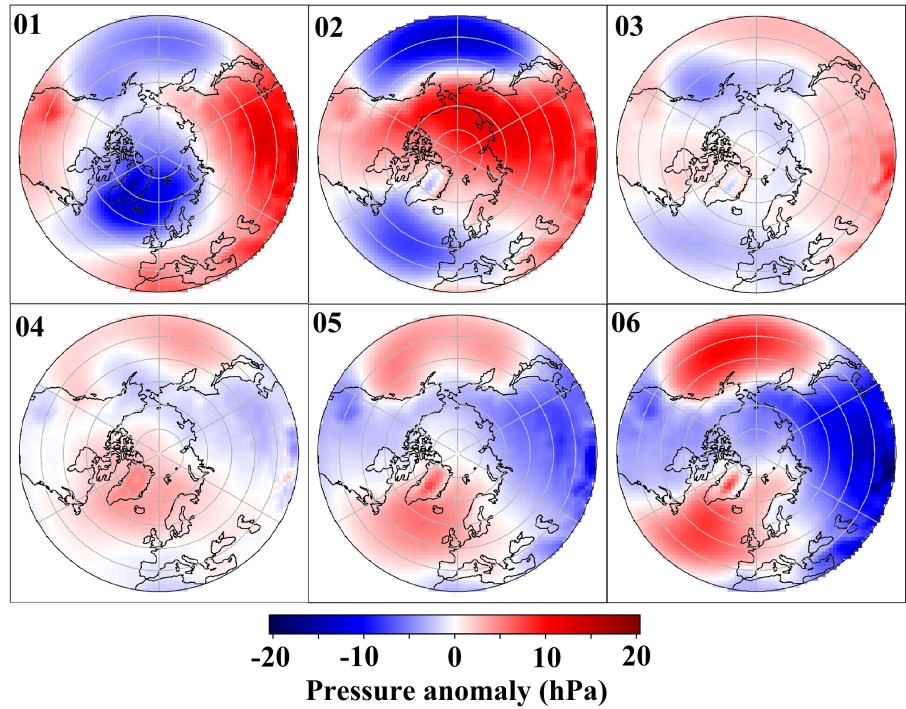

**Figure D3.** Same as Figure 8, but the clusters were derived from applying the k-mean clustering on the data space.

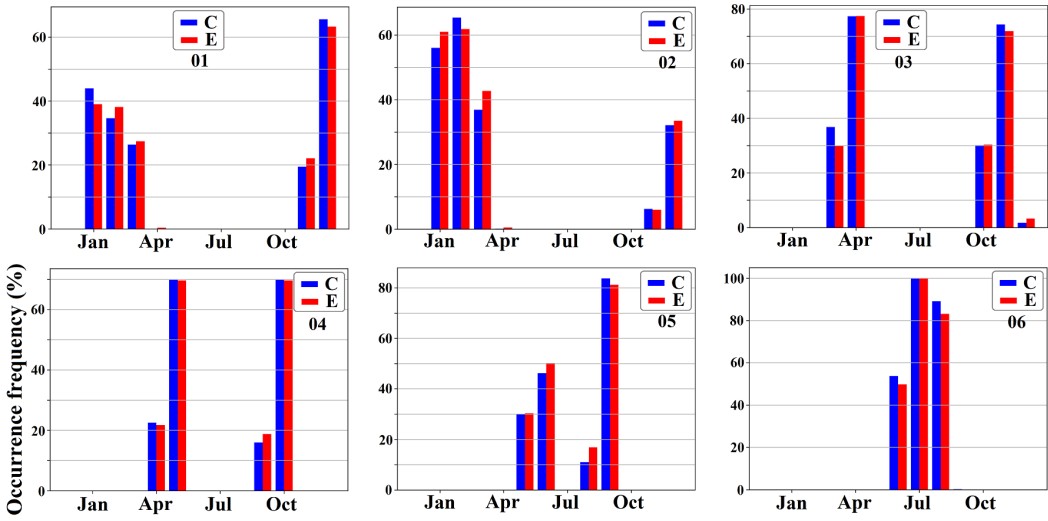

**Figure D4.** The monthly occurrence frequency of the 6 clusters shown in Figure D3 for the Control (blue) and Experiment (red) runs.

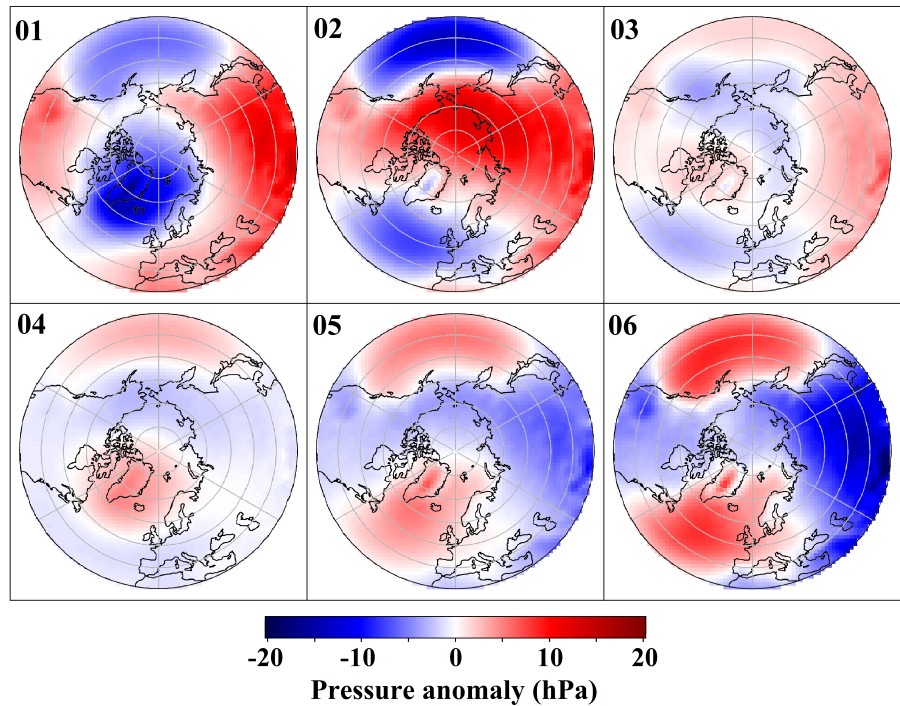

**Figure D5.** Same as Figure 8, but the clusters were derived from applying the k-mean clustering on PC time series derived from the 5 leading EOFs.

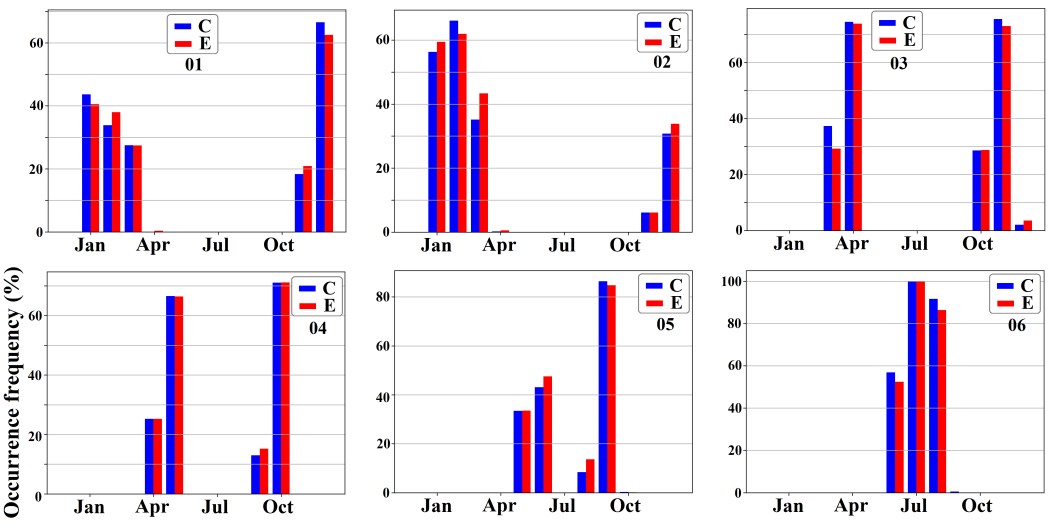

**Figure D6.** The monthly occurrence frequency of the 6 clusters shown in Figure D5 for the Control (blue) and Experiment (red) runs.

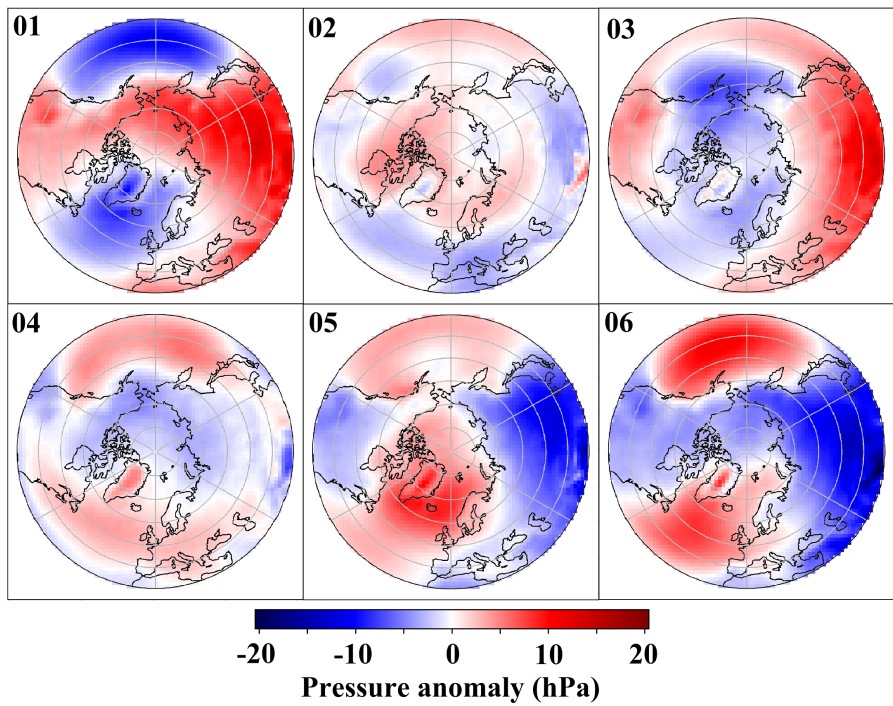

**Figure D7.** Same as Figure 8, but the clusters were derived from applying the k-mean clustering on the FAE's latent space representation of the data points.

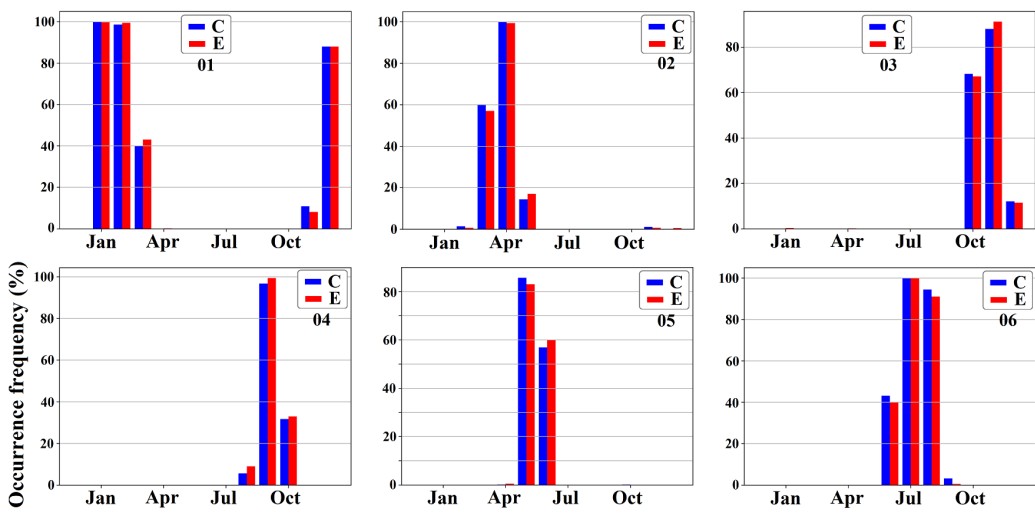

**Figure D8.** The monthly occurrence frequency of the 6 clusters shown in Figure D7 for the Control (blue) and Experiment (red) runs.

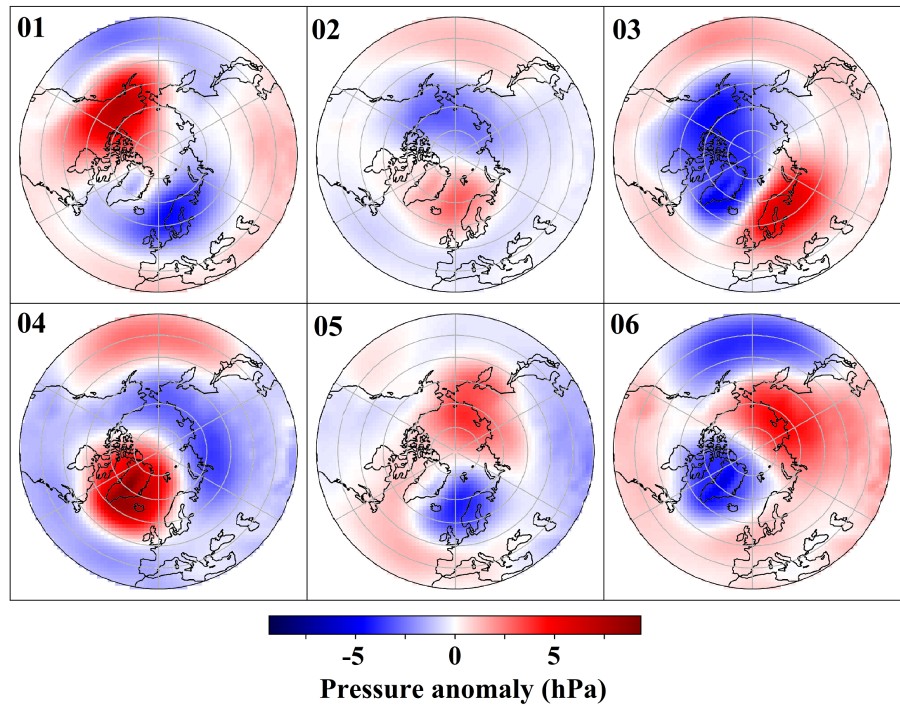

**Figure D9.** Same as Figure 8, but the clusters were derived from applying the k-mean clustering on the PMSET indices representation of data points.

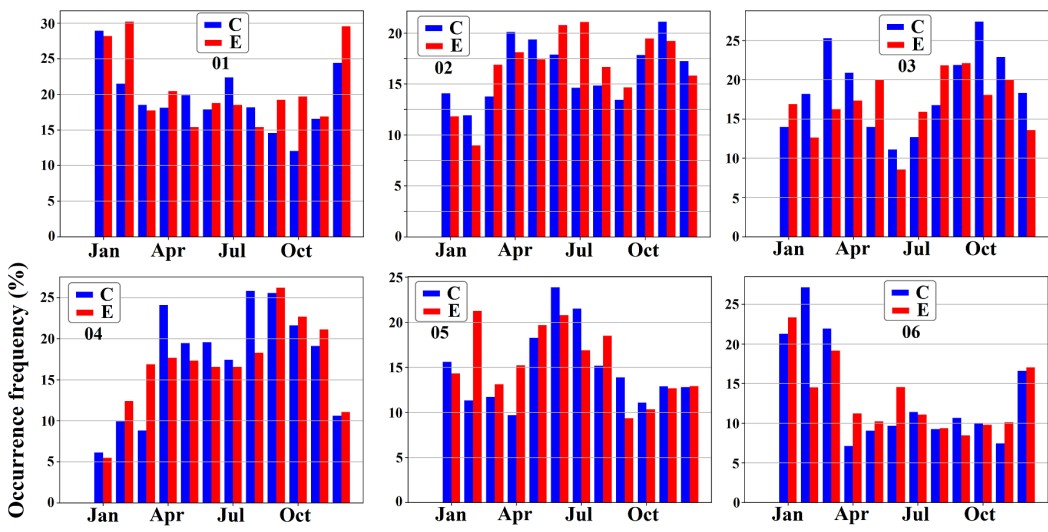

**Figure D10.** The monthly occurrence frequency of the 6 clusters shown in Figure D9 for the Control (blue) and Experiment (red) runs.

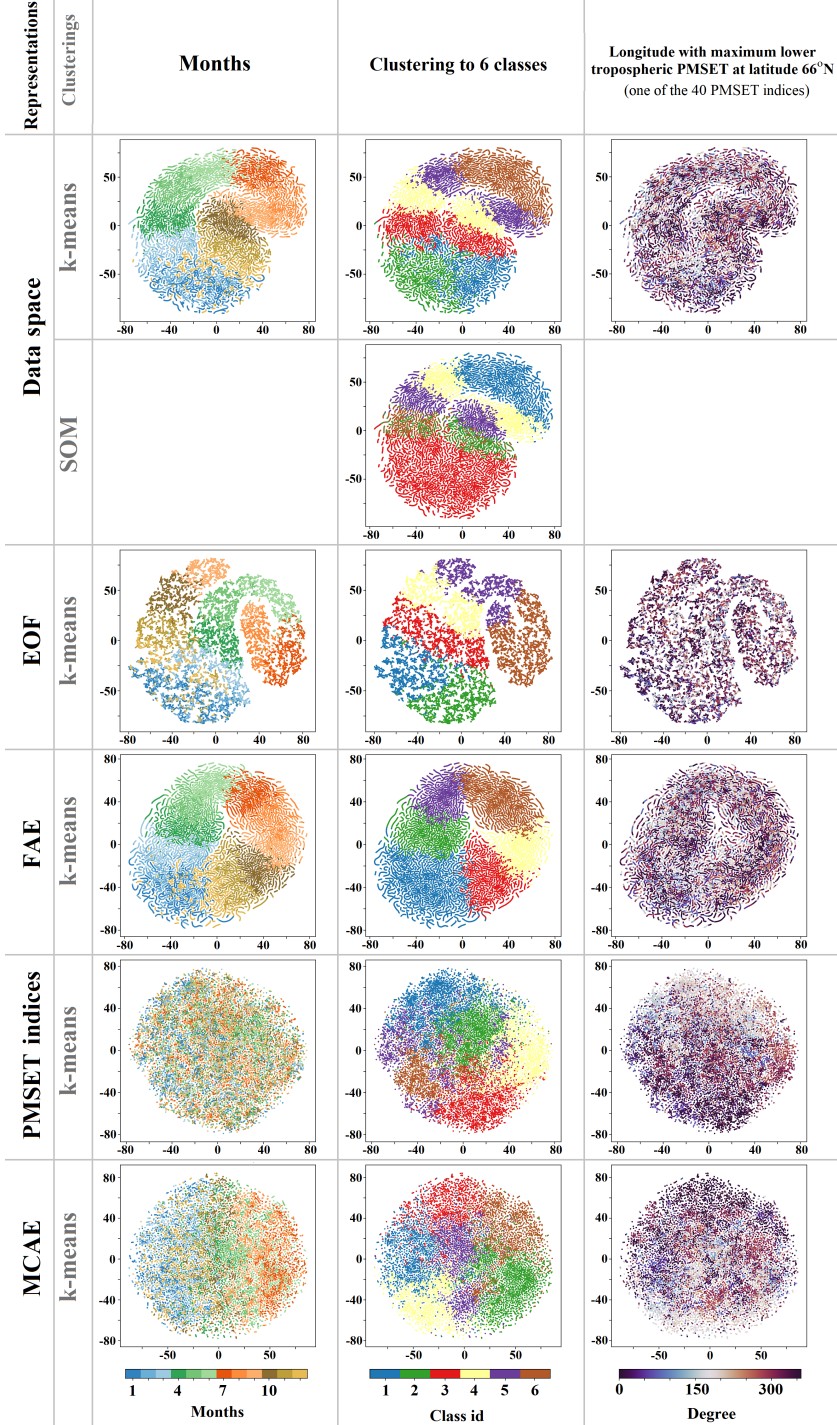

**Figure D11.** t-SNE visualization of the different data representations used for different clustering methods (distinguished by rows). For the SOM clustering (second row), the data space representation is used. The embedded data points are color-coded based on: the occurrence months (left column), assigned clusters (middle column), and the longitude of maximum PMSET at 66° N in the lower troposphere (right column). The latter is one of the PMSET indices.

## Appendix E:  MCAE class centers discrepancies

Using the clustering results derived from k-means clustering on the latent space of MCAE, we calculated the seasonal discrepancies between the class centers across 1,000 bootstrap ensembles. These discrepancies were determined by subtracting the seasonal class center in the Experiment run from that of the Control run. Figure E1 presents these seasonal discrepancies.

In the Experiment run, the centers for each cluster (C1 to C6) reveal seasonal MSLP discrepancies compared to the Control run. For C1, lower MSLP values are consistent over the central Arctic across all seasons. Higher pressure is seen over Eurasia in winter and autumn, contrasted by lower pressure in spring. C2 shows higher MSLP over the Central Arctic in autumn and spring, with a pattern of lower pressure over North Siberia and higher pressure over Eurasia appearing in summer.

C3's Experiment run in autumn features higher MSLP over the Central Arctic, Greenland, and East Asia while indicating lower MSLP over the North Atlantic, Europe, North America, and the Pacific Ocean. The pattern shifts in winter to higher pressure over the Pacific Ocean, Asia, the North Atlantic, and Greenland, and lower pressure in the western Arctic. Springtime sees higher pressure over the Arctic Ocean and North Atlantic and lower MSLP in Eurasia and North America.

For C4, autumn shows lower MSLP over the Central Arctic and the Pacific and Atlantic sectors of the Arctic, countered by higher pressure over Europe, East Asia, and the western coasts of North America's mid-latitudes. Winter brings lower MSLP over the Pacific sector of the Arctic and higher pressure over Europe, East Asia, and North America. Spring features higher pressure in the Central Arctic, the Pacific sector of the Arctic, Eurasia, and Greenland, accompanied by lower MSLP over the Pacific and Atlantic Oceans.

C5 in autumn exhibits lower MSLP in the Central Arctic and the Pacific sector of the Arctic and higher pressure over East Europe, Central Asia, and the west coasts of North America's mid-latitudes. Winter transitions to higher pressure over the Atlantic sector of the Arctic, Greenland, Central Asia, and the Pacific Ocean. Spring maintains higher pressure over the Central Arctic, Greenland, Asia, and North America's mid-latitudes, contrasted by lower pressure over the Pacific and Atlantic Oceans. Finally, C6 during summer presents higher pressure over Eurasia and the Central Arctic and lower pressure over Greenland and the northern part of North America.

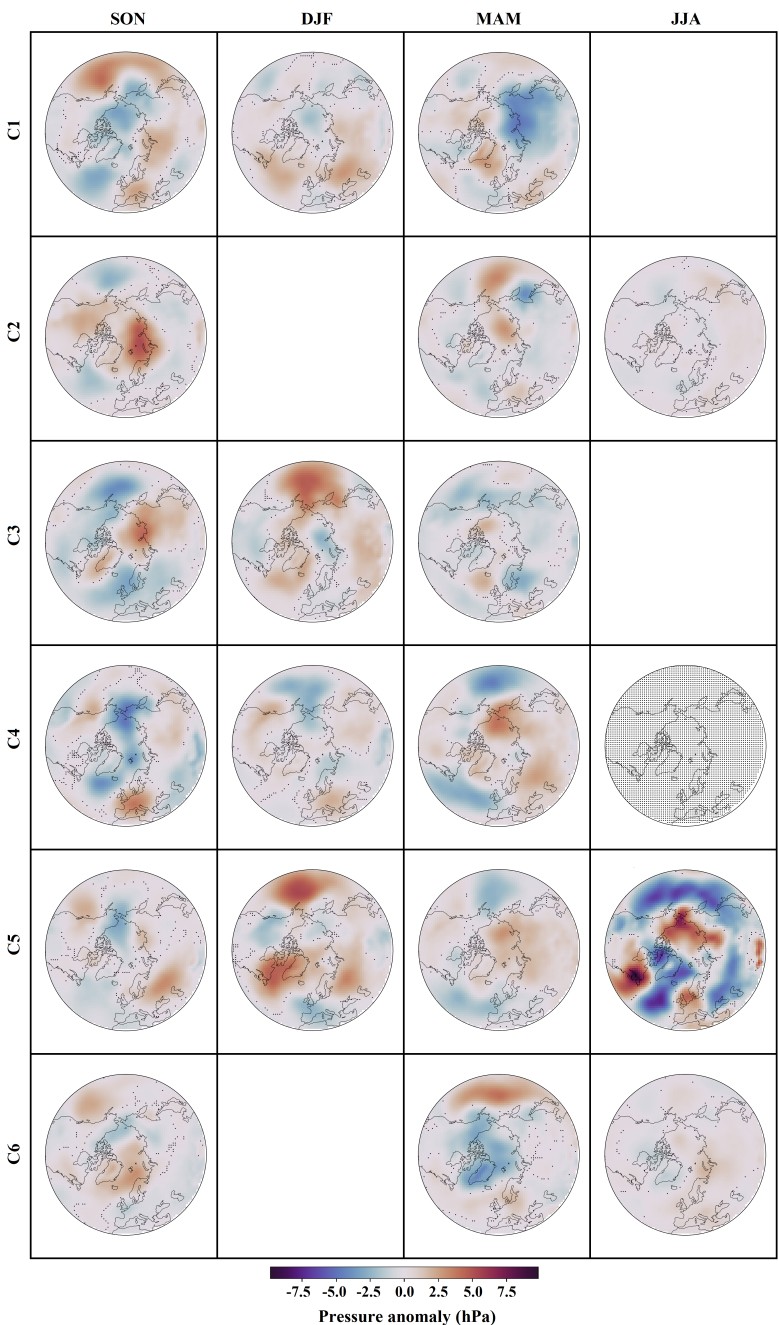

**Figure E1.** Seasonal discrepancy between the Control and Experiment runs, displayed column-wise for each season: autumn (first column), winter (second column), spring (third column), and summer (fourth column). Rows 1 to 6 represent clusters 1 to 6, derived from k-means clustering on the latent space of MCAE. For each season, the discrepancy is obtained by subtracting the cluster centers of the Experiment runs from those of the Control run. The color bar indicating the magnitude of the seasonal discrepancy is found at the bottom of the figure. Statistically non-significant areas are stippled, identified through bootstrapping with 1,000 resamples for each simulation.

**Appendix F: Appendix tables**

**Table A1.** The architecture of the auto-encoders used in this study. The name of layers corresponds to their names in Keras library (Chollet et al., 2015). The layers are stacked together in the same order as presented here. The BatchNormalization layers are used to normalize the data.

| Auto-encoder components | AAE/ AAEE/ FAE layers | MCAE layers |
|---|---|---|
| Encoder | Input(shape=(8,82,82,2)) | Input(shape=(8,82,82,2)) |
| | Conv3D(filters=128) | Conv3D(filters=32) |
| | MaxPooling3D(pool_size=(2, 2, 2)) | MaxPooling3D(pool_size=(1, 2, 2)) |
| | BatchNormalization | BatchNormalization |
| | Conv3D(filters=128) | Conv3D(filters=64) |
| | MaxPooling3D(pool_size=(2, 2, 2)) | MaxPooling3D(pool_size=(1, 2, 2)) |
| | BatchNormalization | BatchNormalization |
| | Conv3D(filters=256) | Conv3D(filters=128) |
| | MaxPooling3D(pool_size=(2, 2, 2)) | MaxPooling3D(pool_size=(2, 2, 2)) |
| | BatchNormalization | BatchNormalization |
| | Conv3D(filters=256) | Conv3D(filters=128) |
| | BatchNormalization | MaxPooling3D(pool_size=(2, 2, 2)) |
| | | BatchNormalization |
| | | Conv3D(filters=256) |
| | | MaxPooling3D(pool_size=(2, 2, 2)) |
| | | BatchNormalization |
| | | Conv3D(filters=256) |
| | | Dense(200) |
| | | Dense(100) |
| Latent space | 25600 free dimensions | 40 MSE index + 60 free dimensions |
| Decoder | Conv3DTranspose(filters=256) | Dense(200) |
| | UpSampling3D(size=(2, 2, 2)) | Dense(1024) |
| | BatchNormalization | Conv3DTranspose(filters=256) |
| | Conv3DTranspose(filters=128) | UpSampling3D(size=(1, 2, 2)) |
| | UpSampling3D(size=(2, 2, 2)) | BatchNormalization |
| | BatchNormalization | Conv3DTranspose(filters=128) |
| | Conv3DTranspose(filters=128) | UpSampling3D(size=(1, 2, 2)) |
| | UpSampling3D(size=(2, 2, 2)) | BatchNormalization |
| | BatchNormalization | Conv3DTranspose(filters=128) |
| | Conv3DTranspose(filters=64) | UpSampling3D(size=(2, 2, 2)) |
| | Conv3DTranspose(filters=2) | BatchNormalization |
| | | Conv3DTranspose(filters=64) |

| Auto-encoder components | AAE/ AAEE/ FAE layers | MCAE layers |
|---|---|---|
| | | UpSampling3D(size=(2, 2, 2)) |
| | | BatchNormalization |
| | | Conv3DTranspose(filters=32) |
| | | UpSampling3D(size=(2, 2, 2)) |
| | | BatchNormalization |
| | | Conv3DTranspose(filters=32) |
| | | Conv3DTranspose(filters=2) |

**Table A2.** Differences in the number of data points allocated to specific clusters for different seasons between the Experiment and Control runs. These differences are calculated as the count from the Experiment run minus that from the Control run. The 95% confidence intervals for these differences, presented in parentheses, are obtained by performing the k-means clustering on the latent space of MCAE 1000 times. For each cluster within each season, (the mean difference for the 1000 clustering) $\pm$ ($2\times$ the standard deviation) is considered as the 95% confidence interval.

| | C1 | C2 | C3 | C4 | C5 | C6 |
|---|---|---|---|---|---|---|
| Autumn (SON) | 29 (18.9:35.3) | -10 (-14.5:-7.9) | -118 (-124.0:-106.9) | 115 (109.5:120.5) | 51 (40.3:60.0) | -67 (-69.1:-62.0) |
| Winter (DJF) | -77 (-90.0:-62.1) | 1 (1.0:1.0) | -14 (-18.5:-10.7) | 111 (100.1:117.2) | -21 (-23.1:-14.6) | 0 (-9.0:8.7) |
| Spring (MAM) | 1 (-4.2:13.7) | -40 (-41.2:-37.4) | -21 (-30.3:-17.4) | 25 (16.0:27.7) | -8 (-14.3:0.1) | 43 (31.0:56.4) |
| Summer (JJA) | 0 (-0.1:0.1) | 88 (85.6:90.4) | -1 (-1.6:-0.9) | 3 (1.29:3.42) | 3 (-4.3:9.9) | -93 (-98.2:-86.0) |

**Table A3.** Summary of the main contributions of different clusters to the seasonal anomalies discussed in Section 3.3, resulting from the negative radiative forcing over Europe. Anomalies mentioned in the table are calculated by subtracting the seasonal mean field of the Control run from the corresponding seasonal field in the Experiment run. The dominant components of each class contribution (WCVC or FSDC) are specified in parentheses.

| Clusters | Class contributions |
|---|---|
| C1 | <ul><li>Contributes to the increased upward and equatorward wave propagation in the upper stratosphere in autumn (WCVC component predominates).</li><li>Contributes to the poleward shift and strengthening of the polar vortex in the upper stratosphere in autumn (dominated by WCVC).</li><li>Major contributor to total tropospheric mean PMSET anomaly in winter (dominated by WCVC).</li><li>Major contributor to the warm Arctic cold mid-latitude anomaly in near-surface air temperature in winter (dominated by WCVC).</li><li>Major contributor to the reduced SIC in the Bering Sea in winter (dominated by WCVC).</li><li>Contributes to the decrease in upward wave propagation extending from the high-latitude upper troposphere to the stratosphere in spring (dominated by WCVC).</li><li>Contributes to the decrease in the zonal mean zonal wind in the high-latitude upper stratosphere in spring (dominated by WCVC).</li></ul> |
| C2 | <ul><li>Major contributor to the autumn mean PMSET increase and decrease in the upper and lower troposphere, respectively (dominated by WCVC).</li><li>Shaped summer PMSET anomalies in the Pacific and East Siberia regions (dominated by WCVC).</li></ul> |
| C3 | <ul><li>Decreased occurrence frequency in autumn leads to a negative contribution to autumn total tropospheric mean PMSET (dominated by FSDC).</li><li>Decreases SIC and increases 2m temperature and PMSET over the Barents-Kara Seas in autumn (dominated by WCVC).</li><li>Increases upward and equatorward wave propagation and reduces EP flux divergence in the mid-latitude stratosphere in autumn (dominated by WCVC).</li><li>Decreased occurrence frequency in autumn leads to minor negative contributions to upward wave propagation in autumn in the mid- and high-latitude upper troposphere (FSDC contribution)</li><li>Contributes to the reduction in upward wave propagation as well as poleward shift and strengthening of the polar vortex in winter (dominated by WCVC).</li><li>Contributes to the increase and decrease in spring mean PMSET in the upper and lower troposphere, respectively (dominated by WCVC).</li></ul> |

| Clusters | Class contributions |
|---|---|
| C4 | <ul><li>Contributes to the increase in 2m temperature over northwest Canada and Alaska in autumn (dominated by WCVC).</li><li>Increased occurrence frequency in autumn leads to a decrease in 2m temperature in North America's mid-latitudes in autumn (FSDC).</li><li>Increased occurrence frequency in autumn leads to reduced upward wave propagation and EP flux divergence in the high-latitude upper stratosphere in autumn (FSDC).</li><li>Contributes to the increased PMSET over Alaska in autumn and winter (both FSDC and WCVC)</li><li>Contributes to the reduced PMSET over East Siberian region in autumn and winter (dominated by WCVC).</li><li>Major Contributor to the decreased SIC over the Sea of Okhotsk and increased 2m temperature over Western Canada in winter (both FSDC and WCVC)</li><li>Major contributor to the reduced upward and equatorward wave propagation, and increased EP flux divergence in the mid-latitude stratosphere in winter (dominated by WCVC).</li><li>Contributes to the increase in upward and equatorward wave propagation in the stratosphere in spring (dominated by WCVC).</li><li>Contributes to the increase in the zonal mean zonal wind in stratosphere in spring (dominated by WCVC).</li></ul> |
| C5 | <ul><li>Contributes to the increased PMSET over Alaska in autumn (dominated by WCVC).</li><li>Contributes to the increased 2m temperature over northwest Canada and Alaska in autumn (dominated by WCVC).</li><li>Contributes to the colder conditions in North America's mid-latitudes in autumn (both FSDC and WCVC).</li><li>Decreases upward wave propagation in the upper stratosphere in autumn (dominated by WCVC).</li><li>Major contributor to the poleward shift and strengthening of the polar vortex in the upper stratosphere in autumn (dominated by WCVC).</li><li>Contributes to the winter mean PMSET decrease and increase in the upper and lower troposphere, respectively (dominated by WCVC).</li><li>Major driver of polar vortex's poleward shift and strengthening in winter (dominated by WCVC).</li><li>Major driver of the decreased EP flux divergence in high-latitude upper stratosphere in spring (dominated by WCVC).</li><li>Major contributor to the increase in the zonal mean zonal wind in the stratosphere in spring (dominated by WCVC).</li></ul> |

| Clusters | Class contributions |
|---|---|
| C6 | • Major contributor to the total tropospheric mean PMSET anomaly in summer (dominated by WCVC).<br>• Contributes to the PMSET anomaly patterns over the Atlantic and Barents-Kara Seas region in summer (dominated by WCVC). |

**Table A4.** Occurrence frequency of major northern hemisphere SSWs during the 30-year simulation period. The table's rows represent Control and Experiment runs, and columns correspond to different months (November to February). Major SSWs are calculated based on the World Meteorological Organization (WMO) criterion, following the methodology used by Karami et al. (2022).

|  | November | December | January | February |
|---|---|---|---|---|
| Control run | 4 | 5 | 5 | 3 |
| Experiment run | 3 | 3 | 7 | 4 |

*Author contributions.* The study was conceived by S.M. and C.J., with significant contributions from all authors. Model simulations were carried out by S.D. and J.Q.. The deep learning framework and the class contribution analysis were developed by S.M. with advice from C.J., D.H., K.K., and I.H.. S.M., C.J., and D.H. contributed to the clustering analysis and its interpretation. The first version of the paper was prepared by S.M. All authors contributed to interpreting the results and writing the final version of the manuscript.

*Competing interests.* The authors declare that no competing interests are present.

*Acknowledgements.* We gratefully acknowledge the funding by the Deutsche Forschungsgemeinschaft (DFG, German Research Foundation) – Projektnummer 268020496 – TRR 172, within the Transregional Collaborative Research Center "ArctiC Amplification: Climate Relevant Atmospheric and SurfaCe Processes, and Feedback Mechanisms (AC)[3]. K.K. and C.J. also acknowledge funding by DFG through grant JA836/47-1.

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
