# Peer review of "Arctic Climate Response to European Radiative Forcing: A Deep Learning Study on Circulation Pattern Changes"

_EGUsphere, 2023_

## Author Comment (AC2)

This paper examines the European radiative forced responses to Arctic climate by using a combination method of two machine learning techniques, the k-mean clustering and convolution neural network. Specifically, the authors classified six patterns and discussed how these six patterns responds to the European radiative forcing. This paper is interesting, and the topic is crucial for the community; however, the manuscript is not well organized and the results are not well highlighted. Therefore, I do not suggest this paper to be published in Weather and Climate Dynamics before a major revision is made.

Thank you very much for your valuable review and for recognizing the interest and importance of the topic for the community. We appreciate your comments and make efforts to improve organization and presentation of our results to highlight the manuscript's contributions more effectively.

To clarify, our study indeed focuses on examining the Arctic climate response to European radiative forcing. We analyze the anomalies calculated as the difference between the experimental runs (pre-industrial era climate with European negative radiative forcing) and the control (pre-industrial era climate without forcing) across key fields such as sea ice concentration (SIC), 2-meter temperature, PMSET, EP flux, and zonal mean zonal wind.

We used a machine learning framework and developed a new analysis method that helped us understand the mechanism through which the observed anomalies happened. That is, we developed our methodology in a way that helps us answer questions like "what are the main mechanisms behind the observed arctic climate response to the forcing."

We are committed to revising the manuscript to address your concerns, including reorganizing the content for better clarity and emphasis on the results. We will also provide additional context to further illustrate the novelty and significance of our findings to the reader.

Once again, we sincerely thank you for your time and constructive feedback, which we believe will greatly improve the quality and impact of our manuscript.

Major comments:

> The authors applied multiple encoding methods and clustered them into six groups of large-scale circulation patterns. However, the main six patterns in section 3.1 are not well-discussed in their physical meaning. Even though the main occurrence seasons are described, the dynamical interactions of each pattern are not investigated. Is there any existing large-scale circulation pattern that is similar with these group? If not, why and how different the patterns found are compared with the existing patterns. For example, by composing the same timing and location with surface temperature or other variables, can we obtain more meaning from these patterns? This will make the later discussions, such as section 3.2, easier since the physical meaning of each pattern is known.

Thank you for highlighting the importance of discussing the physical meaning and dynamical interactions of the identified large-scale circulation patterns. We appreciate the opportunity to clarify how our analysis inherently considers these aspects and tries to find the clusters' significance in forming the observed anomalies as the result of the forcing. To address your comment, we will first delineate the analytical framework we employed to investigate the paper's main scientific questions. This framework will be further highlighted upon more clearly in the revised manuscript.

We considered the large-scale circulation regime as the main mechanism by which our local forcing influences remote regions such as the Arctic. This is why we performed clustering to group the similar large-scale circulation regimes. As our analysis demonstrates that the forcing may not induce new circulation patterns, we examined how changes in the grouped circulation regimes within the Experimental run contributed to observed anomalies in key Arctic climate variables. Given our focus on Arctic climate variables, we utilized the PMSET pattern associated with each circulation as an additional target similarity measure for our clustering. The PMSET pattern, indicative of how each circulation regime transfers energy into the Arctic, is crucial for determining Arctic climate conditions. Therefore, in our clustering approach, circulation regimes within a similar group share not only similar spatiotemporal patterns in MSLP and $\tau_{300-700}$, but also characteristics related to their associated poleward energy transport. Moreover, by reformulating the anomalies to attribute them to different circulation groups, we identified two main mechanisms through which a circulation cluster can contribute to the anomaly: 1) through changes in the occurrence frequency of that cluster as a result of the forcing (FSDC), and 2) through slight adjustments in the cluster's mean characteristics as a result of the forcing (WCVC). This framework helps us delineate how changes in circulation patterns, whether in occurrence frequency or in the cluster's mean characteristics, contribute to the observed anomalies in the key climate variables.

We showed the cluster centers in the data space for the MSLP field in Figures 6 and 8. The class centers give us information on the main cluster spatial patterns. It was nice that we saw some cluster centers that are similar to the well-known circulation patterns. This means that our clustering with a focus on similarities in data points' spatiotemporal patterns and PMSET leads to the circulation regime classification that somehow resembles the well-known large-scale circulation patterns. However, we were careful in associate our clusters with specific well-known circulation patterns, because they have been defined based on different concepts. We will add this argument to section 3.1.

One of the main goals of the paper is to identify the contributions of various classes to the anomalies observed as a consequence of the applied forcing. We aim to understand the dynamical interactions of each pattern that lead to the observed anomaly. In addressing them, we developed our class contribution formulation, focusing specifically on the dynamic interactions of each cluster that lead to the observed anomalies. This approach, as detailed in our manuscript, emphasizes the interplay between class dynamics and the resultant climatic effects, rather than solely on generic class interactions. We will add the emphasis of this point in the revised manuscript.

In conclusion, we developed a method for clustering circulation regimes, utilizing the associated poleward energy transport as a similarity measure. The clusters derived from this method differ from those generated by conventional approaches, leading us to be careful in associating our clusters with specific, well-recognized circulation patterns. Furthermore, through our class contribution method, we explored various class interactions that contributed

to the observed anomaly, thus analyzing the cluster dynamics in the context of the observed anomaly. These insights will be emphasized more in the revised manuscript. We are grateful for your constructive comment, which will undeniably strengthen our manuscript and ensure the clarity and impact of our contributions to the community.

Another suggestion is that the authors discuss Fig. 13 and 14 in details in section 3.1, and point out the most meaningful patterns that we need to focus on. Then in the section 3.2 and so on, the authors do not need to discuss all the patterns, which makes the comparison simpler.

Thank you for your constructive suggestions. In response, we will reorganize the manuscript to enhance its clarity. The revised results section will begin by comparing the mean states of the Control and Experiment runs (highlighting anomalies) across key climate variables. This will be followed by the large-scale circulation clustering and the class contribution analysis. This modification will show the analytical framework of the paper more clearly.

One of the findings of the paper is the varied contributions of different clusters to the observed anomalies across different seasons and locations. Therefore, in most case, one cannot identify one cluster as the sole source of the observed anomaly. This has been highlighted with some examples in lines 630-645. We agree that we didn't emphasize enough on this finding, and we will highlight this argument more in the revised version. Moreover, we will try to put all the most meaningful results in a more compact format like a table to make these part of the paper more concise.

These modifications should make the paper more accessible and emphasize the novelty and significance of our contributions.

The authors have too many figures and no clear storyline is provided. Also, the tittle is about Arctic climate responses to European Radiative forcing, which is too broad and no specific results. The authors should summarize the main responses with a more specific title and describe the corresponding physical process, rather than discuss all the results a little bit. In section 3.1, the authors should decide which patterns to focus and try to understand the physical meaning of the patterns. For instance, the patterns have stronger occurrence in summer are not important for poleward wave propagation. Or in Lines 685-694, the authors focus mainly on one pattern and discuss the possible dynamical process of it, which should be more emphasized or considered as the main conclusion of this manuscript.

Thank you for your valuable feedback, which has highlighted the need for a clearer storyline and a more focused analysis within our manuscript. We acknowledge the complexity introduced by the extensive results, which may have obscured the central storyline of our study. In response, we will reorganize the paper to make the analytical framework of the paper more clear and strength the story line.

Regarding the title, as we analyze the impact of the negative radiative forcing on the Arctic climate by focusing on various climate variables, finding a specific title is challenging, and

we had to use a broad title. However, we will probably consider refining the title to better reflect the paper's content and focus.

As mentioned earlier, the main focus of the study is to evaluate the effect of different circulation clusters on the observed anomalies. This perspective also guides our writing approach, where we identify the most significant patterns in the anomalies and attempt to attribute them to specific clusters. In some case, as you also noted (lines 685-694), the observed anomalies (the anomalies over Barents-Kara sea in autumn) can be attributed to a single cluster. Then, the discussion about attributing the anomaly to the main contributors is reduced to one cluster. However, in most cases, the situation is more complex, with different clusters playing a role. This is why in discussions on the attribution of significant observed anomalies, we typically consider more than one cluster. However, as you suggested, adopting another perspective for discussing the results could involve focusing on a single cluster at a time and detailing its contributions to the anomalies. However, while this approach facilitates a deeper understanding of each cluster's interactions, it complicates the task of comparing the relative contributions of the clusters to the observed anomalies. This is because the emphasis shifts towards the clusters themselves, rather than attributing the observed anomalies across multiple clusters.

In our paper's discussion, we demonstrate the capability of our methodology to identify potential mechanisms behind each significant anomaly. Some examples include:

- Lines 588-592: Our method successfully attributed the SIC loss in Northeast Canada during autumn to mechanisms hidden in seasonal mean anomalies.
- Lines 593-604: This section illustrates our method's ability to attribute an anomaly associated with a single cluster to changes in both the occurrence frequency of the cluster and the main characteristics of the class. Ultimately, we can explain how these changes, driven by the forcing, impact the observed anomaly.
- Lines 605-613: We attributed the 'warm Arctic and cold midlatitude' pattern to changes in the main characteristics of C1.
- Lines 614-634: C3 is identified as the primary driver of sea ice loss in the Barents-Kara seas. Although a reduction in C3's occurrence frequency in autumn slightly decreases upward wave propagation, this effect is minor compared to the dominant increase in wave propagation driven by changes in the class's mean characteristics.
- Lines 635-640: This section provides an example of how different classes interact to form the observed anomaly.
- Lines 641-650: We discuss our method's limitations in identifying mechanisms and offer possible explanations for these limitations.

In the revised manuscript, we will also incorporate a concise summary table that captures the main findings derived from our methodology, emphasizing our study's contributions to understanding the complex mechanisms driving the observed anomalies in the Arctic. This table, alongside a refined narrative, will ensure that our methodology's capabilities and limitations are presented in a coherent and accessible manner.

We hope that the revised manuscript will present a more focused, comprehensible exploration of our research findings.

In section 2.4, the authors trained the AAE with only Control run data and lead to the conclusion in Lines 219-220, "This implies that the negative forcing over Europe **does not introduce new discernible spatiotemporal patterns** in the Northern Hemisphere

extratropical large-scale circulation." However, this conclusion seems to be overstated. The difference of reconstruction loss distribution (Fig.3) between the Control and Experiment runs are limited, indicating the same encoder can be used for both Control and Experiment runs. That is, if there are new discernible spatiotemporal patterns in the Experiment run data, they can also be captured by the encoder of the Control run. I suggest the authors also do another way round (train with Experiment) to confirm that the pattern found in both data can be inter-changeably used. Also, I would suggest to weaken the sentence in Lines 219-220.

Thank you very much for this suggestion and for engaging so deeply with our methodology. We appreciate the opportunity to further discuss the logic behind our approach and consider your valuable feedback.

We understand your concern regarding our conclusion and the potential benefits of training an Autoencoder with Experiment run data as well. Our initial choice to train the AAE exclusively with Control run data was driven by the aim to investigate whether the Experiment run introduces any new spatiotemporal patterns that could not be captured by a model trained on the control run. Moreover, in light of your feedback, we will soften the statement in Lines 219-220 to more accurately reflect the limitations and scope of our findings. More specifically, we will revise it from "This implies that the negative forcing over Europe does not introduce new discernible spatiotemporal patterns in the Northern Hemisphere extratropical large-scale circulation." to "This implies that the negative forcing over Europe **may not** introduce new discernable spatiotemporal patterns in the Northern Hemisphere extratropical large-scale circulation."

Your suggestion to also train a separate Auto encoder with Experiment run data to assess the interchangeability of patterns captured in both Control and Experiment runs is indeed an interesting approach. While it does not directly investigate whether there is any new spatiotemporal pattern in the Experiment run, it could potentially provide additional insights into the similarity or divergence of spatiotemporal patterns between the two scenarios.

We implemented training an auto-encoder with solely Experiment run data points as a supplementary analysis. The reconstruction loss of this autoencoder, calculated by inputting data points from both the Experiment and Control runs, is illustrated in Figure 1. Similar to AAE, the new auto-encoder managed to reconstruct data points from both the Control and Experiment runs with comparable levels of accuracy. This outcome suggests a notable degree of pattern interchangeability between the Control and Experiment runs, as captured by the new model. We will add the resulting reconstructed loss and associated discussion in the appendix of the paper. This will not only address your suggestion but also strengthen our conclusions by providing a more comprehensive understanding of the spatiotemporal pattern dynamics between Control and Experiment runs.

*Figure 1: Reconstruction loss distribution for the Control (C) and Experiment (E) runs generated by an auto-encoder, with a similar architecture as AAE, trained only on the Experiment run data points with the same training strategy as AAE.*

There is little discussion on the mean states between Control and Experiment runs. The authors should first discuss how different it is when the forcing is imposed through traditional methods, such as composite or even EOF/PCA. Since the authors directly start using the six patterns for explaining the forced signature, the results are hard to follow. The authors should give a basic state of how the forced signal look like. And why we need to use such complicated methods to study the forced signal?

Thank you for your insightful comments. We agree that addressing them will significantly enhance the reader's comprehension of our findings.

We will revise the manuscript to begin the results section with a detailed discussion of the mean states and anomalies of various climate variables (PMSET, 2m temperature, SIC, zonal mean zonal wind, and EP flux) as observed in the Control and Experiment runs. The mean state of the different climate variables were discussed at the beginning of the related sections in the pre-print version. Starting our results section by discussion on the climate variables mean states and anomalies, will set a clear foundation for subsequent analyses on large-scale circulation and cluster contributions. We believe this approach will greatly improve the clarity of our analytical framework and storyline.

We also implemented EOF based clustering for comparison purposes. The EOF analysis was applied to the whole Control and Experiment runs data points treating them as one time series. The first five leading EOFs were only used for spanning the reduced representation. The corresponding PC deriving from these EOFs serves as the features within the reduced representation. The five leading EOFs account for 64.7% of the total variance within our dataset. The centers of these classes, along with their monthly occurrence frequencies, are depicted in Figure 2 and Figure 3, respectively. The class centers and occurrence frequencies resulting from this clustering approach are similar to those obtained by performing k-means

clustering directly on the data space. We will elaborate on this comparison in the Appendix section of the revised manuscript.

Additionally, we will include t-SNE visualizations on different data representations in the Appendix to further argue the advantages of our chosen methodology over traditional approaches. This visualization aims to provide a compelling argument for the proficiency of our method by highlighting the distinct data embeddings and the insights they offer.

We are confident that the addition of these analyses and the reorganization of our results section will not only strengthen our argument but also significantly improve the overall quality of our manuscript.

[Figure]

*Figure 2: Mean MSLP fields for clusters derived from k-means clustering (6 clusters) on PC time series derived from the 5 leading EOFs. The fields for each cluster are plotted as anomalies relative to the mean MSLP field for all data points, encompassing both the Control and Experiment runs.*

[Figure]

Figure 3: The monthly occurrence frequency of the 6 clusters showed in Figure 2 for the Control (blue) and Experiment (red) runs.

Other comments are following:

B1-04 is similar with Fig. B3-01; however, their occurrence seasons are so different. Why? Same for the Fig. B5-05. Are they corresponding to the C1 and C2 in Fig.6? Even though the methods are different, the results should match between each other in certain degree.

Thank you for pointing out the similarities and differences in occurrence seasons between the Figures B1, B3 and 6. Your question highlights a key aspect of different clustering methodologies, which is the dependency of clustering outcomes on the chosen representation space. Different representation spaces can influence which data points are considered close to each other according to the specific geometry of that space. Clustering algorithms like k-means then group data points based on their proximity in the chosen representation space. For instance, the clustering performed on PMSET indices (Figure B1) is guided solely by similarities in PMSET indices. That is, data points with similar PMSET indices lie closer together in this representation. This contrasts with the clustering performed on data space (Figure B3), where the Euclidean pixel-wise distance between data points dictates the data points proximity and the resulting clustering.

As you mentioned, there are similarities between Class 04 in Figure B1 and Class 01 in Figure B3, suggesting they could be perceived as negative phases of each other if viewed as EOF patterns. However, they have different monthly occurrence frequency. It is important to notice that they are not EOF patterns, nor do they originate from clustering with the identical focus, as discussed in the previous paragraph. Cluster 04 in Figure B1 is derived from clustering data points based on their similarities in PMSET indices, meaning that data points forming this cluster primarily exhibit similar PMSET characteristics. On the other hand, Cluster 01 in Figure B3 represents one of the mean states of winter circulation. It emerges from clustering based on data space, which primarily groups the seasonal cycle.

To identify similarities in patterns, one should compare clustering results that share a similar focus. For instance, k-means clustering on data space, as shown in Figures B3 and B4, and clustering on the FAE latent space both primarily reveal the seasonal cycle as the dominant pattern within the dataset. Consequently, the class centers in these representations are

somewhat comparable. A case in point is Cluster 06 in Figure B3 and Cluster 01 in Figure B5 can be perceive as similar clusters. Similarly, clustering based on PMSET indices (Figures B1 and B2) and clustering on the MCAE latent space exhibit resemblances because both approaches utilize PMSET indices as a similarity measure at different levels. For example, Cluster C5 in Figure 6 of the manuscript and Class 04 in Figure B1 show some similarities.

The divergent outcomes from clustering across different representation spaces underscores the sensitivity of clustering outcomes to the representation space. Conventional clustering methodologies only focus on data point similarities using standard measures of similarity, such as Euclidean pixel-wise distance between data points or PCA-reduced (Principal Component Analysis) space representation. One of the main contribution of this paper is to introduced a method that can produce a representation of the data points that can effectively distance the data points based on both the similarities among the data points and their alignment with a target objective similarity, here the PMSET patterns. We will better emphasize this contribution in the revised version and further use t-SNE visualization to provide deeper insights into how various clustering methods perform across different representations.

Line 231, why 40 indices in total? 4 latitudes with 5 PMSET values and two levels of troposphere? It is easy to miss the two levels of troposphere.

Our decision to choose these specific indices was driven by a desire to comprehensively capture the three-dimensional spatial patterns of the PMSET associated with the data points, which is crucial for understanding the dynamics of the atmosphere energy transport in our study.

To elaborate, the five PMSET indices per profile are intended to represent the main characteristics of the PMSET profiles. Furthermore, the choice of four latitudes was aimed at capturing the latitudinal progression of the PMSET. Additionally, incorporating two vertical levels within the troposphere was intended to provide insights into the vertical dynamics of the PMSET. The two vertical levels allow us to observe how PMSET values change with altitude, offering insight into the vertical distribution of atmospheric energy transport.

We appreciate the opportunity to clarify this in our manuscript. We will revise section 2.5.1 of the manuscript to highlight our reasoning behind selecting these indices.

Line 349, how does the authors consider the "statistically significant increases" in Fig. 10?

As we mentioned in line 119-122, the independent two sample t-test and Welch's unpaired t-test were used throughout the study to determine the statistical significance levels when samples have equal or unequal variances, respectively, and samples with p-value less than 0.05 were considered statistically significant. Using the same methodology, the doted points in figure 10 are consider statistically significant

Line 359-361, "between the daily mean PMSET" of upper and lower troposphere?

Exactly. We appreciate you highlighting this point. In the revised version of our manuscript, we will refine this sentence to more precisely express that our analysis involves comparing the daily mean PMSET values between the upper and lower levels of the troposphere.

As a final remark, we would like to highlight the main contributions of our paper, which we plan to better emphasize on the final revision of the manuscript:

1 - We introduced a deep-learning-based method that generates an effective representation of data points. This representation accounts for both the similarities among the data points and their alignment with a target objective similarity, here the PMSET patterns, to effectively distance the data points. This method was applied to data points consisting of 8 consecutive days of MSLP and $\tau_{300-700}$, capturing dynamic patterns in the circulation data points.

2- We developed a formulation for class contribution, focusing on the dynamic interactions of different circulation patterns that lead to observed anomalies. By reformulating the anomalies to attribute them to different circulation groups, we identified two main mechanisms through which a circulation cluster can contribute to the anomaly: (1) through changes in the occurrence frequency of that cluster as a result of the forcing (FSDC), and (2) through slight adjustments in the cluster's mean characteristics as a result of the forcing (WCVC). This framework delineates how changes in circulation patterns, whether in occurrence frequency or in the cluster's mean characteristics, contribute to the observed anomalies in key climate variables.

3- We utilized this advanced methodological approach to understand the role of different circulation patterns in forming the anomalies. These patterns were grouped based on similarities in their spatiotemporal behavior and their associated PMSET patterns. This grouping and the class contribution formulation helped us understand how these interactions lead to observed anomalies resulting from negative radiative forcing over Europe. Among the main findings are:

- In most cases, the change in a cluster's main behavior is more significant than changes in the occurrence frequency of that cluster when considering the cluster's contribution to the anomaly. For example, the decreased occurrence frequency of a cluster associated with a high-pressure system over Northern Eurasia and Scandinavia in autumn, observed in the Experiment run, led to reduced upward wave propagation. However, a slight adjustment in the mean behavior of this cluster in the Experiment run resulted in an increase in upward wave propagation. This increased wave propagation dominates the cluster's contribution to the observed anomaly in upward wave propagation.
- Our method's capability to attribute observed anomalies, such as the SIC loss in the Barents-Kara Seas in autumn, the warm Arctic and cold mid-latitude in winter, and others.

---

## Author Response (AR1)

We thank the referees for the insightful comments, which have greatly helped us improve the manuscript. Below, we repeat the reviewers' remarks in red italics, and add our respective responses in normal text.

**_Reviewer 1:_**

_This paper examines the European radiative forced responses to Arctic climate by using a combination method of two machine learning techniques, the k-mean clustering and convolution neural network. Specifically, the authors classified six patterns and discussed how these six patterns responds to the European radiative forcing. This paper is interesting, and the topic is crucial for the community; however, the manuscript is not well organized and the results are not well highlighted. Therefore, I do not suggest this paper to be published in Weather and Climate Dynamics before a major revision is made._

Thank you very much for your valuable review and for recognizing the interest and importance of the topic for the community. We appreciate your comments and make efforts to improve organization and presentation of our results to highlight the manuscript's contributions more effectively.

To clarify, our study indeed focuses on examining the Arctic climate response to European radiative forcing. We analyze the anomalies calculated as the difference between the Experiment runs (pre-industrial era climate with European negative radiative forcing) and the Control run (pre-industrial era climate without forcing) across key fields such as sea ice concentration (SIC), 2-meter temperature, PMSET, EP flux, and zonal mean zonal wind.

We used a machine learning framework and developed a new analysis method that helped us understand the mechanism through which the observed anomalies happened. That is, we developed our methodology in a way that helps us answer questions like "what are the main mechanisms behind the observed arctic climate response to the forcing."

We revised the manuscript to address your concerns, including reorganizing the content for better clarity and emphasis on the results. We also provided additional context to further illustrate the novelty and significance of our findings to the reader. Specifically, we have enhanced the manuscript throughout by adding detailed explanations where necessary to clarify the structure and storyline of the paper. The abstract is revised to contain the main contributions of the paper. The results section has been restructured to begin with a comparison of the mean states of the Control and Experiment runs, followed by analyses of large-scale circulation clustering and class contributions. Furthermore, we have introduced Appendices A and C in the revised manuscript and enriched the content and analysis in Appendix D.

Once again, we sincerely thank you for your time and constructive feedback, which we believe greatly improved the quality and impact of our manuscript.

_Major comments:_

*The authors applied multiple encoding methods and clustered them into six groups of large-scale circulation patterns. However, the main six patterns in section 3.1 are not well-discussed in their physical meaning. Even though the main occurrence seasons are described, the dynamical interactions of each pattern are not investigated. Is there any existing large-scale circulation pattern that is similar with these group? If not, why and how different the patterns found are compared with the existing patterns. For example, by composing the same timing and location with surface temperature or other variables, can we obtain more meaning from these patterns? This will make the later discussions, such as section 3.2, easier since the physical meaning of each pattern is known.*

Thank you for highlighting the importance of discussing the physical meaning and dynamical interactions of the identified large-scale circulation patterns. We appreciate the opportunity to clarify how our analysis inherently considers these aspects and tries to find the clusters' significance in forming the observed anomalies as the result of the forcing. To address your comment, we first delineate the analytical framework we employed to investigate the paper's main scientific questions. This framework is also further highlighted upon more clearly in the revised manuscript.

We considered the large-scale circulation regime as the main mechanism by which our local forcing influences remote regions such as the Arctic. This is why we performed clustering to group the similar large-scale circulation regimes. As our analysis demonstrates that the forcing may not induce new circulation patterns, we examined how changes in the grouped circulation regimes within the Experimental run contributed to observed anomalies in key Arctic climate variables. Given our focus on Arctic climate variables, we utilized the PMSET pattern associated with each circulation as an additional target similarity measure for our clustering. The PMSET pattern, indicative of how each circulation regime transfers energy into the Arctic, is crucial for determining Arctic climate conditions. Therefore, in our clustering approach, circulation regimes within a similar group share not only similar spatiotemporal patterns in MSLP and $\tau_{300-700}$, but also characteristics related to their associated poleward energy transport. Moreover, by reformulating the anomalies to attribute them to different circulation groups, we identified two main mechanisms through which a circulation cluster can contribute to the anomaly: 1) through changes in the occurrence frequency of that cluster as a result of the forcing (FSDC), and 2) through slight adjustments in the cluster's mean characteristics as a result of the forcing (WCVC). This framework helps us delineate how changes in circulation patterns, whether in occurrence frequency or in the cluster's mean characteristics, contribute to the observed anomalies in the key climate variables (these points are highlighted in lines 5, 319-324, 377-387, in the revised manuscript).

We show the cluster centers in the data space for the MSLP field in Figures 9 and 11 of the revised manuscript. The class centers give us information on the main cluster spatial patterns. It was nice that we saw some cluster centers that are similar to the well-known circulation patterns. This means that our clustering with a focus on similarities in data points' spatiotemporal patterns and PMSET leads to the circulation regime classification that somehow resembles the well-known large-scale circulation patterns. However, we were careful in associating our clusters with specific well-known circulation patterns, because they have been defined based on different concepts. This argument is added to lines 421-426 of the revised manuscript.

One of the main goals of the paper is to identify the contributions of various classes to the anomalies observed as a consequence of the applied forcing. We aim to understand the dynamical interactions of each pattern that lead to the observed anomaly. In addressing them, we developed our class contribution formulation, focusing specifically on the dynamic interactions of each cluster that lead to the observed anomalies. This approach, as detailed in section 2.6 of the manuscript, emphasizes the interplay between class dynamics and the resultant climatic effects, rather than solely on generic class interactions. This argument is added to in lines 428-434 of the revised manuscript.

In conclusion, we developed a method for clustering circulation regimes, utilizing the associated poleward energy transport as a similarity measure. The clusters derived from this method differ from those generated by conventional approaches, leading us to be careful in associating our clusters with specific, well-recognized circulation patterns. Furthermore, through our class contribution method, we explored various class interactions that contributed to the observed anomaly, thus analyzing the cluster dynamics in the context of the observed anomaly. These insights is emphasized more in the revised manuscript. We are grateful for your constructive comment, which undeniably strengthened our manuscript and ensure the clarity and impact of our contributions to the community.

*Another suggestion is that the authors discuss Fig. 13 and 14 in details in section 3.1, and point out the most meaningful patterns that we need to focus on. Then in the section 3.2 and so on, the authors do not need to discuss all the patterns, which makes the comparison simpler.*

Thank you for your constructive suggestions. In response, we reorganized the manuscript to enhance its clarity. The revised results section begin by comparing the mean states of the Control and Experiment runs (highlighting anomalies) across key climate variables (section 3.1 of the revised manuscript). We also added the introductory paragraph for the result part, making it clearer how this part is structured (lines 319-324 of the revised manuscript). The anomaly fields for PMSET, 2m temperature, SIC, zonal mean zonal wind, and atmospheric wave propagation, along with the required climatology plots and analysis, have been moved to this section. This section is followed by the large-scale circulation clustering and the class contribution analysis. We hope that this modification shows the analytical framework of the paper more clearly.

One of the findings of the paper is the varied contributions of different clusters to the observed anomalies across different seasons and locations. Therefore, in most case, one cannot identify one cluster as the sole source of the observed anomaly. This has been highlighted with some examples in lines 694-709. Moreover, we put all the main class contributions results in Table A3 to make a more concise version of this part of the paper available.

These modifications made the paper more accessible and emphasize the novelty and significance of our contributions.

*The authors have too many figures and no clear storyline is provided. Also, the tittle is about Arctic climate responses to European Radiative forcing, which is too broad and*

*no specific results. The authors should summarize the main responses with a more specific title and describe the corresponding physical process, rather than discuss all the results a little bit. In section 3.1, the authors should decide which patterns to focus and try to understand the physical meaning of the patterns. For instance, the patterns have stronger occurrence in summer are not important for poleward wave propagation. Or in Lines 685-694, the authors focus mainly on one pattern and discuss the possible dynamical process of it, which should be more emphasized or considered as the main conclusion of this manuscript.*

Thank you for your valuable feedback, which has highlighted the need for a clearer storyline and a more focused analysis within our manuscript. We acknowledge the complexity introduced by the extensive results, which may have obscured the central storyline of our study. In response, we reorganized the paper to make the analytical framework of the paper more clear and strengthen the storyline. Specifically we added the an introductory paragraph at the beginning of each section of the results part (lines 319-324, 377-387, and 428-434)

We change the title to "**Arctic Climate Response to European Radiative Forcing: A Deep Learning Study on Circulation Pattern Changes**". Incorporating a reference to circulation patterns in the title of the paper should underscore their pivotal role in our research, providing a more precise and insightful descriptor of our study's focus. By emphasizing this aspect, the title better communicates the core analytical approach.

As mentioned above, the main focus of the study is to evaluate the effect of different circulation clusters on the observed anomalies. This perspective also guides our writing approach, where we identify the most significant patterns in the anomalies and attempt to attribute them to specific clusters. In some case, as you also noted (lines 749-763 in the revised manuscript), the observed anomalies (the anomalies over Barents-Kara sea in autumn) can be attributed to a single cluster. Then, the discussion about attributing the anomaly to the main contributors is reduced to one cluster. However, in most cases, the situation is more complex, with different clusters playing a role. We found it a plausible explanation for the intermittency of certain mechanisms proposed in the literature, such as the role of the Barents-Kara sea's SIC reduction in autumn in polar vortex disturbances (lines 671-672, 685-690, and 758-763). This is why in discussions on the attribution of significant observed anomalies, we typically consider more than one cluster. However, as you suggested, adopting another perspective for discussing the results could involve focusing on a single cluster at a time and detailing its contributions to the anomalies (this approach is considered in forming Table A3 of the revised manuscript). However, while this approach facilitates a deeper understanding of each cluster's interactions, it complicates the task of comparing the relative contributions of the clusters to the observed anomalies. This is because the emphasis shifts towards the clusters themselves, rather than attributing the observed anomalies across multiple clusters.

In our discussion, we demonstrate the capability of our methodology to identify potential mechanisms behind each significant anomaly. Some examples include:

- Lines 634-644 of the revised manuscript: Our method successfully attributed the SIC loss in Northeast Canada during autumn to mechanisms hidden in seasonal mean anomalies. This has been more highlighted at the end of the paragraph (lines 641-644)
- Lines 645-656 of the revised manuscript: This section illustrates our method's ability to attribute an anomaly associated with a single cluster to changes in both the occurrence

frequency of the cluster and the main characteristics of the class. Ultimately, we can explain how these changes, driven by the forcing, impact the observed anomaly.

- Lines 657-665 of the revised manuscript: We attributed the 'warm Arctic and cold midlatitude' pattern to changes in the main characteristics of C1.
- Lines 666-693 of the revised manuscript: C3 is identified as the main factor contributing to sea ice loss in the Barents-Kara seas during autumn, primarily driven by discrepancies in class characteristics in the Experimental run. This sea ice reduction coincides with a disturbed polar vortex in the lower stratosphere, aligning with increased disturbances in stratospheric dynamics. Although a reduction in C3's occurrence frequency in autumn slightly decreases upward wave propagation, this effect is minor compared to the dominant increase in wave propagation driven by changes in the class's mean characteristics. Overall, the influence of C3 on stratospheric dynamics, though pivotal, is part of a broader interplay of various factors, indicating that its singular contribution is less dominant than the collective influence of other classes. Some sentence parts are added to the paragraph to make argument more clear (lines 671-673,and 685-690).
- Lines 694-699 of the revised manuscript: This section provides an example of how different classes interact to form the observed anomaly, particularly, focus on how different clusters contribute to forming the stratospheric dynamics in autumn.
- Lines 710-724 of the revised manuscript: We discuss our method's limitations in identifying mechanisms and offer possible explanations for these limitations.

In the revised manuscript, we also incorporate Table A3 which provides a concise summary of the main findings derived from our methodology, emphasizing our study's contributions to understanding the complex mechanisms driving the observed anomalies in the Arctic. This table, alongside a refined narrative, ensure that our methodology's capabilities and limitations are presented in a coherent and accessible manner.

We believe that the revised manuscript present a more focused, comprehensible exploration of our research findings.

*In section 2.4, the authors trained the AAE with only Control run data and lead to the conclusion in Lines 219-220, "This implies that the negative forcing over Europe **does not introduce new discernible spatiotemporal patterns** in the Northern Hemisphere extratropical large-scale circulation." However, this conclusion seems to be overstated. The difference of reconstruction loss distribution (Fig.3) between the Control and Experiment runs are limited, indicating the same encoder can be used for both Control and Experiment runs. That is, if there are new discernible spatiotemporal patterns in the Experiment run data, they can also be captured by the encoder of the Control run. I suggest the authors also do another way round (train with Experiment) to confirm that the pattern found in both data can be inter-changeably used. Also, I would suggest to weaken the sentence in Lines 219-220.*

Thank you very much for this suggestion and for engaging so deeply with our methodology. We appreciate the opportunity to further discuss the logic behind our approach and consider your valuable feedback.

We understand your concern regarding our conclusion and the potential benefits of training an auto-encoder with Experiment run data as well. Our initial choice to train the AAE exclusively

with Control run data was driven by the aim to investigate whether the Experiment run introduces any new spatiotemporal patterns that could not be captured by a model trained on the Control run. Moreover, in light of your feedback, we softened the statement in Lines 223-224 of the revised manuscript to more accurately reflect the limitations and scope of our findings. More specifically, we revised it from "This implies that the negative forcing over Europe does not introduce new discernible spatiotemporal patterns in the Northern Hemisphere extratropical large-scale circulation." to "This implies that the negative forcing over Europe **may not** introduce new discernable spatiotemporal patterns in the Northern Hemisphere extratropical large-scale circulation."

Your suggestion to also train a separate auto-encoder with Experiment run data to assess the interchangeability of patterns captured in both Control and Experiment runs is indeed an interesting approach. While it does not directly investigate whether there is any new spatiotemporal pattern in the Experiment run, it could potentially provide additional insights into the similarity or divergence of spatiotemporal patterns between the two scenarios.

We implemented training an auto-encoder with solely Experiment run data points as a supplementary analysis. The reconstruction loss of this auto-encoder, calculated by inputting data points from both the Experiment and Control runs, is illustrated in Figure 1 of our reply. Similar to AAE, the new auto-encoder managed to reconstruct data points from both the Control and Experiment runs with comparable levels of accuracy. This outcome suggests a notable degree of pattern interchangeability between the Control and Experiment runs, as captured by the new model (this argument is added to lines 224-226 and 598-600 of the revised manuscript). We also added the resulting reconstruction loss and associated discussion in the Appendix A of the revised paper. This not only addressed your suggestion but also strengthened our conclusions by providing a more comprehensive understanding of the spatiotemporal pattern dynamics between Control and Experiment runs.

[Figure]

*Figure 1: Reconstruction loss distribution for the Control (C) and Experiment (E) runs generated by an auto-encoder, with a similar architecture as AAE, trained only on the Experiment run data points with the same training strategy as AAE.*

*There is little discussion on the mean states between Control and Experiment runs. The authors should first discuss how different it is when the forcing is imposed through traditional methods, such as composite or even EOF/PCA. Since the authors directly start using the six patterns for explaining the forced signature, the results are hard to follow. The authors should give a basic state of how the forced signal look like. And why we need to use such complicated methods to study the forced signal?*

Thank you for your insightful comments. We agree that addressing them before applying our method will significantly enhance the reader's comprehension of our findings.

We revised the manuscript to begin the results section with a detailed discussion of the mean states and anomalies of various climate variables (PMSET, 2m temperature, SIC, zonal mean zonal wind, and EP flux) as observed in the Control and Experiment runs. This section title is "Mean anomalies exerted by the forcing" in line 325 of the revised manuscript. The mean state of the different climate variables are discussed at the beginning of the related sections in the pre-print version are revised and transferred to this section. Starting our results section by discussion on the climate variables mean states and anomalies, set a clear foundation for subsequent analyses on large-scale circulation and cluster contributions. We have also added an introductory paragraph to each section of the results part of the paper, emphasizing the rationale for the analysis and the analytical framework of the paper. We believe this approach greatly improved the clarity of our analytical framework and storyline.

We also implemented EOF based clustering for comparison purposes. The EOF analysis was applied to the whole Control and Experiment runs data points treating them as one time series. The first five leading EOFs were only used for spanning the reduced representation. The corresponding PC deriving from these EOFs serves as the features within the reduced representation. The five leading EOFs account for 64.7% of the total variance within our dataset. The centers of these classes, along with their monthly occurrence frequencies, are depicted in Figure 2 and Figure 3 here, respectively (Figures D5 and D6 in the revised manuscript). The class centers and occurrence frequencies resulting from this clustering approach are similar to those obtained by performing k-means clustering directly on the data space. We elaborated on this comparison in the Appendix D of the revised manuscript (also introduced in lines 266-268 of the revised manuscript).

Additionally, we included t-SNE visualizations on different data representations in the Appendix D to further argue the advantages of our chosen methodology over traditional approaches (see Appendix D of the revised manuscript- Figure D11). This visualization provided a compelling argument for the proficiency of our method by highlighting the distinct data embeddings and the insights they offer discussed in Appendix D of the revised manuscript.

We are confident that the addition of these analyses and the reorganization of our results section not only strengthened our argument but also significantly improved the overall quality of our manuscript.

[Figure]

*Figure 2: Mean MSLP fields for clusters derived from k-means clustering (6 clusters) on PC time series derived from the 5 leading EOFs. The fields for each cluster are plotted as anomalies relative to the mean MSLP field for all data points, encompassing both the Control and Experiment runs.*

[Figure]

*Figure 3: The monthly occurrence frequency of the 6 clusters showed in Figure 2 for the Control (blue) and Experiment (red) runs.*

*Other comments are following:*

> *B1-04 is similar with Fig. B3-01; however, their occurrence seasons are so different. Why? Same for the Fig. B5-05. Are they corresponding to the C1 and C2 in Fig.6? Even though the methods are different, the results should match between each other in certain degree.*

Thank you for pointing out the similarities and differences in occurrence seasons between the Figures B1, B3 and 6 of preprint version of the manuscript (corresponded to Figures D9, D3

and 9 in the revised manuscript, respectively). Your question highlights a key aspect of different clustering methodologies, which is the dependency of clustering outcomes on the chosen representation space. Different representation spaces can influence which data points are considered close to each other according to the specific geometry of that space (lines 872-885 of Appendix D in the revised manuscript). Clustering algorithms like k-means then group data points based on their proximity in the chosen representation space. For instance, the clustering performed on PMSET indices (Figure D9 of the revised manuscript) is guided solely by similarities in PMSET indices. That is, data points with similar PMSET indices lie closer together in this representation. This contrasts with the clustering performed on data space (Figure D3 of the revised manuscript), where the Euclidean pixel-wise distance between data points dictates the data points proximity and the resulting clustering.

As you mentioned, there are similarities between Class 04 in Figure D9 and Class 01 in Figure D3 of the revised manuscript, suggesting they could be perceived as negative phases of each other if viewed as EOF patterns. However, they have different monthly occurrence frequency. It is important to notice that they are not EOF patterns, nor do they originate from clustering with the identical focus, as discussed in the previous paragraph. Cluster 04 in Figure D9 of the revised manuscript is derived from clustering data points based on their similarities in PMSET indices, meaning that data points forming this cluster primarily exhibit similar PMSET characteristics. On the other hand, Cluster 01 in Figure D3 of the revised manuscript represents one of the mean states of winter circulation. It emerges from clustering based on data space, which primarily groups the seasonal cycle.

To identify similarities in patterns, one should compare clustering results that share a similar focus. For instance, k-means clustering on data space, as shown in Figures D3 of the revised manuscript, and clustering on the FAE latent space, Figures D7 of the revised manuscript, both primarily reveal the seasonal cycle as the dominant pattern within the dataset. Consequently, the class centers in these representations are somewhat comparable. A case in point is Cluster 06 in Figure D3 of the revised manuscript and Cluster 06 in Figure D7 of the revised manuscript can be perceive as similar clusters. Similarly, clustering based on PMSET indices (Figures D9 and D10 of the revised manuscript) and clustering on the MCAE latent space exhibit resemblances because both approaches utilize PMSET indices as a similarity measure at different levels. For example, Cluster C5 in Figure 9 of the revised manuscript and Class 04 in Figure D9 of the manuscript show some similarities.

The divergent outcomes from clustering across different representation spaces underscores the sensitivity of clustering outcomes to the representation space. Conventional clustering methodologies only focus on data point similarities using standard measures of similarity, such as Euclidean pixel-wise distance between data points or PCA-reduced space representation. One of the main contributions of this paper is to introduce a method that can produce a representation of the data points that can effectively distance the data points based on both the similarities among the data points and their alignment with a target objective similarity, here the PMSET patterns. We emphasized on this contribution in the revised version (lines 377-387, 421-426 and Appendix D of the revised manuscript) and further used t-SNE visualization to provide deeper insights into how various clustering methods perform across different representations (Appendix D).

*Line 231, why 40 indices in total? 4 latitudes with 5 PMSET values and two levels of troposphere? It is easy to miss the two levels of troposphere.*

Our decision to choose these specific indices was driven by a desire to comprehensively capture the three-dimensional spatial patterns of the PMSET associated with the data points, which is crucial for understanding the dynamics of the atmosphere energy transport in our study.

To elaborate, the five PMSET indices per profile are intended to represent the main characteristics of the PMSET profiles. Furthermore, the choice of four latitudes was aimed at capturing the latitudinal progression of the PMSET. Additionally, incorporating two vertical levels within the troposphere was intended to provide insights into the vertical dynamics of the PMSET. The two vertical levels allow us to observe how PMSET values change with altitude, offering insight into the vertical distribution of atmospheric energy transport.

We appreciate the opportunity to clarify this in our manuscript. We added this in line 237-238 of the revised manuscript.

*Line 349, how does the authors consider the "statistically significant increases" in Fig. 10?*

As we mentioned in line 123-126 of the revised manuscript, the independent two sample t-test and Welch's unpaired t-test were used throughout the study to determine the statistical significance levels when samples have equal or unequal variances, respectively, and samples with p-value less than 0.05 were considered statistically significant. Using the same methodology, the doted points in Figure 7 of the revised manuscript (Figure 10 in the preprint version) are consider statistically significant

*Line 359-361, "between the daily mean PMSET" of upper and lower troposphere?*

Exactly. We appreciate your highlighting this point. In the revised version of our manuscript, we refined this sentence to more precisely express that our analysis involves comparing the daily mean PMSET values between the upper and lower levels of the troposphere (lines 441-442 of the revised manuscript).

As a final remark, we would like to highlight the main contributions of our paper, which we better emphasized on the final revision of the manuscript:

1- We introduced a deep-learning-based method that generates an effective representation of data points. This representation accounts for both the similarities among the data points and their alignment with a target objective similarity, here the PMSET patterns, to effectively distance the data points. This method was applied to data points consisting of 8 consecutive days of MSLP and $\tau_{300-700}$, capturing dynamic patterns in the circulation data points.

2- We developed a formulation for class contribution, focusing on the dynamic interactions of different circulation patterns that lead to observed anomalies. By reformulating the anomalies to attribute them to different circulation groups, we identified two main mechanisms through which a circulation cluster can contribute to the anomaly: (1) through changes in the occurrence frequency of that cluster as a result of the forcing (FSDC), and (2) through slight adjustments in the cluster's mean characteristics as a result of the forcing (WCVC). This framework

delineates how changes in circulation patterns, whether in occurrence frequency or in the cluster's mean characteristics, contribute to the observed anomalies in key climate variables.

3- We utilized this advanced methodological approach to understand the role of different circulation patterns in forming the anomalies. These patterns were grouped based on similarities in their spatiotemporal behavior and their associated PMSET patterns. This grouping and the class contribution formulation helped us understand how these interactions lead to observed anomalies resulting from negative radiative forcing over Europe. Among the main findings are:

- In most cases, the change in a cluster's main behavior is more significant than changes in the occurrence frequency of that cluster when considering the cluster's contribution to the anomaly. For example, the decreased occurrence frequency of a cluster associated with a high-pressure system over Northern Eurasia and Scandinavia in autumn, observed in the Experiment run, led to reduced upward wave propagation. However, a slight adjustment in the mean behavior of this cluster in the Experiment run resulted in an increase in upward wave propagation. This increased wave propagation dominates the cluster's contribution to the observed anomaly in upward wave propagation (we incorporate this argument to the paper abstract).
- Our method's capability to attribute observed anomalies, such as the SIC loss in the Barents-Kara Seas in autumn, the warm Arctic and cold mid-latitude in winter, and others.

*Title: Arctic Climate Response to European Radiative Forcing: A Deep Learning Approach*

*Authors: Sina Mehrdad, Dörthe Handorf, Ines Höschel, Khalil Karami, Johannes Quaas, Sudhakar Dipu, and Christoph Jacobi*

*Summary: This study utilizes deep learning techniques to investigate the Arctic atmospheric circulation responses to European radiative forcing, which is both intriguing and significant for advancing our understanding of applying convolutional auto-encoder frameworks in Arctic climate change research. Overall, the manuscript is well-written, and the authors extensively discuss the consistent dynamical responses, although interpreting causality remains challenging. Below, I provide a few major and minor comments for the authors' consideration.*

Thank you very much for your insightful review. We greatly appreciate your recognition of the research topic's significance. We acknowledge your point regarding the challenges in interpreting causality within our study. Indeed, one of the key contributions of our work is to highlight that deriving causality in the context of regional forcing on Arctic climate is complex and multifaceted.

As you have pointed out our presentation of the results and the mechanisms derived from our analysis might not have been as clear as necessary. We revised the manuscript to better highlight these aspects. Specifically, we have enhanced the manuscript throughout by adding more detailed explanations where necessary to clarify the structure and storyline of the paper. The abstract is revised to contain the main contributions of the paper. The results section has been restructured to begin with a comparison of the mean states of the Control and Experiment runs, followed by analyses of large-scale circulation clustering and class contributions. Furthermore, we have introduced Appendices A and C in the revised manuscript and enriched the content and analysis in Appendix D.

*Major comments:*

*Baseline machine learning model comparison. This study presents an innovative application of convolutional autoencoders (AEs), which appears to be one of the first attempts in studying Arctic response to radiative forcing. However, I am still unclear about the motivation behind choosing convolutional AEs. For example, the authors mentioned that in Lines 538-539 that the FAE is a compelling method for generating a concise and informative representation, but did not elaborate on compare to what. Have the authors compared the performance of convolutional AEs with other machine learning or statistical methods? For example, similar clustering analyses could be conducted using self-organizing maps (SOMs), which are computationally less expensive than training convolutional AEs. SOMs have been used in studying atmospheric moisture transport in the Arctic or large-scale atmospheric circulations (e.g., Skific et al. 2009; Lee 2017). Additionally, principal component analysis (PCA)*

*or empirical orthogonal function (EOF) analysis could be employed for clustering tests. Why not start with these simpler methods before diving into complex deep learning models? However, if the authors can demonstrate that convolutional AEs outperform SOMs or PCA/EOF, it would strengthen the justification for using convolutional AEs in this study. Perhaps the authors could consider quickly implementing these simpler methods and comparing the results with those obtained from convolutional AEs.*

Thank you for these comments and the opportunity to clarify our motivation to choose our deep learning approach. Your insights have encouraged us to more thoroughly compare our deep learning method with other established conventional techniques.

Our initial motivation for employing a deep learning approach, particularly convolutional operations, comes from their documented success in meteorological applications, as outlined in lines 70-84 of our revised manuscript. The effectiveness of convolutional neural networks (CNNs) arises from the way they treat the dataset, taking into account the local context (such as 2D and 3D neighborhood structures) which, in our case, is crucial for understanding the spatial configurations of weather patterns and their changes over time. They further exploit hierarchical feature extraction. This focus is discussed in lines 198-206 in our revised paper. In particular, we highlight our motivation to employ a convolutional auto-encoder architecture, inspired by its success in weather prediction by Weyn et al. (2019, 2020). These studies illustrate the architecture's proficiency in capturing spatiotemporal patterns within weather datasets, a point we emphasize in lines 578-580 of our revised manuscript.

The flexibility of deep learning methods, which allows us to customize the latent space to align with our specific research goals, is a further motivation to use this approach in our research (line 244-245 of our revised manuscript). Specifically, our aim is for data points yielding similar PMSET patterns to lay more closely in the MCAE latent space, as detailed in lines 258-260 of the revised manuscript. In the discussion section lines 603-613, we evaluate the MCAE's latent space representation of the dataset. We compare this with the traditional data space representation, which is dominated by the seasonal cycle. Additionally, we compare it to the PMSET index representation, which focuses solely on PMSET similarities. This level of flexibility in data representation, unachievable with conventional methods, further motivate us to use the deep learning approach. Moreover, additional insights are gained by comparing the t-SNE visualization of different representations of the data space to the MCAE latent space, as shown in Figure D11 and Appendix D of the manuscript.

In the opening paragraph of the discussion (lines 580-592 of the revised manuscript), we aimed to assess the data representation obtained from the FAE. Particularly, in lines 588-592 of the revised manuscript, we compare unsupervised clustering results performed on the data space (k-means and SOM) and the FAE's latent space. This comparison reveals that the FAE's latent space allows us to discern datapoints corresponding to transitional seasons, such as autumn and spring. This observation leads to our concluding remark in lines 590-592 of the manuscript, where we state that the FAE offers an effective method for creating a concise and informative representation.

We acknowledge the importance of comparing our convolutional auto-encoders with simpler, less computationally expensive methods such as SOMs and EOF analysis. Therefore, we implemented unsupervised clustering using SOM and EOF and compare their performance with our clustering analysis (lines 262-277, and Appendix D of the revised paper). The EOF analysis

was applied to the whole Control and Experiment runs data points treating them as one time series. The first five leading EOFs were only used for spanning the reduced representation. The corresponding PC deriving from these EOFs serves as the features within the reduced representation. The five leading EOFs account for 64.7% of the total variance within our dataset. The centers of these classes, along with their monthly occurrence frequencies, are depicted in Figure 4 and Figure 5 below, respectively (Figures D5 and D6 in the revised manuscript). Moreover, we configured the SOM to categorize the data into six classes, arranged in a 3x2 grid, over 100 iterations. Due to the sequential nature of SOM training, leveraging the parallel processing capabilities of GPUs offers challenges, necessitating reliance on the sequential processes of CPUs. Consequently, the SOM training was time-consuming, requiring approximately one week on our local server, due to the extensive dataset and its large input shape. This extensive training time limited our ability to fine-tune the SOM parameters to optimize performance. The class centers of these SOM classes, along with their monthly occurrence frequency are illustrated in Figure 6 and Figure 7 below, respectively (Figures D1 and D2 in the revised manuscript). The class centers and occurrence frequencies resulting from these conventional clustering approaches are similar to those obtained by performing k-means clustering directly on the data space. We elaborate on this comparison in the Appendix D of the revised manuscript.

Additionally, we included t-SNE visualizations on different data representations in the Appendix D (Figure D11 of the revised manuscript) to further argue the advantages of our chosen methodology over traditional approaches. This visualization aims to provide a compelling argument for the proficiency of our method by highlighting the distinct data embeddings and the insights they offer. This addition will not only enhance the robustness of our argument but also provide a comprehensive evaluation of the convolutional auto-encoder effectiveness in capturing complex spatiotemporal patterns in response to the excreted. The argument in this regard is added to Appendix D of the revised manuscript.

We appreciate your constructive feedback, which has guided us to further validate our methodology. We believe that these additional comparisons emphasize the importance of using convolutional autoencoders in our research.

[Figure]

*Figure 4: Mean MSLP fields for clusters derived from k-means clustering (6 clusters) on PC time series derived from the 5 leading EOFs. The fields for each cluster are plotted as anomalies relative to the mean MSLP field for all data points, encompassing both the Control and Experiment runs.*

[Figure]

*Figure 5: The monthly occurrence frequency of the 6 clusters showed in Figure 4 for the Control (blue) and Experiment (red) runs.*

[Figure]

*Figure 6: Same as Figure 4, but the clusters were derived from applying SOM clustering on the data space.*

[Figure]

*Figure 7: The monthly occurrence frequency of the 6 clusters showed in Figure 6 for the Control (blue) and Experiment (red) runs.*

*What new physical or dynamical insight do we learn? I am curious about the new insights or knowledge gained from the new clustering method employed in this study. Lines 566-567 mention that well-established large-scale circulation patterns (e.g., NAO, AO, PNA) are identified, and consistent dynamical responses in the troposphere and stratosphere can be demonstrated. I assume that similar conclusions may be drawn from other clustering methods as well. Could we uncover new dynamical pathways in which the Arctic responds to European radiative forcing differently from previous understandings based on stratosphere-troposphere coupling? It would be helpful if the*

*authors could create a table summarizing the dynamical responses associated with each cluster, indicating which dynamical responses are already known and which are new. Similarly, Lines 324-327 discuss the seasonality changes. What do these seasonality changes signify physically, and what can we learn from them?" In addition, what is the separation of WCVC and FSDC components brings us new insights?*

Thank you for your insightful feedback. We revised section 3.2 and Appendix D of the revised manuscript in response to this. To respond to your comment first we need to note a key aspect of different clustering methodologies, which is the dependency of clustering outcomes on the chosen representation space. Different representation spaces can influence which data points are considered close to each other according to the specific geometry of that space. Clustering algorithms like k-means then group data points based on their proximity in the chosen representation space. For instance, the clustering performed on PMSET indices (Figure D9 of the revised manuscript) is guided solely by similarities in PMSET indices. That is, data points with similar PMSET indices lie closer together in this representation. Conventional clustering methodologies only focus on data point similarities using standard measures of similarity, such as Euclidean pixel-wise distance between data points or PCA-reduced (Principal Component Analysis) space representation. One of the main contributions of this paper is to introduce a deep-learning based method that generates an effective representation of data points. This representation accounts for both the similarities among the data points and their alignment with a target objective similarity, here the PMSET patterns, to effectively distance the data points. (this argument is highlighted on lines 421-426 and Appendix D of the revised manuscript)

In the offered representation, both the similarity in data points' spatiotemporal patterns and the associated PMSET pattern play a role in the data points' proximity in the representation and consequently determine the clustering results. The expression in lines 620-621 emphasizes the fact that with a focus on the similarity in data points spatiotemporal patterns and the associated PMSET pattern, we were able to have patterns with a similar focus as the well-established circulation regime. While similar patterns might be recognizable through other clustering techniques, our method enhances the interpretability and relevance of these patterns by grounding them in the specific context of PMSET dynamics. Thus, the new insight we offer is not merely the identification of these patterns but a deeper understanding of their dynamical significance, facilitated by a clustering approach that integrates physical relevance with data-driven techniques (the clustering motivation and deviation from other clustering method is emphasized in lines 377-387 and Appendix D of the revised manuscript)

Thank you for pointing out the need for a more detailed discussion on the physical significance of seasonality changes and the insights gained from separating the WCVC and FSDC components in our analysis. Our findings indicate that the applied forcing may not necessarily induce new circulation patterns. Instead, we explored how alterations in the grouped circulation regimes within the Experimental run have contributed to observed anomalies in critical Arctic climate variables. We attribute these changes within the grouped circulation regimes to the applied forcing, as it is the only transient perturbation in the Experiment run.

One of the main goals of the paper is to identify the contributions of various classes to the anomalies observed as a consequence of the applied forcing. We tried to understand the dynamic interactions of each pattern that led to the observed anomaly. In addressing them, we developed our class contribution formulation. We reformulated the anomalies to attribute them to different circulation groups. We identified two main mechanisms through which a circulation cluster can contribute to the anomaly: 1) through changes in the occurrence frequency of that cluster as a result of the forcing (FSDC), and 2) through slight adjustments in the cluster's mean

characteristics as a result of the forcing (WCVC). This framework helps us delineate how changes in circulation patterns, whether in occurrence frequency or in the cluster's mean characteristics, contribute to the observed anomalies in the key climate variables. This approach not only enhances our understanding of the dynamical interactions leading to the observed anomalies but also offers valuable insights into the complex ways in which atmospheric circulation responds to external forcing. It underscores the importance of considering both the occurrence and the mean characteristics of circulation patterns in assessing their impact on climate anomalies (this has been more emphasized in lines 399-413 of the revised manuscript).

Regarding uncovering new dynamical pathways, we utilized this advanced methodological approach to understand the role of different circulation patterns in forming the anomalies (see lines 321-324 of the revised manuscript). These patterns were grouped based on similarities in their spatiotemporal behavior and their associated PMSET patterns. This grouping and the class contribution formulation helped us understand how these interactions lead to observed anomalies resulting from negative radiative forcing over Europe. Among the main findings are:
- In most cases, the change in a cluster's main behavior is more significant than changes in the occurrence frequency of that cluster when considering the cluster's contribution to the anomaly. For example, the decreased occurrence frequency of a cluster associated with a high-pressure system over Northern Eurasia and Scandinavia in autumn, observed in the Experiment run, led to reduced upward wave propagation. However, a slight adjustment in the mean behavior of this cluster in the Experiment run resulted in an increase in upward wave propagation. This increased wave propagation dominates the cluster's contribution to the observed anomaly in upward wave propagation (we incorporate this argument to the paper abstract).
- Our method's capability to attribute observed anomalies, such as the SIC loss in the Barents-Kara Seas in autumn, the warm Arctic and cold mid-latitude in winter, and others.

We highlighted this argument more in the revised version. Moreover, we tried to put all the most meaningful results in a more compact format in Table A3 of the revised manuscript. This approach will not only make the information more concise but also emphasize our contributions more effectively. These modifications will undoubtedly make the paper more accessible and emphasize the novelty and significance of our contributions. In Table A3, however, we did not specify whether these results were previously known because most of the proposed mechanisms involve a combination of different interactions. In the discussion section of the paper, we evaluate our class contribution patterns alongside proposed mechanisms. For example, the mechanism linking sea ice loss in the Barents-Kara Seas during autumn to disturbances in the polar vortex or relating the blocking over Scandinavian region to the disturbed polar vortex (lines 666-693 of the revised manuscript), and the role of high pressure over Eurasia in Arctic temperature variations observed in the C1 class contribution (lines 657-665 of the revised manuscript). We also discuss the ability of our methodology to identify potential mechanisms behind each significant anomaly. Some examples include:

- Lines 634-644 of the revised manuscript: Our method successfully attributed the SIC loss in Northeast Canada during autumn to mechanisms hidden in seasonal mean anomalies. This has been more highlighted at the end of the paragraph (lines 641-644)
- Lines 645-656 of the revised manuscript: This section illustrates our method's ability to attribute an anomaly associated with a single cluster to changes in both the occurrence frequency of the cluster and the main characteristics of the class. Ultimately, we can explain how these changes, driven by the forcing, impact the observed anomaly.

- Lines 657-665 of the revised manuscript: We attributed the 'warm Arctic and cold midlatitude' pattern to changes in the main characteristics of C1.
- Lines 666-693 of the revised manuscript: C3 is identified as the main factor contributing to sea ice loss in the Barents-Kara seas during autumn, primarily driven by discrepancies in class characteristics in the Experimental run. This sea ice reduction coincides with a disturbed polar vortex in the lower stratosphere, aligning with increased disturbances in stratospheric dynamics. Although a reduction in C3's occurrence frequency in autumn slightly decreases upward wave propagation, this effect is minor compared to the dominant increase in wave propagation driven by changes in the class's mean characteristics. Overall, the influence of C3 on stratospheric dynamics, though pivotal, is part of a broader interplay of various factors, indicating that its singular contribution is less dominant than the collective influence of other classes. Some sentence parts are added to the paragraph to make argument more clear (lines 671-673,and 685-690).
- Lines 694-699 of the revised manuscript: This section provides an example of how different classes interact to form the observed anomaly, particularly, focus on how different clusters contribute to forming the stratospheric dynamics in autumn.
- Lines 710-724 of the revised manuscript: We discuss our method's limitations in identifying mechanisms and offer possible explanations for these limitations.

*Linking the responses to European radiative forcing. I noticed that the discussion on the results seems to focus less on the direct response to radiative forcing and more on the subsequent atmospheric circulation responses and PMSET. For example, how does the European radiative forcing lead to increased upward EP flux for cluster 3 in SON (Figure 17)? Or how does the localized radiative forcing in Europe give rise to changes in 2m temperature across the entire Northern Hemisphere, as depicted in Figure 13? Some of the temperature increases appear contradictory to the cooling effect of aerosol negative forcing (or specifically here, the cloud forcing).*

Thank you for highlighting the need for a clearer connection between the response to European radiative forcing and the observed atmospheric circulation change in our analysis. We considered the large-scale circulation regime as the main mechanism by which our local forcing influences remote regions such as the Arctic. This is why we performed clustering to group the similar large-scale circulation regimes, and analyzed the impact of changes in these grouped circulation regimes on the observed anomaly, as a consequence of the applied forcing. Given our focus on Arctic climate variables, we utilized the PMSET pattern associated with each circulation as an additional target similarity measure for our clustering. The PMSET pattern, indicative of how each circulation regime transfers energy into the Arctic, is crucial for determining Arctic climate conditions (this argument is added to lines 319-324 and 377-387 of the revised manuscript).

Our analysis reveals that while the forcing may not introduce new circulation patterns, it can modify existing circulation patterns in two significant ways: by altering the occurrence frequency of a pattern (the FSDC component of the class contribution) or by changing the mean characteristics of the circulation pattern (the WCVC component of the class contribution). Consequently, we calculated each class's contribution to the observed anomaly. As mentioned in line 300 and equation 2 of the revised manuscript, the sum of these class contributions yield the observed anomaly. Each class's contribution can then be decomposed into two parts: WCVC and FSDC. This decomposition enables us to attribute different aspects of the anomaly to specific clusters and to the particular type of change within that cluster, whether it's a change

in frequency or in mean characteristics. In response to your comment, We added the argument in lines 319-324 and 377-387 to enhance the manuscript to more explicitly draw these connections between European radiative forcing and its effect on Arctic climate via changes in atmospheric circulation.

Thank you for your insightful questions about the specific impacts of European radiative forcing on different atmospheric dynamics. These phenomena indeed highlight the complex and nonlinear responses of the climate system to localized radiative forcing. We demonstrated that the increased upward EP flux for C3 during SON is primarily attributed to the adjustment in the mean characteristics of this cluster in the experimental run (WCVC component), despite the reduced occurrence frequency of this cluster, which is commonly known to decrease upward wave propagation. However, this reduction is not the dominant factor in this class's contribution (for more details, see lines 666-693). Furthermore, we attributed the observed warm Arctic and cold midlatitude pattern to changes in the mean characteristics of C1, as detailed on lines 657-665 of the manuscript.

As expected, the forcing does not always yield uniform results; its effects differ depending on the present circulation regime. Unlike linear analyses, which evenly attribute the anomaly across all clusters or data points, our approach captures the nonlinear behavior of the climate system. This means that the forcing leads to different contributions by each class depending on how the forcing change the cluster behavior, and the sum of these contributions is equal to the observed anomaly (see equation 2).

Thanks to your feedbacks, we acknowledge that the presentation of this content within the paper was not sufficiently clear. In general we revised the manuscript structure to better represent our analytical framework of the paper.

*Minor comments:*

> *Lines 58-60: perhaps the authors considering to cite two new studies on this topic: Xu et al. (2023) and Liang et al. (2024).*

Thank you very much for recommending the studies by Xu et al. (2023) and Liang et al. (2024). We agree that these recent contributions are highly relevant to our work and can enrich the context of the manuscript. We incorporated these studies in the revised version of our manuscript (lines 48, 57, 60, 673, and 688 of the revised manuscript).

> *Line 123 and Figure 1: why there are statistically significant radiative forcing increase in eastern Siberia, Asia, and North Pacific?*

You've raised an interesting point. These increases are indeed noteworthy and could be attributed to dynamical feedback mechanisms related to the atmospheric circulation change by European radiative forcing. While we acknowledge the importance and potential implications of these observed increases, a detailed exploration of their causes and consequences lies beyond the current scope of our research.

> *Lines 161-162: why 8 days? Does this indicate a certain physical process dominating?*

The choice of an 8-day period for our data points was motivated by a combination of methodological considerations and preliminary testing to identify a timeframe that captures dynamic atmospheric processes. In our initial tests, we observed that an 8-day period yielded the best performance regarding the DL algorithm reconstruction loss and overfitting measure (line 166 of the revised manuscript). This performance could be due to its efficiency to capture atmospheric dynamics or just related to our deep learning architecture. The choice of 8 days aligns well with our deep-learning architecture, particularly the use of 3D convolutional pooling and upsampling within our autoencoder architecture. The binary nature of the number 8 ($2^3$) potentially enhances processing efficiency and effectiveness within this architecture.

*Line 230: the maximum PMSET? Or both maximum and minimum PMSETs?*

Thank you for your keen observation. Indeed, our analysis considers both the maximum and minimum PMSETs, not solely the maximum PMSET. We corrected this typo in the revised manuscript (line 236 of the revised manuscript).

*Lines 246-248: could the authors provide a figure to demonstrate this grid transformation?*

Thank you for suggesting the inclusion of a figure to demonstrate the transformation applied to angular features in the MCAE's latent space (explained in lines 252-255 of the revised manuscript). To address the continuity challenge between angles close to 0 and 360 degrees, we indeed transform each angular feature into two dimensions using sine and cosine functions. This approach effectively preserves the circular nature of angular measurements, ensuring that values close in angular space (for example, 355° and 5°) remain proximate in the transformed two-dimensional space, despite their apparent numerical distinction.

For instance, an angular feature represented by a value within the 0-360 degree range can create discontinuity issues where angularly adjacent values appear numerically distant. By converting each angle into a pair of coordinates, using the sine and cosine of the angle, we create a two-dimensional representation where the proximity of angles accurately reflects their true angular relationship. In Figure 8, points A and B are represented by their angles, θ1 and θ2, respectively. Despite θ2 being numerically far from θ1, their two-dimensional representations (using sine and cosine) on the unit circle are close, accurately reflecting the angular proximity.

We initially opted for a written description of the transformation in the paper. However, in response to your valuable feedback, we added Appendix C, which explain this transformation. Figure C1 in Appendix C of the revised manuscript visualizes the transformation process, enhancing clarity and aiding understanding for our readers.

[Figure]

*Figure 8 visualization of the transformation applied to angular features to address the angular continuity challenge*

**Lines 298 and 301: Are there any reference papers for WCVC and FSDC?**

The concepts of the Within-Cluster Variability Contribution (WCVC) and Frequency-weighted Seasonal Deviation Contribution (FSDC) introduced in our study represent novel contributions to the field. These were developed as part of our class contribution formulation, aimed at examining the dynamic interactions of circulation patterns and their roles in producing the observed anomalies in the Experiment run. This approach allows us to attribute anomalies to specific changes within circulation clusters, either through variations in their occurrence frequency (FSDC) or adjustments in their mean characteristics (WCVC) in response to radiative forcing.

Given the innovative nature of this framework, there are no direct reference papers that discuss WCVC and FSDC in the context we have applied them. To the best of our knowledge, our study is the first to formalize these concepts as distinct mechanisms through which atmospheric circulation clusters can influence climate anomalies.

In section 2.6 of our manuscript, we comprehensively explained these concepts. We believe that our work may serve as a foundation for future research in this area, potentially encouraging further studies that attribute anomalies to different components of the system.

**Figure 8: it seems the signal-to-noise ration is quite small. How could the authors say these responses are important?**

As emphasized in lines 414-420 of the manuscript, in Figure 11 of the revised manuscript (Figure 8 in preprint version), our primary objective is to demonstrate the effectiveness of our clustering methodology in identifying and classifying similar atmospheric patterns across different simulations and throughout different seasons. The class centers were calculated relative to the seasonal mean fields depicted in the figure's first row. The low variation in class center patterns across seasons underscores the robustness and consistency of our clustering approach, even without explicitly incorporating information about the seasonal cycle into the network.

Regarding the signal-to-noise ratio mentioned, it's crucial to recognize that the subtlety of the variations among the class centers highlights the method's sensitivity and precision in capturing

the dynamics of the climate system. We appreciate this opportunity to clarify the intention and findings related to Figure 8.

*Figures 9 and Figure 10: combine these two figure into one figure?*

Thank you for suggesting the combination of Figures 6 and 7 (corresponding to Figures 9 and 10 in the preprint version) in the revised manuscript. We understand the potential benefits of consolidating these figures for a more streamlined presentation. However, after careful consideration, we have decided to keep these figures separate. Figure 6 presents the climatology, while Figure 7 focuses specifically on the anomalies. Combining these figures could potentially obscure the distinct and critical information each aims to convey, particularly given the differing nature of the data they represent. We appreciate your feedback and hope this explanation clarifies our rationale for the layout choices in our revised manuscript.

*Figure 12: the arrows are hardly seen. Perhaps the authors can try to enhance the visibility of the arrows.*

Thank you for pointing out the visibility issue with the arrows in Figure 13 of the revised manuscript (Figure 12 in preprint version). We appreciate your suggestion and enhanced the visibility of the arrows in the revised version of the figure.

*Lines 458-460: but the seasonal distribution changes?*

Thank you for bringing attention to the subtle change in the seasonal distribution of sudden stratospheric warming (SSW) occurrences. Table A4 in the revised manuscript indicates a subtle shift in the seasonal distribution of SSW occurrences in the Experiment run, with a decrease by one occurrence in autumn and a compensatory increase by one occurrence in winter.

As illustrated in Figure 16 of the revised manuscript, the autumn season shows an increase in the zonal mean zonal wind in the high-latitude upper stratosphere and a decrease in the lower stratosphere. This pattern of variability does not directly correlate with the slight decrease in SSW occurrences in autumn, suggesting that the observed changes in zonal wind patterns cannot solely be attributed to the frequency of SSW events.

Similarly, during winter, we observed an increase in high-latitude middle atmospheric zonal mean zonal wind, coinciding with an increase in occurrences of SSW events. Generally, an increase in SSW events would be expected to have a converse effect, namely a decrease in the zonal mean zonal wind. Therefore, attributing the observed change in wind patterns directly to the singular additional SSW occurrence is not straightforward.

This analysis leads us to conclude that while the seasonal distribution of SSW occurrences experiences minor adjustments, these changes do not significantly alter our findings regarding our upper atmospheric analysis. This argument was added to the revised manuscript in lines 350-355.

*Figure 17: the EP flux divergence does not exactly match the pattern of zonal wind anomalies in some seasons and clusters. How could we relate the EP flux change to zonal wind change?*

Thank you for highlighting this subtle observation. It is important to note that the EP flux divergences are the momentum deposited by the resolved waves in the model. In addition to the resolved Rossby waves the changes in the zonal wind can be influenced by the non-resolved parameterized gravity waves (GWs). Changes in the zonal mean zonal wind due to the resolved waves may affect the parameterized momentum depositions by the GWs. This in turn, could affect the prevailing zonal flow and induce changes in the residual circulation below the breaking level. Such dynamics might explain the anomalies of the zonal mean zonal wind far away from the regions of significant EP fluxes (Cohen et al., 2013; Limpasuvan et al., 2016; Chandran et al., 2014). We appreciate your insightful comment and will further clarify this in our revised manuscript. This argument was added to line 528-535 of the revised manuscript.

*Lines 540-542: the authors mentioned that the external forcing can modify the circulation patterns. But in Lines 778-680, the authors conclude that the radiative forcing only alter the existing circulation patterns, and does not introduce new patterns. These two sentences seem contradictory somewhat. Could the authors clarify and reconcile these statements?*

Thank you for highlighting the need for clarification between our discussion on the general effects of external forcings on circulation patterns and our specific findings related to negative radiative forcing over Europe.

In the general context (lines 593-595), we mention "External forcings can modify circulation patterns in complex and nonlinear ways (Gillett and Fyfe, 2013; Hannachi et al., 2017) by introducing new spatiotemporal patterns, changing the preferred circulation patterns, altering their frequencies, or by a combination of them." However, when focusing on the specific context of negative radiative forcing over Europe (lines 741-744), we mentioned "Our study revealed that negative radiative forcing over Europe, resembling the heterogeneous radiative forcing exerted by aerosols, influences the climate system by altering existing circulation patterns and their frequencies without introducing new patterns." Therefore, while external forcings, in general, have the capacity to induce a wide range of modifications in circulation regimes, our research specifically found that negative radiative forcing over Europe modifies the circulation by adjusting existing patterns and their frequencies, rather than introducing entirely new circulation patterns.

*Line 689: upper troposphere lower stratosphere —> upper troposphere and lower stratosphere?*

Thank you for pointing out this error. We corrected it in the revised version of the manuscript (line 753).

*References:*

*Xu, M., Tian, W., Zhang, J., Screen, J.A., Zhang, C. and Wang, Z., 2023. Important role of stratosphere-troposphere coupling in the Arctic mid-to-upper tropospheric warming in response to sea-ice loss. npj Climate and Atmospheric Science, 6(1), p.9.*

*Lee, C.C., 2017. Reanalysing the impacts of atmospheric teleconnections on cold-season weather using multivariate surface weather types and self-organizing maps. International Journal of Climatology, 37(9), pp.3714-3730.*

*Liang, Y.C., Kwon, Y.O., Frankignoul, C., Gastineau, G., Smith, K.L., Polvani, L.M., Sun, L., Peings, Y., Deser, C., Zhang, R. and Screen, J., 2024. The Weakening of the Stratospheric Polar Vortex and the Subsequent Surface Impacts as Consequences to Arctic Sea Ice Loss. Journal of Climate, 37(1), pp.309-333.*

*Skific, N., Francis, J.A. and Cassano, J.J., 2009. Attribution of projected changes in atmospheric moisture transport in the Arctic: A self-organizing map perspective. Journal of Climate, 22(15), pp.4135-4153.*

We also made references to the following papers::

Chandran, A., Garcia, R. R., Collins, R. L., & Chang, L. C.: Secondary planetary waves in the middle and upper atmosphere following the stratospheric sudden warming event of january 2012. Geophysical Research Letters, 40 (9), doi: https://doi.org/10.1002/grl.50373, 2013.

Cohen, N.Y., Gerber, E.P., Bühler, O.: Compensation between resolved and unresolved wave driving in the stratosphere: Implications for downward control. Journal of the Atmospheric Sciences 70, 3780 – 3798, doi:10.1175/JAS-D-12-0346.1, 2013

Gillett, N. and Fyfe, J.: Annular mode changes in the CMIP5 simulations, Geophysical Research Letters, 40, 1189–1193, https://doi.org/10.1002/grl.50249, 2013.

Hannachi, A., Straus, D. M., Franzke, C. L., Corti, S., and Woollings, T.: Low-frequency nonlinearity and regime behavior in the Northern Hemisphere extratropical atmosphere, Reviews of Geophysics, 55, 199–234, doi: 10.1002/2015RG000509, 2017.

Limpasuvan, V., Orsolini, Y. J., Chandran, A., Garcia, R. R., & Smith, A. K.: On the composite response of the mlt to major sudden stratospheric warming events with elevated stratopause. Journal of Geophysi-cal Research: Atmospheres, 121 (9), 4518-4537, doi: 10.1002/2015JD024401, 2016.

Weyn, J. A., Durran, D. R., and Caruana, R.: Can machines learn to predict weather? Using deep learning to predict gridded 500-hPa geopotential height from historical weather data, Journal of Advances in Modeling Earth Systems, 11, 2680–2693, doi:10.1029/2019MS001705, 2019.

Weyn, J. A., Durran, D. R., and Caruana, R.: Improving data-driven global weather prediction using deep convolutional neural networks on a cubed sphere, Journal of Advances in Modeling Earth Systems, 12, e2020MS002 109, doi:10.1029/2020MS002109, 2020.

---

## Referee Report (RR1)

**Title:** Arctic Climate Response to European Radiative Forcing: A Deep Learning Approach

**Authors**: Sina Mehrdad, Dörthe Handorf, Ines Höschel, Khalil Karami, Johannes Quaas, Sudhakar Dipu, and Christoph Jacobi

I would like to thank the authors for their efforts in addressing my comments. The structure and storyline of the manuscript have been improved, allowing for a clearer delivery of the results. However, regarding causality, I am hesitant about the statement the authors made in their replies: 'Deriving causality in the context of regional forcing on Arctic climate is complex and multifaceted.' In a coupled climate system, it seems nearly impossible to clearly distinguish causality. Is it necessary to provide causality, or should one acknowledge that deriving it is very difficult, and/or not necessary? Overall, I am satisfied with this version but have a few additional comments for the authors to consider.

For the machine learning approach, I think using a convolutional auto-encoder architecture to study this topic is both exciting and promising. I also appreciate that the authors provided baseline models (e.g., SOM and EOF) for comparison. However, extending this point further, what new scientific insights can these methods bring to us? For example, the authors mentioned that 'We compare this with the traditional data space representation, which is dominated by the seasonal cycle.' These results may seem straightforward to most meteorologists or climate scientists, as some of their research is based on signals after removing the seasonal cycle. Additionally, the stratosphere-troposphere coupling has been well-known for decades. Does using deep learning reveal something new? Related to this, the authors stated that 'a high-pressure system over Northern Eurasia and Scandinavia in autumn, observed in the Experiment run, led to reduced upward wave propagation.' Is this a new finding that previous studies did not document? The authors may want to elaborate more on this aspect and highlight the new scientific findings from this manuscript, as well as those not found in previous studies.

For the forcing and its connection to remote impacts, the authors argue that the large-scale circulation regime is the main underlying mechanism. My follow-up question is: how can one justify that the forced signal stands out from internal variability, given that large-scale circulation is intrinsically affected by internally-driven components? Many previous studies have stressed out the large internal variability could mask out the forced signal. The authors also mentioned that the forcing does not always produce uniform results. Resonating with previous comments, the authors could emphasize the role of machine learning approaches in this context and highlight what new insights these methods bring to the table and deal with the non-stationary results from the same forcing.

---

## Author Response (AR2)

We thank the referees for the insightful comments, which have greatly helped us improve the manuscript. Below, we have reiterated the reviewers' remarks in red italics, followed by our corresponding responses in standard black text. The references to specific lines and figures pertain to the provided diff file.

**Reviewer 1:**

*This paper examines how the European radiative forced responses can impact Arctic climate by using a combination method of two machine learning techniques, the k-mean clustering and convolution neural network. Specifically, the authors classified six patterns and discussed how these six patterns responds to the European radiative forcing and to impact the Arctic. This paper is interesting, and the topic is crucial for the community, especially the method constraining the k-mean clustering for auto-encoder with PMSET. Therefore, I do suggest this paper to be published in Weather and Climate Dynamics.*

We greatly appreciate your positive feedback and recognition of the significance of our topic to the community. We are pleased that you found our methodology, which leverages the flexibility of Deep Learning (DL) to provide a framework for analyzing climate patterns, to be of interest.

*Other point-by-point comments are following (Lines are from highlighted change version):*

*1. Lines 10-11, "the negative forcing over Europe"? Negative radiative forcing? Less radiation?*

Thank you for your remark. We revised it to "negative radiative forcing" in line 10 of the diff file.

*2. Lines 18-19, "the shifts in the mean characteristics of the atmospheric circulation patterns"? Not clear. What does it mean the shifts inf the mean characteristics? Spatial? Temporal? Or others?*

Thank you for highlighting the unclear statement. We mainly refer to shifts in spatial characteristics. We have revised the sentence in line 13 of the diff file to specify that it refers to spatial characteristics.

*3. Lines 159-160, how other data fields or using only one of the two fields may change the results? Please not simply use "might" but test with simple cases.*

You raise a crucial point regarding the impact of including additional meteorological fields in our analysis. Indeed, understanding how much information can be gained by adding another field is important. This can be effectively analyzed using self-supervised methods in the context of data imputation. However, the primary focus of this study was not to quantify the amount of informational content one can gain for our study's purpose by adding various data fields. Instead, motivated by the work of Weyn et al. (2019) and Weyn et al. (2020), we specifically used the fields of MSLP and $\tau_{300-700}$ for our study. While the impact of adding or removing selected data fields has been analyzed in their studies for forecast contexts, our choice to use these two fields was driven by the aim to capture the most information regarding large circulation regimes with the minimum number of data fields. This has been discussed in the first paragraph of section 2.3.1.

*4. Why separate Figure 6 and 7? If so, why not showing the 2m temperature and SIC mean state in Figure 6 as well?*

Thank you for your comment. Initially, we aimed to keep the climatology and anomaly in separate figures. We did not include the 2m temperature and SIC mean state in Figure 6 because we anticipated that the 2m temperature climatology would show reduced temperatures toward the pole, and the SIC climatology is somewhat represented in the anomaly with the green outline. For the PMSET, we included the climatology to observe how the control run produces PMSET climatology. However, we agree that adding the 2m temperature and SIC fields and combining the climatology and anomaly figures would enhance clarity and simplify the analysis. Therefore, we have merged the two figures and added the climatology for 2m temperature and SIC. Please see Figure 6 and lines 327-333 and 340 in the diff file for these updates. Thank you for your valuable suggestion.

*5. Lines 338-340, "The anomalies in Figure 7 were calculated by subtracting the respective mean fields of the Control from those of the Experiment run". Does this refer to the mean state differences between the Experiment and Control run in annual mean?*

Thank you for highlighting this point. Yes, the anomalies in Figure 6 of the diff file represent the annual mean differences between the Experiment and Control runs. We have revised the text for clarity in lines 330-331 of the diff file.

*6. Line 342, "an inverse correlation"? Does it mean pattern correlation statistically? Or not mathematically, just saying a similar but opposite values? Same in Line 351.*

Thank you for your question. In this context, explaining the patterns in Figure 6 of the diff file, we meant that the patterns in the figures exhibit similar but opposite values. Thus, we did not perform a statistical test for this part, and the term "inverse correlation" was used descriptively rather than mathematically. We added "qualitatively" in lines 335 and 345 of the diff file to make this more clear.

*7. Lines 392-396, are these a repetition of the method section?*

Thank you for your comment. We acknowledge that this section contains a component of repetition. However, we believe it is necessary to maintain the storyline clearly and emphasize the connection between the methodology and the results. Including this repetition in the results section helps the reader keep track of the paper's narrative and better understand the methodological decisions. Therefore, we propose to keep this section as it is.

To evaluate the significance of these changes, we considered two factors: the consistency of clustering results independent of the k-means clustering initialization, and the intermittency of the observed signal. For the former, we performed k-means clustering 1,000 times with different initializations. The resulting changes in occurrence frequencies are reported in Table A2, which shows consistency in the change in occurrence frequencies between the Control and Experiment runs. To evaluate the intermittency of the signal, we implemented a bootstrapping strategy in the revised version. This involved creating 1,000 ensembles for each simulation by resampling with replacements from the Control and Experimental runs. The results have been incorporated into Figure 9 of the diff file and described in lines 405-407. These bootstrapped results also support the consistency of the changes in the clusters' occurrence frequencies.

***Reviewer 2:***

*Title: Arctic Climate Response to European Radiative Forcing: A Deep Learning Approach*

*Authors: Sina Mehrdad, Dörthe Handorf, Ines Höschel, Khalil Karami, Johannes Quaas, Sudhakar Dipu, and Christoph Jacobi*

*I would like to thank the authors for their efforts in addressing my comments. The structure and storyline of the manuscript have been improved, allowing for a clearer delivery of the results. However, regarding causality, I am hesitant about the statement the authors made in their replies: 'Deriving causality in the context of regional forcing on Arctic climate is complex and multifaceted.' In a coupled climate system, it seems nearly impossible to clearly distinguish causality. Is it necessary to provide causality, or should one acknowledge that deriving it is very difficult, and/or not necessary? Overall, I am satisfied with this version but have a few additional comments for the authors to consider.*

Thank you very much for your comments and perspective. Regarding the sentence you referred to, the term "derive" was perhaps not perfectly chosen; "inferring" more accurately describes our intention in the sentence making it 'Inferring causality in the context of regional forcing on Arctic climate is complex and multifaceted'. This complexity arises from the inherent uncertainties in climate modeling and the chaotic nature of the climate system itself which can be observed in the uncertainties in modeling forecast abilities beyond a few days. While it is true that climate systems exhibit long-term unpredictability, they also exhibit deterministic behavior, which allows for the study of causality within certain limits. Many studies use statistical methods to infer potential causal relationships. While these methods do not necessarily predict specific outcomes, they help identify relationships and potential causal influences between variables. These sorts of analyses like our study in this paper, are valuable for hypothesis generation and enhancing our understanding of the climate system, which may subsequently be explored further through more detailed modeling or experimental interventions. In response to your query, we believe that while proving causality in the strictest sense might be exceedingly challenging, attempting to infer and discuss potential causality is crucial for advancing our theoretical framework and guiding future research.

*For the machine learning approach, I think using a convolutional auto-encoder architecture to study this topic is both exciting and promising. I also appreciate that the authors provided baseline models (e.g., SOM and EOF) for comparison. However, extending this point further, what new scientific insights can these methods bring to us? For example, the authors mentioned that 'We compare this with the traditional data space representation, which is dominated by the seasonal cycle.' These results may seem straightforward to most meteorologists or climate scientists, as some of their research is based on signals after removing the seasonal cycle.*

*Additionally, the stratosphere-troposphere coupling has been well-known for decades. Does using deep learning reveal something new? Related to this, the authors stated that 'a high-pressure system over Northern Eurasia and Scandinavia in autumn, observed in the Experiment run, led to reduced upward wave propagation.' Is this a new finding that previous studies did not document? The authors may want to elaborate more on this aspect and highlight the new scientific findings from this manuscript, as well as those not found in previous studies.*

Thank you for expressing the value you find in using convolutional autoencoders in this analysis and in comparing it to the baseline methods. The question you raised is indeed critical, as it seeks to uncover the new scientific insights that these methods can provide. In our view, the capability of Deep Learning (DL) models to extract complex patterns from datasets holds significant promise for climate applications, though it is still at its early stage. DL models offer unprecedented flexibility, allowing us to direct the model's focus on specific aspects of the data we deem important. This project represents an effort to harness this potential in a meaningful way.

In this study, we categorize the circulation patterns based on a target variable that we consider important for our task. Because we are interested in analyzing the effect of circulation on the Arctic climate, we selected the 3D pattern of the associated PMSET as this target variable in our approach. That is, we direct our DL model (MCAE) focus to prioritize the relationship between circulation patterns and PMSET. The MCAE is trained to identify similarities in its input domain, which includes MSLP and layer thickness from 700 to 300 hPa, that correlate with similar PMSET outcomes. The flexibility of the DL models offers various post-training analysis methods; for instance, modifications in the latent space representation can be explored through the decoder to assess changes in data space representation or the convolutional neural network (CNN) kernel weights can be examined to understand decision-making processes within the model. In our analysis, we focused on examining the model's latent space as the post-training analysis. Initially, we identified six distinct density points within MCAE's latent space and subsequently analyzed the similarities within each of these groups to discern what the model emphasizes.

To comprehensively evaluate the performance of the MCAE, we must assess the model's reconstruction loss and its ability to represent both the circulation and PMSET patterns within its latent space. The training objective of the MCAE was to capture the structure of circulation data points and their associated PMSET in its latent space. Evaluation of the presence of the PMSET's patterns in the latent space of MCAE is straightforward and involves comparing clustering results within the MCAE latent space against those derived from a representation solely based on PMSET indices. This comparison demonstrates that the MCAE latent space retains characteristics of the PMSET indices to a certain extent. However, evaluating the representation of circulation data points in the MCAE latent space structures presents more challenges. We utilized the seasonal cycle as the metric for this purpose, given its dominance in the data space representation. Although there is a weak dependency of the seasonal cycle on PMSET indices, it is still the predominant pattern of the circulation data points, thereby serving as a crucial metric for evaluating how well the data space structure is represented in the MCAE latent space.

We retained the seasonal cycle within the dataset to assess how various data representations and methods capture this dominant pattern. Traditional clustering methods primarily categorized data based on the seasonal cycle alone, highlighting their limitations in detecting finer nuances. It was also the case when clustering on FAE latent space. However, its latent space demonstrated a more nuanced distinction by efficiently discriminating between transitional seasons, reflecting the superior quality of its feature extraction capabilities. In contrast, the MCAE also effectively managed the seasonal cycle, incorporating it into its latent space while regulating its domination based on its importance for the task. Notably, with the integration of PMSET target information, the MCAE preserved the main structural elements of the data and accounted for similarities in associated PMSET, showcasing its advanced capability in capturing both the inherent data structure and metadata nuances.

As highlighted earlier, employing DL models for climate diagnostic applications represents a promising research avenue in climate science. This work has demonstrated several advantages of applying DL models to these applications. Even though these advantages have been detailed in the different parts of the paper, we acknowledge the need to wrap all these up in a concluding manner. Thus, we have modified the conclusion section (see lines 742-754 and lines 788-791), which summarizes the key benefits of our DL model discussed here, providing a clear overview and emphasizing the potential impact on future research in the field.

Although in this work we did not focus on stratospheric coupling in deriving the latent space of MCAE and just focused on the PMSET associated with the circulation patterns, we still yielded noteworthy insights into tropospheric-stratospheric interactions. With the developed class contribution formulation, which is independent of the DL model, we attributed observed anomalies to changes in clusters behavior during experimental runs. Notably, as highlighted in your question, a cluster associated with a high-pressure system over Northern Eurasia and Scandinavia (C3) in autumn plays a role in upward wave propagation. Our quoted sentence in the context is as follows. "The decreased occurrence frequency of a cluster associated with a high-pressure system over Northern Eurasia and Scandinavia in autumn, observed in the Experiment run, led to reduced upward wave propagation. However, a slight adjustment in the mean behavior of this cluster in the Experiment run resulted in an increase in upward wave propagation". Each cluster can change in the Experiment run relative to the Control run in two primary ways: 1) changes in its seasonal occurrence frequency, and 2) minor alterations in the cluster's mean pattern. Our findings show that a reduced occurrence frequency of cluster C3 in autumn is associated with decreased upward wave propagation. This correlation is consistent with the established understanding of the influence of similar circulation patterns on upward wave dynamics. However, the extent to which this reduction in frequency shapes upward wave propagation remains an open question. Our results indicate that even minor adjustments in the mean pattern of this cluster can have a more significant effect on upward wave propagation than changes in its seasonal occurrence frequency. This analysis enables a detailed comparison of how variations in C3's occurrence frequency and adjustments in its mean pattern each contribute to the overall anomaly observed in upward wave propagation. Furthermore, our methodology offers a quantitative approach to assess the contributions of different clusters to the observed anomaly in upward wave propagation, marking a significant advancement in the analysis of complex systems such as the climate. These points have been extensively discussed in lines 671-699 and 766-780 of the diff file, which we refined further for enhanced clarity.

*For the forcing and its connection to remote impacts, the authors argue that the large-scale circulation regime is the main underlying mechanism. My follow-up question is: how can one justify that the forced signal stands out from internal variability, given that large-scale circulation is intrinsically affected by internally-driven components? Many previous studies have stressed out the large internal variability could mask out the forced signal. The authors also mentioned that the forcing does not always produce uniform results. Resonating with previous comments, the authors could emphasize the role of machine learning approaches in this context and highlight what new insights these methods bring to the table and deal with the nonstationary results from the same forcing.*

Thank you for highlighting this crucial aspect of the analysis. Differentiating the forced signal from internal variability, especially in large-scale circulation responses, is indeed challenging. In our study, we acknowledge the limitations posed by not having access to a very large ensemble of simulations. However, our strategy involved 30 years of climate simulation, which, while not extensive, is significant.

For the anomaly fields, we employed the t-test to define statistically significant anomalies. Additionally, to evaluate the robustness of the k-means clustering in determining seasonal occurrence frequencies, we conducted k-means clustering 1,000 times with random initializations, as detailed in Table A2 of the manuscript. However, we recognize the need to further assess the robustness of the clusters' seasonal discrepancies between the Control and Experiment runs (as illustrated in Figures 10 and E1 of the diff file), as well as the cluster monthly occurrence frequencies (Figure 9). These elements are pivotal to our class contribution analysis within this study.

To address the intermittency of observed signals, we implemented a bootstrapping strategy in the revised manuscript, creating 1,000 ensembles for each simulation (lines 405-407 of the diff file). This method involved resampling with replacement from the Control and Experimental runs, and the results have been incorporated into Figures 9 and E1 of the diff file. Importantly, these bootstrapped results support the consistency of our general conclusions. Figures 9 of the diff file now includes error bars representing the standard deviation of the monthly occurrence frequency for each cluster at the top of the bar charts. These error bars are typically smaller than the differences in monthly occurrence frequencies between the Control and Experiment runs, particularly during seasons with high differences. Figure E1 of the diff file illustrates the clusters' seasonal discrepancies between the Control and Experiment run across the 1,000 bootstrapped ensembles for both simulations (line 907 of the diff file). Compared to the last version, the areas without statistical significance are highlighted with dotted in Figure E1, and the main patterns remain consistent and show statistical significance. An exception is the summer discrepancy for cluster C4, attributed to its extremely low occurrence frequency during this season. In some ensemble members, C4 is absent, rendering the pattern statistically insignificant.

As previously discussed, our findings indicate that the impact of the forcing varies depending on the existing circulation regime, underlining the system's non-uniform response. Our approach diverges from linear analyses by capturing the climate system's nonlinear behavior, showing that different circulation clusters react distinctly to the same forcing. Such insights

underscore the advantages of our methodology, which can handle the complex responses of the climate system to external forcing. This nuanced understanding stems from our class contribution formulation, which, while independent of DL models, benefits significantly from the use of a machine learning algorithm for clustering. These points have been elaborated upon in lines 755-759 of the diff file.

As noted, the MCAE is developed to make sure that the circulation data points with similar patterns and similar associated PMSET are grouped together in the clustering. This enhances the quality of clustering ensuring it closely meets our research objectives. However, the primary purpose of employing DL in our study was to refine data classification and facilitate anomaly analysis, rather than to generate or resample model ensembles. The focus of the DL algorithm was not on resampling, down sampling, or generating model ensembles based on the simulations at hand. These methods, which focus on addressing the signal-to-noise challenges in climate forcing responses, remain a compelling motivation for further research. Nevertheless, our methodology has successfully captured the nonlinear behavior of the climate system and attributed observed anomalies to each circulation class, thereby facilitating the climate forcing analysis.

**References**

Weyn, J. A., Durran, D. R., and Caruana, R.: Can machines learn to predict weather? Using deep learning to predict grid1140 ded 500-hPa geopotential height from historical weather data, Journal of Advances in Modeling Earth Systems, 11, 2680–2693, https://doi.org/10.1029/2019MS001705, 2019.

Weyn, J. A., Durran, D. R., and Caruana, R.: Improving data-driven global weather prediction using deep convolutional neural networks on a cubed sphere, Journal of Advances in Modeling Earth Systems, 12, e2020MS002 109, https://doi.org/10.1029/2020MS002109, 2020.